# Probabilistic tsunami forecasting for early warning

J. Selva [1✉], S. Lorito [2], M. Volpe[2], F. Romano [2], R. Tonini [2], P. Perfetti[1], F. Bernardi [2], M. Taroni[2], A. Scala[3], A. Babeyko[4], F. Løvholt[5], S. J. Gibbons [5], J. Macías [6], M. J. Castro[6], J. M. González-Vida[6], C. Sánchez-Linares[6], H. B. Bayraktar [2,3], R. Basili [2], F. E. Maesano [2], M. M. Tiberti [2], F. Mele[2], A. Piatanesi[2] & A. Amato[2]

Tsunami warning centres face the challenging task of rapidly forecasting tsunami threat immediately after an earthquake, when there is high uncertainty due to data deficiency. Here we introduce Probabilistic Tsunami Forecasting (PTF) for tsunami early warning. PTF explicitly treats data- and forecast-uncertainties, enabling alert level definitions according to any predefined level of conservatism, which is connected to the average balance of missed-vs-false-alarms. Impact forecasts and resulting recommendations become progressively less uncertain as new data become available. Here we report an implementation for near-source early warning and test it systematically by hindcasting the great 2010 M8.8 Maule (Chile) and the well-studied 2003 M6.8 Zemmouri-Boumerdes (Algeria) tsunamis, as well as all the Mediterranean earthquakes that triggered alert messages at the Italian Tsunami Warning Centre since its inception in 2015, demonstrating forecasting accuracy over a wide range of magnitudes and earthquake types.

[1] Istituto Nazionale di Geofisica e Vulcanologia, Bologna, Italy. [2] Istituto Nazionale di Geofisica e Vulcanologia, Rome, Italy. [3] Department of Physics "Ettore Pancini", University of Naples, Naples, Italy. [4] German Research Centre for Geosciences (GFZ), Potsdam, Germany. [5] Norwegian Geotechnical Institute (NGI), Oslo, Norway. [6] Grupo EDANYA, Universidad de Málaga, Málaga, Spain. ✉email: jacopo.selva@ingv.it

Tsunamis may strike a coastal population close to the earthquake location within minutes after its origin time. Tsunami Early Warning Systems (TEWS) must forecast the tsunami threat rapidly following any potentially tsunamigenic earthquake. Tsunami impact prediction immediately after the event is subject to large uncertainty stemming mainly from the unknown details of the earthquake source, which implies large variability in the estimated tsunami inundation[1]. The uncertainty is amplified by the necessity to act rapidly to maximize the evacuation lead time. Given the available information, a vast number of different forecast outcomes are possible. The forecasts should assign a probability to each of these outcomes (like in, for example, weather forecasting[2,3]). Present-day tsunami forecasts are non-probabilistic, producing single-outcome forecasts. The uncertainty is often accommodated only implicitly through conservative choices (e.g. safety factors) to minimize missed alarms, at the cost of increasing the rate of false alarms[4]. Supplementary Table 1 summarizes all the symbols and acronyms used.

For sufficiently distant earthquakes, tsunami forecasts can be constrained with moment tensors[5], yet these forecasts are still characterized by significant uncertainty. Deep-sea sensors, where available, can further help constraining the tsunami through inversion and data assimilation techniques[4,6–10]. However, locally, the tsunami may inundate after minutes[11] and initial tsunami forecast must be performed solely from basic earthquake parameters. Innovative rapid source estimation techniques are steadily progressing[12–19], and next-generation sensors and methods could bring dramatic improvements to reduce uncertainties[7,20–23]. Yet, some uncertainty sources are intrinsic[24,25], and the earthquake and tsunami characteristics may be surprising and remain elusive even years after the event[26–28]. Therefore, uncertainty quantification and reduction efforts must be synergistically undertaken.

The need to deal with uncertainty in early warnings has been long recognized[29,30] and recently emphasized also for TEWS[31,32]. Following the 2004 Indian Ocean tsunami, the cost of "insist(ing) on certainty" was highlighted[33]. Despite subsequent attempts to define methods to quantify tsunami forecast uncertainty[34–38], operational tsunami forecasting in TEWSs is still non-probabilistic (http://www.ioc-tsunami.org/). Specifically, Tsunami Service Providers (TSPs) worldwide adopt Decision Matrices (DMs, look-up tables linking earthquake parameters with alert levels) or Envelopes (ENVs, selecting a local maximum over a selection of scenarios), or consider one or a few Best-Matching Scenarios (BMSs, scenarios matching the seismic and/or tsunami data available at the time of the estimation) to define initial alert levels[16,32,39–45]. Specific strategies (e.g. maximum credible magnitude, safety factors, etc.), usually rooted in the analysis of past events, are sometimes adopted to implicitly replace uncertainty quantification[16,32], but TSPs do not yet apply any formal probabilistic method. For example, a proxy of existing uncertainty is sometimes derived from the statistics of the scenarios selected with ENV methods [e.g. [44]) or of the along-coast variability of the forecast [e.g. [45]). However, in this way, the tsunami forecast cannot be tested quantitatively against observations and consequently the procedure cannot be calibrated[46,47]. In addition, the commonly adopted safety measures generally tend to overestimate the forecasts, although underestimations may still occur[43].

The use of a single estimation of the tsunami intensity to define alert levels, typical of non-probabilistic forecasts, also mixes to some extent scientific tsunami forecasts with political decision making. For example, a safety factor introduces a positive bias in the forecast to reduce the missed alarms rate: this is not done to improve the accuracy of the forecast, but to reduce potential societal consequences, which is a typical decision-making task. The decision-making process requires competences beyond the field of tsunami science. It is therefore considered fundamental to have effective and transparent uncertainty communication from scientists to decision makers[48–51] to make the process more traceable and to optimize the risk-reduction management[33,51–56]. For fast evolving phenomena like tsunamis, this can be realized adopting pre-defined rules, to be used automatically during an emergency.

An effective and transparent communication of uncertainty may be realized through Probabilistic Tsunami Forecasting (PTF). The PTF workflow should allow for a full propagation of uncertainty, from the earthquake hypocentre and magnitude estimation to alert-level definition, accounting for all the available information at the time of the estimate. This also clarifies the separation between scientific components (uncertain tsunami forecast through hazard curves) and political duties (alert-level definition for risk mitigation), following the hazard-risk separation principle[57,58]. This strategy is similar to the one used for seismic risk reduction: scientists determine the probability of different shaking intensities in the target area in a given time window (e.g. 50 years), and decision makers define seismic building codes selecting a design exceedance probability[59]. Similarly, the rule of conversion from PTF to alert levels can be defined by the authorities-in-charge by selecting a target probability value (e.g. one particular percentile), corresponding to a pre-defined level of conservatism for risk-reduction actions. This separation, enabled by the uncertainty quantification, is becoming a standard also for long-term coastal planning against tsunamis[60–63] and tsunami building code definition[64].

In this work, PTF is introduced and applied to a wide range of past events to discuss the feasibility of its real-time application and to test it against observations. To illustrate its potentiality for tsunami warning, we define alert levels from PTF based on different probability thresholds corresponding to different levels of conservatism, and we compare the results with the alert levels that would have been obtained applying a range of current-practice non-probabilistic methods. We demonstrate that PTF is statistically accurate in its forecasts for a wide range of events, from relatively small crustal events to large magnitude subduction zone earthquakes. We show that PTF can be timely produced also for near-field tsunami warning and that, adopting real-time conversion rules established in advance, probabilistic forecasts accounting for real-time uncertainty can be transparently transformed into alert levels, allowing to implement any desired level of conservatism based on all the available information at the time of the estimation.

## Results

**The PTF workflow**. The procedure introduced here, coined Probabilistic Tsunami Forecasting (PTF), explicitly quantifies the uncertainty in real-time forecasts and enables uncertainty-informed alert-level definition in operational tsunami early warning (Fig. 1). PTF can provide the probability distribution of a Tsunami Intensity Measure (TIM, e.g. maximum run-up or near-coast wave amplitude) at multiple forecast points almost immediately, as soon as an earthquake location and magnitude estimates are available, typically few minutes after origin time (Fig. 1a). The method rigorously embeds uncertainty in tsunami forecast at the time of the estimate by quantifying the probability distribution for one (or more) TIM at each forecast point (Fig. 1c). The quantification is managed through an ensemble of tsunami scenarios defined by a set of sources weighted by the probability of being consistent with available real-time observations (Fig. 1a; e.g. seismic, geodetic, tsunami), as well as with local

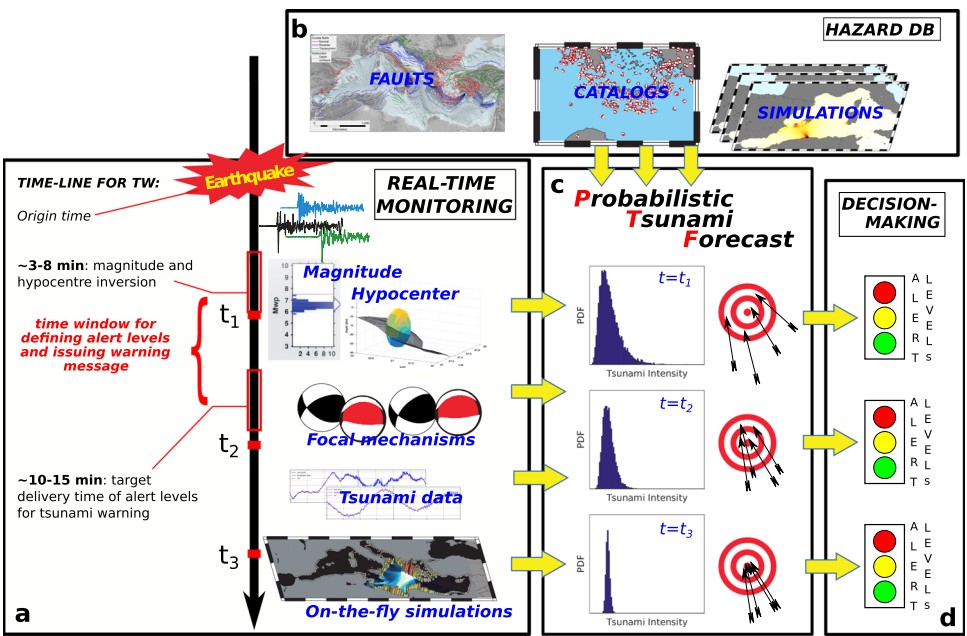

**Fig. 1 PTF concept. a** Timeline for tsunami warning: real-time information from an earthquake that just occurred and from the ongoing tsunami gradually integrates **b** local long-term hazard information, **c** progressively increasing the precision of the probabilistic forecasts (hazard curves) produced by the Probabilistic Tsunami Forecasting (PTF). **d** At any time, PTF can be transformed into alert levels (here represented as traffic lights) useful for decision making. In the current study, implementation refers to the time $t_1$, when only earthquake magnitude and hypocentre estimates are available from real-time observations.

earthquake and tsunami hazard information (Fig. 1b; e.g., pre-computed tsunami scenarios, long-term frequencies) derived from hazard and/or other long-term forecast models. PTF can be refined continuously with updated information (i.e., seismic moment tensor, tsunami data, Fig. 1a) to reduce the uncertainty in the forecasts (Fig. 1c). The evolving probability distributions can be used to define at any time, according to pre-defined rules, alert levels for specific points/areas (Fig. 1d), which in turn correspond to actions for risk reduction (for example, evacuation).

We implement PTF for near-field tsunami warning, that is, for sites proximal to the earthquake epicentre. This is a challenging task for TEWS[16]. To define the needs for near-field tsunami warning, we take as reference tsunami warning in the Mediterranean Sea. Here, seismically induced tsunamis always originate relatively close to some coastline, and tsunami inundation often occurs minutes after the earthquake. To maximize lead times, TSPs in the NEAMTWS (North-eastern Atlantic, the Mediterranean and connected seas Tsunami Warning System) currently adopt Decision Matrices (DMs, e.g., http://www.ioc-tsunami.org/), with a target delivery time of 10–15 min after earthquake occurrence (Fig. 1a). With this tight temporal constraint, while the seismic hypocentre and magnitude probabilities can be computed from real-time earthquake data to input PTF, faulting geometry and mechanism probabilities are not yet available. However, this missing information can be derived from long-term seismo-tectonic constraints. Considering that hypocentre and magnitude solutions are typically available in 3–8 min[42], the target delivery time of alert-levels can be matched by PTF with computational times on the order of a few minutes (e.g. <2 min, Fig. 1a).

For the Mediterranean Sea, long-term source information is derived from the regional hazard database NEAMTHM18[65–67], which assumed that earthquakes may occur in principle everywhere in the Earth's crust. Thus, NEAMTHM18 provides a database of sources covering the entire Mediterranean Sea with any potential mechanism. For any given target event, an ensemble

of sources and corresponding probability consistent with both real-time and past observations (as expressed by NEAMTHM18 focal mechanism probability[66,68]) can be defined starting from real-time information. Using the NEAMTHM18 pre-computed database of tsunami simulations, the sources in the ensemble are propagated to the forecast points (Supplementary Fig. 1) through numerical tsunami simulations. The hazard is quantified combining source probabilities and tsunami propagation, including an additional basic treatment of tsunami modelling uncertainty accounting for approximations in source, propagation, and inundation[39]. Maximum wave amplitude extrapolated at 1 m depth (hereinafter near-coast wave amplitude) is selected as the TIM. The PTF computational time correlates with the ensemble size, which can be controlled by adopting cut-offs on source probabilities. Testing four different cut-offs, we found that a cut-off of 2 standard deviations offered a good compromise between stability of the results and computational time (<2 minutes, see Supplementary Table 4), matching the target response time for the warning (Fig. 1a). So, while computation times can be lowered further by code optimization, PTF can be applied in its present configuration to any possible source in the Mediterranean Sea, satisfying response-time demands for its operational use for NEAMTWS.

This PTF implementation can be extended to any other source area by (i) defining a database of potential sources covering the selected target area adopting the same strategy used in NEAMTHM18 for the Mediterranean, and (ii) using a workflow for high-performance computing[69] to produce all the simulations required in the ensemble of sources. The details of the PTF implementation can be found in Methods.

**PTF for the 2003 Mw 6.8 Zemmouri-Boumerdes earthquake.** To illustrate the PTF workflow, we first consider the 2003 Mw 6.8 Zemmouri-Boumerdes earthquake (Fig. 2) that occurred on the Tell-Atlas fold-and-thrust belt (likely on a south-dipping fault), triggering a tsunami causing damage at several harbours in the

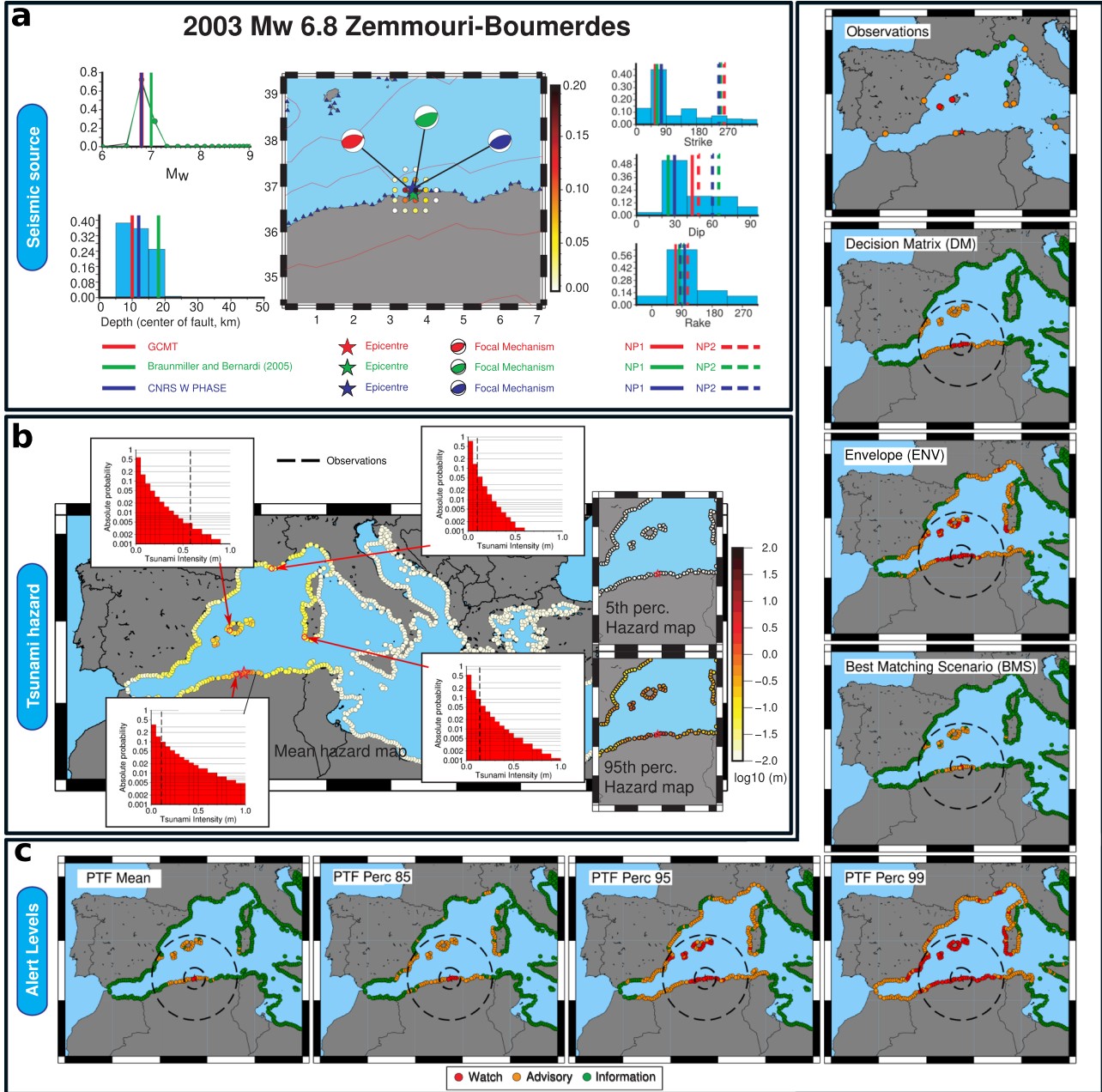

**Fig. 2 PTF workflow: example for the 2003 Zemmouri-Boumerdes tsunami. a** PTF source model: marginal distributions for earthquake magnitude and depth (left), location (centre), and fault parameters (right) for the ensemble describing source variability. Several revised moment tensor solutions are plotted as vertical lines for comparison. Distributions are consistent with seismic observations. For example, the ~30° southern-dipping realistic fault plane is strongly emphasized in the PTF source ensemble. **b** PTF results: tsunami intensity measure distribution (hazard curve) at four selected coastal locations in the western Mediterranean compared with observations (dashed vertical lines), and hazard maps involving all forecast points derived from different PTF's statistics (mean, 5–95th percentiles), showing uncertainty and spatial pattern of the tsunami forecast. **c** NEAMTWS Alert levels assigned from observations[40], decision matrix (DM), best-matching-scenario (BMS), envelope (ENV), and PTF mean, and 85, 95, and 99th percentiles; dashed lines indicate local and regional areas, as defined in the DM (Supplementary Table 8). NEAMTWS considers three alert levels (Information, Advisory, and Watch), each corresponding to off-coast tsunami wave amplitudes intervals: alert levels are assigned comparing tsunami near-coast wave amplitude with alert-level intervals.

western Mediterranean[9,70–72]. PTF is implemented in hindcasting mode, retrospectively simulating a real-time application. Real-time data (hypocentre and magnitude; Supplementary Table 2) are reconstructed using standard CAT-INGV operating procedures[42] on archived data (details in Supplementary Note 1). The resulting discrete joint distributions (Fig. 2a) for hypocentre, faulting geometry and mechanism are consistent with the most recent moment tensor estimations (Supplementary Table 3,

refs. [5,19,73–75]). Marginal distributions for strike, dip, and rake angles emphasize the expected geometry and mechanism for an event at that location, based on the local seismotectonics derived from the long-term hazard model[66–68]. The fault plane ambiguity is correctly resolved with the south-dipping reverse fault more probable than the conjugate plane.

For this event, the ensemble of sources is composed of approximately 15,000 scenarios (Supplementary Table 4). The

results are visualized through probability density functions for the selected TIM at each forecast point (Fig. 2b). TIMs and relative uncertainties are visualized through conditional hazard maps, whereby the mean or percentiles of the probability distributions are mapped. Despite combining a large number of scenarios, the forecast impact pattern is largely controlled by the dominant source orientation and by the tsunami propagation and generally agrees with observations. The specific observations can be compared with PTF distributions summarizing the expectations at each specific point. For the four locations reported Fig. 2b, all observations fall inside the PTF distributions. The tsunami observed in the Balearic Islands was relatively larger than expected and thus is in the right tail of the PTF distribution.

Alert levels are then assigned directly from PTF distributions (Fig. 2c). Different methods can be defined based on PTF statistics and/or on evaluating the probability of pre-defined TIM intervals (see Supplementary Note 2). In the NEAMTWS, three alert levels (Information, Advisory, and Watch) are defined corresponding to near-coast tsunami wave amplitudes that are negligible (we here assume <10 cm), 10–50 cm, >50 cm respectively (or twice these values for maximum run-up). Alert levels for each location are here assigned by comparing a TIM derived from the PTF with the relevant amplitude intervals. Different statistics of the PTF (e.g. the mean or a given percentile) can be used to extract this value, leading to alternative definitions of alert levels. Overall, we adopt the simplest method for illustrative purposes: mapping PTF statistics into alert levels' reference intensity intervals, which is equivalent to the definition of probability thresholds for long-term hazard (see Supplementary Note 2)[60–64].

To discuss PTF alert-level assignments, we take as reference three methods representative of standard non-probabilistic operational procedures to define alert levels. As reference for conservative methods, we consider (i) the Decision Matrix (DM) adopted by the Italian tsunami warning centre CAT-INGV (representative for the NEAMTWS operational procedures, see Supplementary Note 3), and (ii) an envelope (ENV) method resembling the one described in Catalan et al.[44], in which the maximum tsunami wave amplitude is selected at each coastal site from a set of scenarios compatible with the ongoing event. We consider all scenarios within half fault length (derived from[76,77]) from the epicentre and with magnitude best approximating the available magnitude incremented by 0.5. As reference for single simulation methods (current practice in many TSPs worldwide[32,39–41]), we consider a single Best-Matching Scenario (BMS) selected as the most likely source in the PTF ensemble, whose simulation results are used to define alert levels.

For 2003 Zemmouri-Boumerdes, DM and ENV-based alert levels tend to be more conservative than those based on the PTF mean or on the BMS (Fig. 2c). DMs associate alert levels with forecast points depending on earthquake location and magnitude through a discontinuous, decreasing function of the distance from the epicentre (see Supplementary Note 3). Thus, DMs do not consider that both source orientation and bathymetry control the tsunami propagation pattern and features, which is evident also for this event[70,71]. ENV, BMS, and PTF-based alert levels on the other hand embed the tsunami propagation footprint through numerical simulations. BMS results are comparable to PTF central values (e.g. the mean, Fig. 2c), ENV results to the high percentiles of the PTF. All non-probabilistic methods produce specific alert-levels, while PTF allows specification of a desired level of caution through choosing higher or lower percentiles, corresponding to TIMs with a high or low probability of exceedance. Consequently, the overall spatial extent of and the number of high alert levels (i.e. advisory/watch) is controlled by the selected percentile (the higher the percentile, the larger the

affected area), with high percentiles including less likely larger TIMs from the tail of PTF distributions. Figure 2c shows that several observations correlate better with conservative simulation-based methods like ENV and high-percentile PTF alert levels (e.g. 95th percentile): the reason is that this event challenged numerical modellers due to basin and harbour-related amplifications that occurred for instance in the Balearic Islands harbours[71]. Either higher resolution tsunami modelling is introduced, or only a conservative definition of alert levels can then include these values.

To examine more closely the reliability of PTF TIM forecasts, we compare PTF distributions directly with all the available observations (Fig. 3). Direct observations for this tsunami include data from several coastal sea-level stations (hereinafter, tide-gauge data) in the western Mediterranean[70–72]. The time-series are, however, few and coarsely sampled[40–42]. To enrich the comparison, we also consider other indirect observations and hind-casted models. Several moment tensors and finite-fault model estimates are available in the literature (Supplementary Table 3). A spatially homogeneous tsunami dataset for the test can be obtained simulating the tsunami from such available finite-fault models[78–84], retrieved by separate or joint seismic and geodetic data inversion (details in Supplementary Note 4). These data collectively sample our best assessment of the epistemic uncertainty of the source process almost two decades after the earthquake. The numerical simulations map this source uncertainty onto a synthetic tsunami dataset.

The maximum near-coast wave amplitude simulated from finite-faults models (red lines) generally falls within PTF's inner confidence intervals (defined through the 5–95th percentile interval), and the means (red and black solid lines) are highly clustered (we note that PTF distributions are not necessarily Gaussians and percentiles are here used to define confidence intervals). This agreement indicates that, while our PTF implementation simplifies the source representation (since NEAMTHM18 scenarios use uniform slip for crustal faults), the source variability in the PTF ensemble and the log-normal distribution we use to quantify the uncertainty embed the tsunami source uncertainty, as quantified by the range of available finite-fault models[46,78–84].

Conversely, observations at the tide-gauges are more scattered (yellow squares in Fig. 3). Several observations from Eastern Spain, the Balearic Islands, and western Italy fall into the tails of the PTF distributions. The misfits of some local maxima of the observations are present for both the PTF's central values and numerical simulations from best-fit source models. These misfits are probably due to the above-mentioned basin and harbour-related amplifications that likely occurred in several areas[71], and that cannot be reproduced without high-resolution tsunami numerical modelling. As they fall inside the upper tail of the PTF distributions, only the alert level corresponding to conservative choices (high percentiles of the PTF) include such maxima, resulting in a better correlation with the observations noted above (Fig. 2c). This demonstrates that even the relatively simple uncertainty model implemented to manage uncertainty in tsunami generation and propagation (see "Quantification of PTF's propagation factor" in Methods) can deal to some extent with these hard-to-predict amplifications, leading to forecasts that can encompass observations within uncertainty bounds. In the future, forecast precision may be improved through more advanced techniques to better quantify local amplifications and related uncertainty[63,85–91]. Notably, also other potential sources of local deviations exist, for example, the contribution of seismically induced landslides. While significant efforts in these directions are ongoing, research is still required to fully implement such methods in near-field real-time forecasts[92,93].

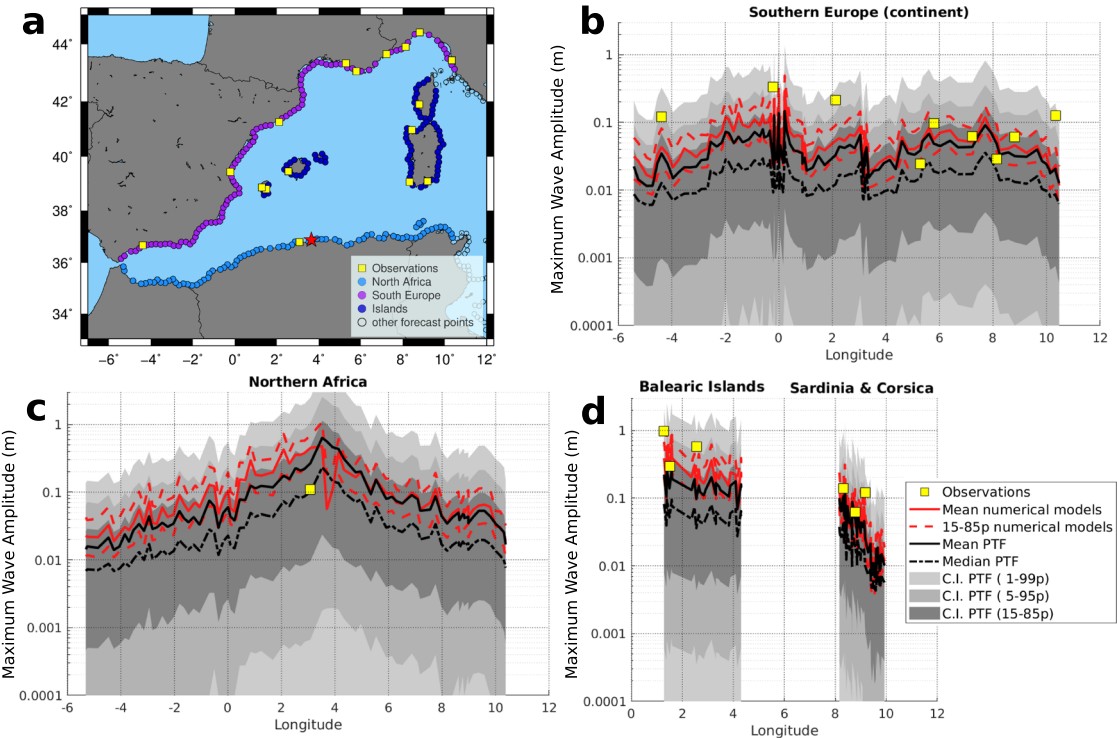

**Fig. 3 PTF for the 2003 Zemmouri-Boumerdes tsunami. a** selection of forecast points for specific comparison and **b–d** graphical comparison between tsunami observations (yellow squares) and maximum wave amplitudes evaluated from numerical models (red lines: mean and 15–85 percentiles with solid and dashed lines, respectively), and PTF statistics (black lines: mean and percentiles with solid and dashed lines, respectively) at all forecast points in **b** northwest Africa, **c** southwest Europe, and **d** the main islands.

**PTF for the 2010 Maule Mw 8.8 earthquake**. To illustrate PTF behaviour for larger magnitudes, we implement the PTF also to the NEAMWave17 ICG/NEAMTWS exercise scenario, a synthetic Mw 8.5 earthquake on the Hellenic Arc in southwestern Greece (see Supplementary Note 6), and to the 2010 Maule, Chile, Mw 8.8 earthquake[94] (Fig. 4). The latter required the extension of PTF implementation to cover the Chilean subduction zone (see Methods)[95]. For such large magnitudes, the source model includes in the ensemble only subduction earthquakes (Supplementary Table 4) with heterogeneous slip distributions also featuring shallow slip amplification[65–67,95,96], mimicking to some extent tsunami earthquakes (events generating a tsunami larger than expected from seismic magnitude[96,97]).

The 2010 Maule event provides the opportunity to compare PTF results with a larger and higher-quality dataset of tsunami observations, including coastal and deep-sea tsunami sensors (DART and tide-gauges) and run-up data[98,99] (Fig. 4). To compare with tsunami amplitude at the coast, run-up data are halved ([100] and reference therein). The results for this event show that PTF inner confidence intervals (15–85th percentiles) encompass all the observations, including run-ups (Fig. 4e,f), despite their possibly relatively large measurement errors. This result is coherent with the results of Catalan et al.[44], who show that the scenario envelope includes observations. The prediction at tide-gauges (Fig. 4c,d) shows a slight tendency towards overestimation, which remains within the uncertainty bounds. For the much smaller Mw 6.8 Zemmouri-Boumerdes event in the Mediterranean, we observed an opposite tendency towards underestimation. A possible reason is that, for smaller earthquakes on steeper faults like this, local resonances and amplification play a more important relative role due poorly modelled smaller tsunami wavelengths; for the Mw 8.8 Maule event, shelf and basin resonances occur also at longer periods[101], but they are well-captured on a 30 arc-sec grid (see Methods).

Moreover, the scenarios in the PTF ensemble of large magnitudes (Mw > 8.1[66]) are modelled on the 3D subduction geometry and with randomly sampled slip distributions (see Methods; the smaller crustal scenarios discussed above are instead modelled with simplified planar-fault uniform-slip sources). Consequently, the tsunami modelling uncertainty (accounting for tsunami generation, propagation, and inundation simplifications[102,103]) might be slightly overestimated in this case, as source representation is more advanced for such magnitudes, then compensating some underestimation due to local tsunami effects. This possible slight overestimation is also present when focussing on DART, even if may be less pronounced due to the larger source-target distance (Fig. 4a,b). Notably, a systematic extension to more case studies with extended high-quality observations may allow, in the future, a finer tuning of the adopted uncertainty modelling in each of the PTF factors, for example, using the large set of tsunami observations that is available in the Pacific Ocean[46,47].

**Testing PTF**. To quantitatively test PTF performance for operational use in TEWS, we should define an unbiased set of events for which a tsunami warning issuance is required, regardless of whether a detectable tsunami was actually generated or not (the Gutenberg-Richter distribution of earthquake magnitudes implies that most of tsunami warnings will be issued close to this condition). To this end, we built a testing dataset (Fig. 5a) composed of all Mediterranean earthquakes that triggered alert messages from the CAT-INGV TSP, without any filter or selection. This includes all the twelve seismic events with initial magnitude estimate Mw ≥ 6.0 that occurred since CAT-INGV became operational in 2015. We added the 2003 Zemmouri-Boumerdes event, to enrich the set of events in the western Mediterranean, reaching a total of thirteen events (Fig. 5a).

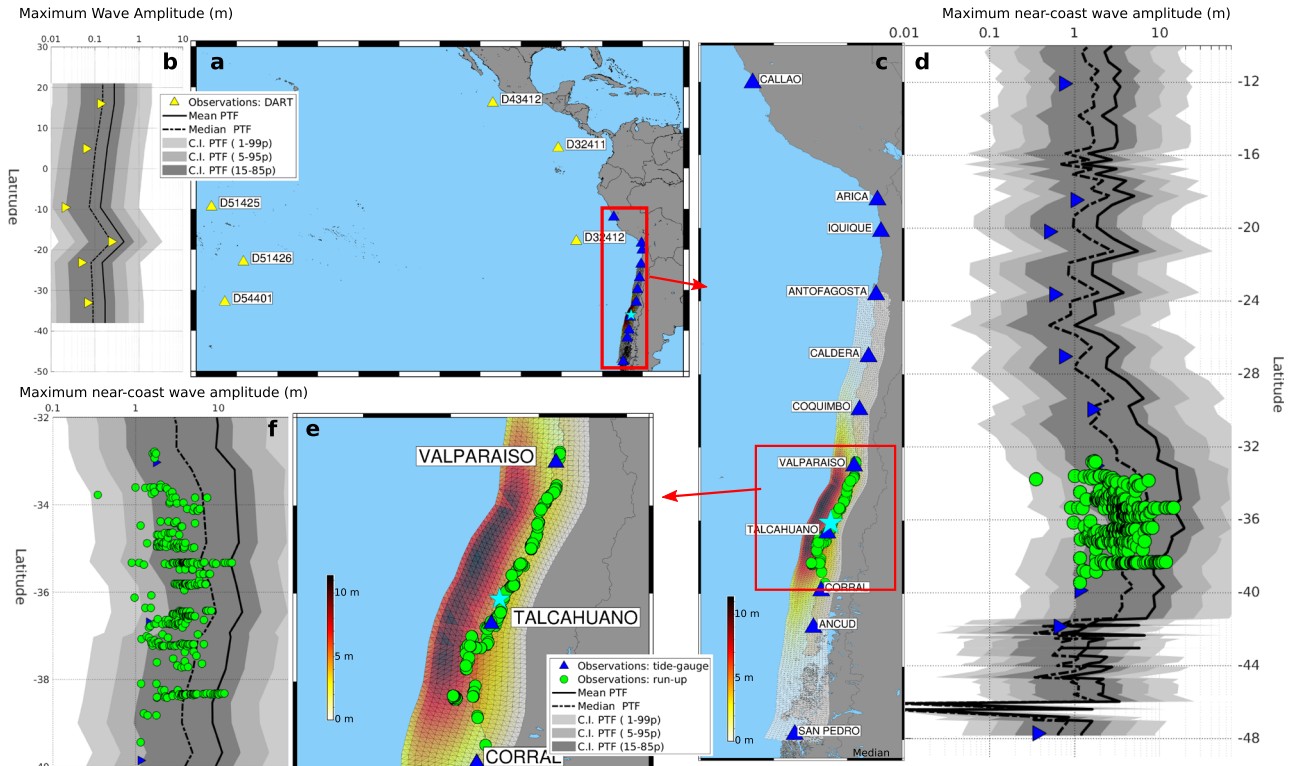

**Fig. 4 PTF for the 2010 M8.8 Maule tsunami. a** Epicentre and location of deep-sea (DART) observations (yellow triangles) and **b** corresponding comparison between deep-sea observations and PTF forecasts (black lines and grey areas). **c** Epicentre (star), average of the slip distributions used in the ensemble, and location of coastal observations (tide-gauges and run-up as blue triangles and green circles, respectively; run-up is halved to compare with wave amplitude, see Supplementary Note 6). **d** Graphical comparison between coastal observations and PTF forecasts (black lines and grey areas). **e**, **f** Same as **b**, **c** zoomed over the area with run-up measures.

Observations for the tests include rapid and revised moment tensor estimates, and tsunami observations from the available tide-gauges and from run-up surveys, when available (more details in Supplementary Note 6).

PTF accuracy is evaluated through formal hypothesis testing to assess the consistency between forecasts and available data and, if need be, to reject the PTF uncertainty model (see Methods). Both intermediate (source mechanism) and final (tsunami intensity) forecasts are tested. Results indicate that overall focal mechanism forecasts are accurate, such that the PTF source model is never rejected (results in Supplementary Table 7). Tsunami data and forecasts are compared simultaneously at all forecast points with observations, and spatial correlations are accounted for (see Methods). Although tsunami observations in many cases are limited, and sometimes with a poor signal-to-noise ratio due to the small event sizes, statistical tests confirm PTF accuracy also regarding tsunami forecasts, both for the events generating an observable tsunami (e.g. the October 30, 2020 Mw 7.0 Samos-Izmir event, Fig. 5b; results for all the six events of this type in Supplementary Fig. 4) and the ones for which a tsunami has not been observed (e.g. the 2017 Mw 6.5 Lesbos event, Fig. 5c; the results for all the seven events of this type in Supplementary Fig. 5). The tsunami generated by the Mw 7.0 Samos-Izmir earthquake (maximum run-up ~3.8 m[104]), as well as by the May 2, 2020 Mw 6.7 Ierapetra event, offered us a unique opportunity to perform a blind test for PTF, since the complete evaluation system was in place before the events occurred. The same test can be applied to the 2010 Maule tsunami, using both deep-sea and coastal observations as well as near-field and far-field observations; the results confirm the overall accuracy of PTF also for large magnitude event (Supplementary Fig. 4). On the other hand, for all the events that did not generate any measurable tsunami, PTF consistently forecasts an essentially negligible tsunami (<0.10 m) at all the observation points (Supplementary Fig. 5). While specific events may tend toward over/underestimation, altogether they pass the statistical test (accuracy level of 0.05). More details in testing results are discussed in Supplementary Note 7.

**PTF and alert levels.** Using the same testing dataset, we finally compare the PTF alert levels with those produced by the reference non-probabilistic methods (DM and BMS, Fig. 6). The comparison with data (Fig. 6a) is limited to the forecast points where observations are available. Comparisons are grouped in three categories as: correct-assignment (assigned = observed); false-alarm (assigned > observed); and missed-alarm (assigned < observed).

The three non-probabilistic methods give significantly different results (Fig. 6a and Supplementary Table 9). DM and ENV produces relatively few missed alarms (about 3%) but generates many false alarms (about 55%). This high percentage is in line with other conservative methods worldwide[4]. Conversely, BMS optimizes the correct assignments (about 86%), minimizing false alarms but increasing the missed alarms (11%). This reflects the fact that DM and ENV are worst case oriented to reduce missed alarms. On the contrary, the aim of BMS is to stay as close as possible to the actual event.

The alert levels computed from PTF shows a large variability, which depends on the selected percentile. High percentiles of PTF compare with conservative non-probabilistic methods (DM and ENV). The highest PTF percentiles (e.g. the 99th) are even more conservative than DM and ENV, further reducing missed alarms at the cost of further increasing false alarms. Decreasing the PTF

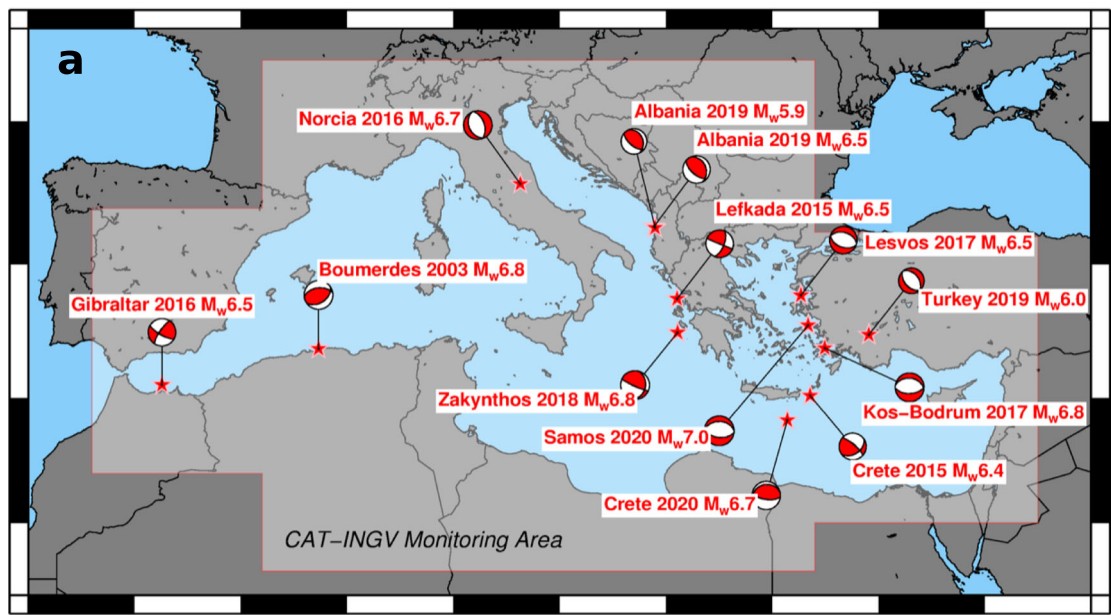

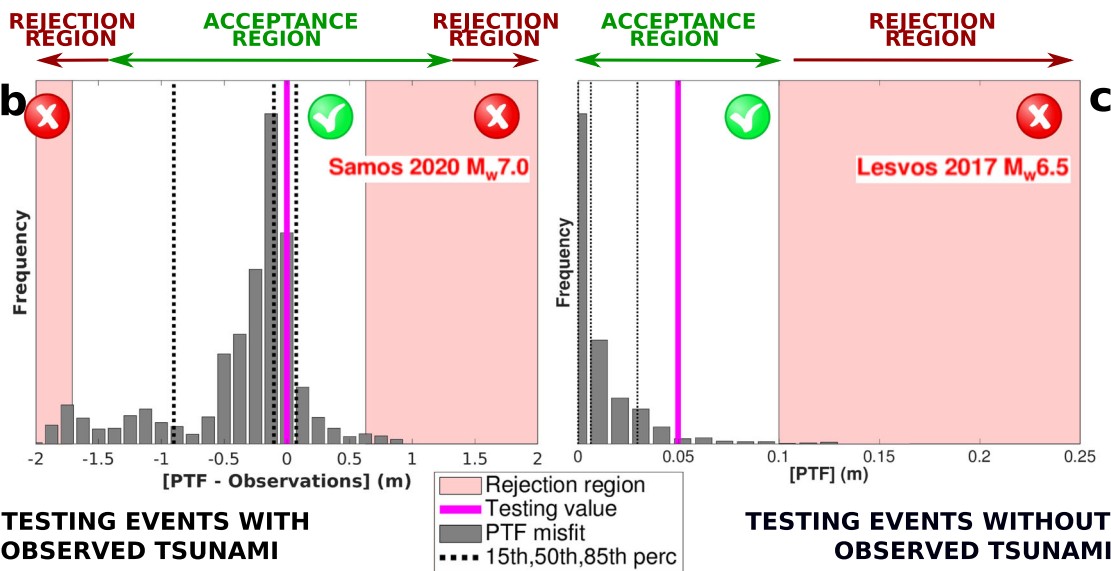

**Fig. 5 Testing PTF. a** Testing dataset and monitoring area of CAT-INGV; additional details are reported in Supplementary Table 2. **b** Example of test for events with observed tsunami: the case of 2020 Mw 7.0 Samos-Izmir earthquake. PTF misfit distribution (*[PTF-Observations]*) is evaluated as the difference between near-coast wave amplitudes sampled from the PTF source ensemble and observations and staked for all observation points (see Methods). Gray bars report the misfit distribution, along with its 15, 50, and 85 percentiles (dashed lines). The model is rejected if the testing value (null misfit, purple line) falls in the rejection area (light red area); otherwise, the test is passed. **c** Example of test for events without observed tsunami: the case of 2017 Mw 6.5 Lesbos earthquake. The PTF distribution (*[PTF]*), obtained sampling from the PTF source ensemble, is expected to encompass small values. The model is rejected if the testing value (the 95th percentile of *[PTF]*, purple line) falls in the rejection area (light red area: near-coast wave amplitude < 0.1 m); otherwise, the test is passed. To keep spatial correlations, in both *[PTF-Observations]* and *[PTF]* the uncertainty in propagation is averaged (see Methods). All the other case studies are reported in Supplementary Figs. 4 and 5.

percentile, the number of correct assignments progressively increases: most false alarms are suppressed, while missed alarms increase. The increase of correct assignments (green bars in Fig. 6a) and the decrease of false alarms (orange bars) are due to a reduction of the overall number of alerted (advisory or watch) forecast points (Fig. 6b), observed at all forecast points independently from the position and number of observations. PTF median and mean match with a best-match method like BMS. The BMS and the PTF median produces a similar percentage of correct alarms (85% vs 86%), while the PTF mean produce a slightly larger percentage of correct alarms, fewer missed alarms, but more false alarms and alerted forecast points (Fig. 6a, b).

Overall, PTF percentiles encompass and go beyond the range of behaviours and associated level of conservatism of DM, ENV, and BMS. The percentage of missed alarms can be strongly reduced with conservative choices (PTF high percentiles), that is from 14% to <1% passing from the median to the 99th percentile, at the cost of an increase in the percentage of false alarms, from <1 to 53%. Intermediate-high PTF percentiles (80th or 85th) are somehow between such extrema, progressively modulating the rates of missed/correct/false alarms.

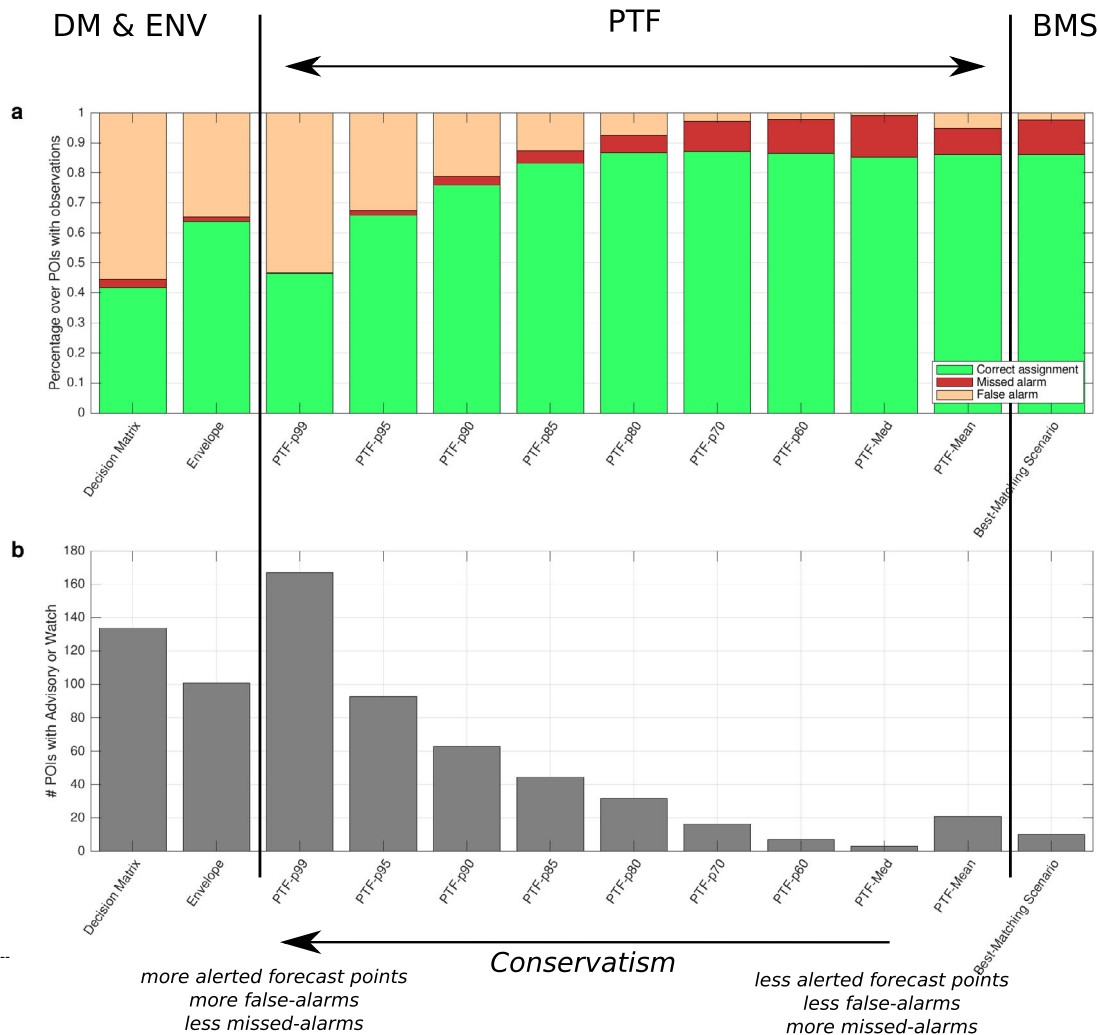

**Fig. 6 Alert levels from PTF and non-probabilistic methods.** We compare the assigned and observed alert levels based on DM, ENV, BMS, and PTF statistics for the 13 events in the testing dataset considered in this paper (Fig. 5a). **a** Average percentage of correct- (green), false- (yellow) and missed alarms (red) at forecast points with observations. **b** Average total number of forecast points with advisory and watch levels at all forecast points. Note that CAT-INGV DM is less conservative than the original NEAMTWS DM. The different PTF statistics allow covering the full range of conservative choices, encompassing the range defined by existing non-probabilistic methods. The selection of a specific PTF percentile can be explicitly linked to a pre-defined level of conservatism, quantifying the expected rate of false/missed alarms.

Hence, PTF allows better interpretation of the role of conservatism in present-day non-probabilistic methods, for an explicit and systematic selection of the desired level of conservatism.

Finally, we note that PTF helps overcoming the potential instabilities of DMs with events close to the defined magnitude thresholds. This instability can be well illustrated through the recent 2020 Samos-Izmir event. In the first minutes after the event, real-time magnitude estimations oscillated just around the DM threshold of Mw = 7 (with uncertainty bounds ~6.8–7.2, see Supplementary Table 2). As a consequence, small oscillations in the central magnitude could determine a significant change in the alert levels; for example, using the reference DM of CAT-INGV, all regional forecast points (<400 km) would pass from advisory to watch, and regional forecast points (>400 km) from information to advisory (see Supplementary Fig. 6), with a number of alerted forecast points passing from 297 (29 watch) to 1107 (297 watch). On the contrary, PTF solutions are not based on any threshold and they account for estimation uncertainty, then they are stable with respect to such oscillations.

## Discussion

We present an approach dealing with uncertainty in real-time tsunami forecasting and linking alert-level definition for tsunami early warning to such uncertainty, coined Probabilistic Tsunami Forecasting (PTF). Current practices do not quantify uncertainty in tsunami forecasting and define alert levels deterministically. To reduce missed alarms, they typically adopt safety factors that increase the number of false alarms. PTF addresses this issue through explicit uncertainty quantification, linking alert levels to the desired level of conservatism.

This approach has been implemented for near-field tsunami warning and tested against all available data in the Mediterranean, including two blind tests (the recent 2020 Mw 6.7 Ierapetra and Mw 7.0 Samos-Izmir earthquakes), as well as for the 2010 Mw 8.8 Maule earthquake and tsunami, one of the largest events ever recorded. The results show that PTF is statistically accurate in its forecasts, ranging from relatively small crustal earthquakes to large magnitude subduction zone events.

We have shown that uncertainty forecasts can be quantitatively and transparently transformed into alert levels, using real-time

conversion rules established in advance. Current practice bases this transformation on some generic rules defined in agreement with authorities, fusing the scientific and political aspects of defining alert levels[16,32,39–45]. As quantitative information about how certain is a forecast is not available, is not based on the effective real-time uncertainty on observations, and is not communicated, it is not possible to be sure regarding the degree of conservatism that is being applied. The formal quantification of uncertainty of PTF allows instead accounting for real-time uncertainty, covering explicitly the full range of possible choices, from conservative methods minimizing missed alarms to best-guess methods maximizing correct alarms. In this way, the desired average performance can be explicitly selected, allowing optimizing choices for each risk-reduction action. Choosing such rules requires competences outside tsunami science, as they depend on decision-makers needs, on acceptable risks, tolerated false/missed alarms rates, and other contextual factors. In any case, not only missed alarms but also false alarms may generate significant economical and societal consequences[4,105]. Considering that both missed and false alarms are due to uncertainty in the forecast, and both exist in current-practice methods, a transparent management of uncertainty is preferable[48–51], for example quantifying the potential socio-economical consequences of alternative choices, as evaluated from the expected long-run rate (over multiple events) of false/missed alarms[33,51–56,106–111]. Real-time uncertainty forecasts could also be exploited in the future by decision makers to define new strategies for risk management. Indeed, a range of different risk-mitigation actions (also beyond evacuation, such as the activation of mitigation procedures in industrial plants or automatic stops in lifelines) can lead to different choices for different targets and/or different actions with different tolerances to missed and false alarms[52–54]. This possibility is prevented in present-day common practice, but is made possible by an explicit quantification of uncertainty in real-time. This approach to tsunami warning would also complement the ongoing efforts towards uncertainty reduction through enhanced real-time tsunami monitoring capability (GNSS, DART, SMART cables[6,21]) and increase of real-time computational capability[23]. These elements have been already emphasized by the United Nation Decade of Ocean Science for Sustainable Development (2021–2030, https://www.oceandecade.org/).

More extensive testing against tsunami data worldwide will allow a thorough calibration of the uncertainty quantification framework, eventually introducing strategies to reduce uncertainty without losing accuracy. Here, by implementing a PTF applicable worldwide, we have set the scene for both hindcasting and blind tests of PTF performance against events of any magnitude, similarly to other testing experiments (http://cseptesting.org/). Further exploiting high-performance computing infrastructures, we can extend quantitative testing of tsunami forecasts and their underlying science worldwide to a larger set of tsunami events[46]. Moreover, several important specific issues are still only partially dealt with, like, for example, tsunami earthquakes[97] or more complex coastal dynamics. Testing and calibration must include these specific aspects to make PTF operational and fully suitable for science-informed decision making.

## Methods

**Probabilistic Tsunami Forecasting (PTF) evaluation.** The uncertainty existing at the time $t > t_E$ on the potential tsunami generated by the event E occurring at the time $t_E$ is summarized through a probability distribution conditional upon E. The corresponding survivor function $h_E(x, p, t) = P(X > x | E; p, t)$ describes a hazard curve for a given Tsunami Intensity Measure (TIM) $x$ in the target forecast point $p$, corresponding to the probability density function $dh_E(x, p, t)$. The function $h_E(x, p, t)$ can be estimated from the uncertain knowledge about E at time $t$ based

on an ensemble of tsunami simulations corresponding to tsunami sources compatible with the information about E available at the time $t$. The available information is constituted by the estimates of the source parameter values (e.g. earthquake location and magnitude) as derived from available seismic, geodetic and/or tsunami records (Fig. 1); different techniques may be applied to obtain this information, ranging from source inversion to data assimilation. The quantity and the quality of the information available may increase through time, eventually reducing uncertainty. Applying the total probability theorem, $h_E(x, p, t)$ reads:

$$h_E(x, p, t) = P(X > x | E; p, t) = \int_S P(X > x | s; p) g(s | E; t) ds$$
$$\approx \sum_i P(X > x | s_i; p) P(s_i | E; t) \qquad (1)$$

where $P(X > x | s; p)$ (propagation factor) is the probability that the earthquake scenario $s$ produces a tsunami exceeding the TIM value $x$ at the location $p$; $g(s | E; t)$ (source factor) is the probability that each scenario $s$ can be considered as a good approximation of E based on the uncertainty on the source parameters at the time $t$; the set $S$ includes all the possible scenarios $s$ in the area.

In the right-hand side of Eq. 1, we approximate the infinite set $S$ with a discrete set $\{s_i\}$, defining a finite ensemble of source scenarios resembling E. This discretization is possible if the databank $\{s_i\}$ is built to represent all the possible earthquakes in the area, reasonably covering all the natural variability. The probabilities $P(s_i | E; t)$ can be interpreted as weighting factors for each source within the ensemble. To speed up the evaluation of $h_E(x, p, t)$, the databank $\{s_i\}$ and corresponding propagation factors $\{P(X > x | s_i; p)\}$ can be prepared in advance. As time passes, $\{s_i\}$ and $\{P(X > x | s_i; p)\}$ can be refined accounting for the incoming information about the source and about the tsunami, eventually including data assimilation[7–10]. In addition, $\{s_i\}$ can be enhanced with new and possibly more accurate scenarios better resembling the observations (Fig. 1a). The forecast (Fig. 1c) and the alert level (Fig. 1d) can be updated accordingly.

The best candidate databank $\{s_i\}$ is the source model of a time-independent long-term PTHA (Probabilistic Tsunami Hazard Analysis[102]) for three main reasons. First, PTHA source models, by construction, should guarantee or approximate well enough the source completeness. Second, one of the ingredients of the PTHA is the databank of $\{P(X > x | s_i; p)\}$ used for tsunami propagation. Third, PTHA provides long-term source frequency and conditional probability for all scenarios, which makes it a suitable backup for not yet available real-time information. It may then provide all the elements depicted in Fig. 1b.

Given that $\{P(X > x | s_i; p)\}$ may be pre-calculated and used as a look-up table in real-time, the computational time is dominated by the quantification of $P(s_i | E; t)$, the retrieval of $\{P(X > x | s_i; p)\}$ from the databank and the aggregation procedure. Being the quantification of $P(s_i | E; t)$ and the aggregation computationally inexpensive, the main time-consuming step is the retrieval of the scenarios from the databank, which is a problem quite common in informatics that can be further optimized by code engineering with respect to present implementation. Time can be saved by reducing the number of scenarios (the ensemble size), for example, by discarding scenarios with negligible $P(s_i | E; t)$ through pre-defined cut-offs, whose practical implementation is discussed in the following section. Probabilities must be re-normalized accordingly to avoid biases. The larger the reduction, the larger the loss of accuracy in the tails of $dh_E(x, p, t)$. We stress that by coupling appropriate cut-off and specific code engineering, computational time can be probably reduced to a few seconds.

The presented formulation is in principle valid also for non-seismic tsunami sources. However, source parameters are more difficult to obtain in real-time for non-seismic sources and source variability is less constrained. For the same reason, also the creation of scenario databases is more challenging. As a result, present-day PTHA studies are primarily focused on earthquakes[92,102]. Since TEWSs are nowadays mostly devoted to seismic sources only, as a starting point we will also focus our attention to seismic sources.

**Quantification of PTF's source factor.** The source factor of Eq. 1 deals with the real-time uncertainty on the source of the event E, quantifying the proximity between the scenarios $\{s_i\}$ and E, based on information available at time $t > t_E$. In principle, $P(s_i | E; t)$ can be estimated using any type of real-time observations, including seismic and geodetic data, as well as tsunami records.

To deal with local tsunamis, delivery time for alert levels should be shorter than, say, 10 min (Fig. 1). For $t - t_E < 10'$, no direct measurements of the sea level anomaly associated with the ongoing tsunami are typically available, thus $P(s_i | E; t)$ should be estimated based on source parameters. Each scenario $s_i$ can be parameterized as $\sigma_i = \sigma_i(M_k, c_l, o_m)$, where $M_k$ is the magnitude, $c_l$ the geometrical centre of fault, and $o_m$ a vector with all of the other rupture parameters (e.g. strike, dip, rake, slip, other kinematic rupture parameters). Consequently, $P(s_i | E; t)$ can be factorized as a chain of conditional probabilities:

$$P(s_i(M_k, c_l, o_m) | E, t) = P(o_m | c_l, M_k; E, t) P(c_l | M_k; E, t) P(M_k | E, t) \qquad (2)$$

where $P(M_k | E, t)$ is the probability of the magnitude bin corresponding to $M_k$, $P(c_l | M_k; E, t)$ is the probability of the 3D volume bin (lon, lat, z) corresponding to $c_l$ and depending on $M_k$, and $P(o_m | c_l, M_k; E, t)$ describes the dependence of all the other unknown earthquake parameters on position and magnitude.

For $t - t_E < 10'$, not even a complete seismic source characterization is usually available. Real-time information typically includes only hypocentre and magnitude

estimation, while robust estimates of the other parameters $o_m$ are available only at later times, such as a moment tensor solution. Nevertheless, given an earthquake of a given magnitude at a given location, the possible values of all other seismic parameters $o_m$ are not all equally probable. They depend on the local long-term seismo-tectonic behaviour, and their likelihood can be retrieved from long-term PTHA, conditional to the magnitude and hypocentre real-time estimates. Therefore, $P(s_i(M_k, c_l, o_m)|E, t)$ turns out to be a mixture of real-time (RT) and long-term (LT) estimations:

$$P(s_i(M_k, c_l, o_m)|E, t - t_0 < 10') = P(o_m|c_l, M_k)^{LT} P(c_l|M_k; E, t)^{RT} P(M_k|E, t)^{RT}$$

(3)

where:

- the magnitude probability $P(M_k|E, t)^{RT}$ corresponding to the early automatic estimation uncertainty. We assume a normal distribution and integrate it over the magnitude bins corresponding to $M_k$. The normal distribution is set with the method of moments by setting the mean to the best-guess estimation and the standard deviations as the semi difference between 84 and 16th percentiles, as estimated from the adopted magnitude inversion method (see Supplementary Note 1 and Supplementary Table 2).
- the probability that $c_l$ is the centre of the causative fault can be evaluated as follows. The position of the nucleation $\zeta$ can be seen as the (vector) sum of the spatial position of the centre of the fault $c_l$ and the relative position of the nucleation within the fault, $\chi$, that is $\zeta = c_l + \chi$, and thus $c_l = \zeta - \chi$. Consequently, $P(c_l|M_k; E, t)^{RT}$ can be computed as the convolution between one distribution representing the uncertain position of $\zeta$ (from real-time information) and another distribution representing the uncertainty on the position of $\zeta$ within the fault. The latter depends on $M_k$: the larger the magnitude, the higher the probability that a relatively distant $c_l$ can be associated with $\zeta$. We assume a 3D normal distribution for both the uncertainty on $\zeta$ and $c_l - \zeta$. The former originates from the hypocentre estimation (see Supplementary Note 1 and Supplementary Table 2), while the latter is set centred in 0 with a covariance matrix with diagonal $\sigma_{xx}^2 = \sigma_{yy}^2 = (L/2)^2$, $\sigma_{zz}^2 = \frac{(W/2)^2}{2}$ (for an average dip of 45 degrees), and $\sigma_{xy} = \sigma_{xz} = \sigma_{yz} = 0$. In other words, this distribution, which describes the uncertainty in the position of nucleation within the fault, is obtained by multiplying three independent Gaussians with a horizontal standard deviation equal to $L/2$, and a vertical standard deviation $(W/2)\sin(\pi/4) = W/(2\sqrt{2})$. The fault dimensions $W$ and $L$ are evaluated using empirical scaling relations[76,77] for crustal and subduction interface earthquakes, respectively. Note that Murotani et al.[77] is selected to be more conservative since it provides larger expected areas than other empirical scaling laws available for subduction earthquakes. The convolution of these distributions (again a 3D normal distribution) is integrated over 3D volume bins corresponding to $c_l$.
- the probability $P(o_m|c_l, M_k)^{LT}$ of the other earthquake parameters $o_m$ is taken from long-term hazard estimations. Most earthquake parameters (e.g. faulting type or rupture details) mainly depend on the tectonic regime around the fault location (as evaluated from seismic catalogues) and on the characteristics of the source zone. For example, $c_l$ may lie on a subduction interface, which has a dominantly reverse slip mechanism, whose exact direction may, in turn, depend on the specific location over the slab interface; or $c_l$ may instead lie on the neighbouring outer-rise, with a higher probability for a normal mechanism. Other parameters (e.g. slip distribution) may depend on both position and magnitude. If this information is not available from previous long-term studies (at the global, regional or local scale), maximum ignorance can be modelled using uniform distributions until real-time information (e.g. focal mechanism and/or moment tensor estimations) become available.

To reduce the computational effort and save time, we implemented cut-off thresholds in the real-time estimations of Eq. 2, that is, the real-time quantification of the uncertainty in magnitude and hypocentral location. Scenarios with marginal probabilities smaller than the cut-off are neglected. For simplicity, the threshold in the hypocentral location has been implemented in 2D that is marginalizing in depth. We implemented thresholds corresponding to 1.5, 2, 2.5, and 3 standard deviations (Supplementary Table 4). On average, passing from 2 to 3 standard deviations increases the number of scenarios by one order of magnitude (from $10^3$–$10^4$ to $10^4$–$10^5$), significantly expanding the computational cost. Percentiles 5th-95th of the PTF remain stable for standard deviations $\geq 2$, and computational times are within 2' (the longest being $\sim 100''$), which can be considered an acceptable upper-limit for a non-engineered real-time serial application. Hence, the two standard deviations cut-off is taken as a reference for all examples and tests discussed.

In our prototype implementation for the Mediterranean Sea, the real-time earthquake parameter estimations are computed by the Early-Est software (see Supplementary Note 1). The long-term information is instead based on the NEAMTHM18 tsunami hazard model (http://www.tsumaps-neam.eu/[65–67]). NEAMTHM18 includes millions of scenarios completely covering the Mediterranean sea, considering two seismicity types for dealing selectively with epistemic uncertainty: predominant seismicity (PS), constrained to happen inside geometrically well-constrained subduction interfaces, and background seismicity

(BS), diffused everywhere within the crust. PS includes the Calabrian, Hellenic, and Cyprus Arcs, while BS covers all the Mediterranean with a regular grid, including the relatively less-constrained seismicity potentially occurring on unmapped offshore faults and the surroundings of subduction zones (e.g. in the outer-rise).

Outside the Mediterranean, the discretization strategy defined by NEAMTHM18 is still used, but PS sources are the subduction zones defined in SLAB2 model[112] and BS sources are modelled over a regular grid with size $\sim 0.2 \times 0.2$ degree corrected to define approximately equal size cells[113]. Real-time earthquake parameter estimates are taken from the literature (for the Maule case study[114]) and the forecast of focal mechanisms for crustal faults from[113].

Seismic fault parameters are considered less uncertain for PS than for BS. In the BS branch, all fault parameters are set as free parameters, except for few physical constraints: an upper bound is set for magnitude ($M_k \lesssim 8.1$), and depth is limited by the crustal thickness. Faults are planar with uniform slip and fault size determined from empirical scaling relations[76]. For the PS branch instead, only magnitude, position on the slab, and slip distribution are parametrized, as the geometry is specified by the 3D subduction interface, and the rake is forced to comply with the dominant one on the subduction segment. For $M_k \gtrsim 8.0$, heterogeneous slip is imparted using stochastic models suitable for 3D faults[96]. The magnitude is extended up to the magnitude the interfaces may host. Several alternative strategies are considered to model epistemic uncertainty associated with subduction earthquakes, such as different seismogenic depth ranges, scaling relations, rigidity properties, and stochastic shallow slip amplification[66,96].

The separation between PS and BS is implemented by splitting $P(s_i|E, t)$ of Eqs. 1 and 3 in two terms, that is:

$$P(s_i|E, t) = P(s_i|E, t, PS)P(PS|\zeta, M_k) + P(s_i|E, t, BS)(1 - P(PS|\zeta, M_k))$$ (4)

where $P(PS|\zeta, M_k)$ is the probability that the nucleation started at the point $\zeta$ on one of the three subduction interfaces considered in the Mediterranean Sea (the Calabrian, Hellenic and Cyprian Arcs). This is computed from the uncertainty on $\zeta$ from the real-time estimation, considering a seismogenic volume corresponding to each interface with a buffer of 10 km. For $M_k > 8.1$, earthquakes are assumed to belong to PS, so $P(PS|\zeta, M_k) = 1^{66}$. $P(s_i|E, t, PS)$ and $P(s_i|E, t, BS)$ are both evaluated as in Eq. 2, using the same magnitude distribution. Both the long-term factors (focal mechanism for BS, slip distributions for PS) are taken from NEAMTHM18 (mean of the epistemic uncertainty) for the Mediterranean case studies and, when alternative scenarios were present in NEAMTHM18, they were all included in the databank $\{s_i\}$, weighted by their epistemic credibility. For the Chilean subduction zone, slip distributions were produced, following the same strategy adopted in the NEAMTHM18[95,96].

**Quantification of PTF's propagation factor.** For each source $s_i$, the propagation factor in Eq. 1 is based on the results of one numerical tsunami simulation, often obtained as a linear combination of synthetic tsunamis produced by elementary sources.

The NEAMTHM18 propagation database[66,67] is based on dislocations in a homogeneous elastic medium. Seafloor deformations were processed with a low-pass wavenumber filter (modelled as 1/cosh(kH) following Kajiura approach[115], where k is the wavenumber and H is the average sea depth nearby the fault) to obtain the tsunami initial condition, reconstructed as a combination of Gaussian-shaped elementary sea-level elevations. Tsunami simulations are saved at the 50 m isobaths and, in this regime, nonlinear effects are negligible[116]. Gaussian sources were modelled with the benchmarked GPU-based nonlinear shallow water Tsunami-HySEA code (https://edanya.uma.es/hysea)[117], with eight hours of propagation on a regular grid including the whole Mediterranean Sea, using the 30 arc-sec bathymetric model SRTM30+ (http://topex.ucsd.edu/WWW_html/srtm30_plus.html). The results are obtained at the 50 meters isobath almost evenly spaced at $\sim 20$ km from each other along the coasts of the Mediterranean Sea (Supplementary Fig. 1 and Supplementary Dataset 1). The time step is computed using the usual CFL stability condition, that for a 2D, 2-step numerical scheme writes as $\Delta t = 1/4 \times CFL \times \min (\Delta x/\lambda_{max}, \Delta y/\lambda_{max})$, where $\lambda_{max}$ is the maximum eigenvalue of the matrix associated with the hyperbolic system to be approximated[118]. The CFL number retained is 0.95 (must be $\leq 1$), and the resulting time steps depend on the scenario simulated (mesh size and maximum propagation depth).

The NEAMTHM18 propagation database does not cover scenarios outside the NEAM region. For the scenarios within the Chilean subduction zone, we exploit modern high-performance computing infrastructures[69], performing all the individual simulations required to complete the source ensemble. For the Maule case study, the simulation environment has been set as the fault is modelled with a mesh of triangular elements preserving the variable strike and dip of the Nazca subduction zone as in the SLAB2 model. The numerical simulations have been performed using Tsunami-HySEA code with a bathymetric grid for the Pacific Ocean with a spatial resolution of 30 arc-sec[95].

Wave amplitudes in front of the coast are estimated from the offshore simulation results with the basic version of Green's law[119]: $x_{1m} = x_{50m}\sqrt[4]{50}$. Unlike in NEAMTHM18, the uncertainty related to tsunami generation, propagation, and inundation simplifications[102,103] is here modelled as a log-normal distribution, with median equal to the modelled tsunami near-coast wave amplitude, plus an unknown bias and a standard deviation that may be estimated by comparing modelled tsunamis against observations[102,103]. This uncertainty includes unmodeled source variability (realistic earthquakes are usually more variable than

the scenarios in $\{s_i\}$), local topo-bathymetric features, as well as to the variability of the tsunami along the coastline among different forecast points[92,102]. For simplicity, we neglect the bias, and set

$$P(X > x|s_i; p) = 1 - \Phi([\log(x) - \log(\xi(s_i, p))]/1) \qquad (5)$$

where $\Phi(x)$ is a standard cumulative normal distribution, and $\xi(s_i, p)$ is the value of the selected TIM (here, near-coast wave amplitude) evaluated at the forecast point $p$ due to the scenario $\sigma_i$. Bearing in mind the variability set by other authors[85,96], the variance is here set to 1.

**Testing source geometry and focal mechanisms forecasts**. To test the Zemmouri-Boumerdes forecast, we considered a total of 12 solutions as reported in Supplementary Table 3, five of them based on seismic moment tensor inversion, and the other seven obtained from geodetic finite-fault inversions (see Supplementary Note 4). The null hypothesis H0 is that the estimations can be considered a sample of our forecast model. To test H0, we randomly sampled groups of 12 focal mechanisms from the distribution $P(o_m|c_l, M_k; E, t)$ of Eq. 3, marginalized for all the parameters except the angles strike, dip, and rake. Then, we computed the log-likelihood of each group, assuming independence, and we compared the obtained distribution with the log-likelihood of the observations. Under H0, the rank of observations should be larger than a pre-defined conventional confidence level (one-tailed test).

The same test is performed for all the 12 events in the testing dataset of Fig. 5a, both collectively (all the events together) and individually (all events taken separately). We tested both preferred fault and double-couple planes, and H0 is evaluated at standard confidence levels. Even if the dependence of a single CMT solution is weak, we repeated the same tests restricting to the nine events that occurred after the production of the method[68] (in September 2016).

All the results are discussed in the Main Text and in Supplementary Note 7.

**Testing tsunami forecasts**. The test of PTF against tsunami observations is performed simultaneously at all the locations with available data. Considering that they are correlated to each other, we adopted a two-step strategy. First, we sampled scenarios from the source model $P(s_i|E, t = t^*)$ of Eq. 1 and considered the spatially correlated results. Second, we stacked the comparisons at all forecast points by taking the difference between the observations and the expected value (the mean) of $P(X > x|s_i; p)$ (the propagation factor) for the sampled scenarios, allowing us to compare all locations simultaneously. In this way, the uncertainty on the source is fully sampled, while the uncertainty on the propagation is averaged. Under the null hypothesis that PTF source and propagation factors are not significantly and systematically biased (in the sense of a large systematic over/underestimation), we expect that the distribution of the differences will contain the value 0. Where multiple observations associated with the same forecast point are available, the difference is computed against the maximum observation to guarantee a balanced and robust forecast evaluation. We verified that 0 is not in the tails of the distribution, but it is contained between the percentiles 2.5–97.5 for $\alpha = 0.05$. An example of this test is reported in Fig. 5b.

Whenever the available observations are all equal to 0, a bias would be found by the previous test, since PTF always forecasts >0. This occurs for seven events (see Supplementary Notes 6 and 7). In this case, the test described above is modified by verifying that 0.10 m (minimum threshold of Advisory AL) is unlikely at all the locations where observations are available. Adopting as above $\alpha = 0.05$, we tested that 0.10 m falls at percentiles larger than 95th, respectively (one-tailed test). An example of this test is reported in Fig. 5c.

## Data availability

All data generated or analysed during this study are included in this published article, in its supplementary information files, and in the referenced datasets (e.g., NEAMTHM18: http://www.tsumaps-neam.eu, IRIS Data Services and Data Management Center: https://ds.iris.edu/ds, Orpheus EIDA data services: https://www.orfeus-eu.org/data, VLIZ-IOC/UNESCO repository: http://www.ioc-sealevelmonitoring.org, Earthquake Source Model Database: http://equake-rc.info/SRCMOD).

## Code availability

The PTF Matlab code used for this paper is available on Github at https://github.com/INGV/matPTF.

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

## Acknowledgements

We thank D. Melini, and S. Cacciaguerra, INGV, and P. Lanucara, CINECA, for their support with the computational infrastructures; F. Hernandez for providing some sea-level data archived at the VLIZ-IOC/UNESCO repository; S. Belabbès, B. Delouis, M. Meghraoui, R. Santos for providing their inversion results for the 2003 Zemmouri-Boumerdes earthquake, and especially Y. Yagi and K. Obara who updated their inversion on purpose. This work benefited from the agreement between Istituto Nazionale di Geofisica e Vulcanologia and the Italian Presidenza del Consiglio dei Ministri, Dipartimento della Protezione Civile (DPC). This paper does not necessarily represent DPC official opinion and policies. The research leading to these results has been also partially funded by the European Union's Horizon 2020 research and innovation programme under the ChEESE project, grant agreement No. 823844, and benefited of the PRACE project TSU-CAST—TSUnami ForeCASTing (Proposal 2019215169, Call 20). Some figures were produced with GMT (Generic Mapping Tools, https://www.generic-mapping-tools.org/) and Inkscape, https://inkscape.org/.

## Author contributions

J.S. and S.L., with the support of M.V., F.R., R.T., P.P., F.B., and M.T. developed the Methodology. J.S. implemented the software, with the contribution of P.P., M.V., F.R., R.T., A.B., A.S., B.B., S.L., J.M., M.J.C., J.M.G.V., and C.S.L. A.B., F.L., S.G., F.M., A.P., and A.A. contributed to the conceptualization of the methodology. R.B., F.E.M., M.M.T., F.M., and A.A. developed fundamental Resources in input to the methodology. F.M., A.P., and A.A. supported the development of the methodology from the very early stages. J.S., S.L., and M.V. prepared the original draft, which has been reviewed by all the authors.

## Competing interests

The authors declare no competing interests.
