## [Peer Review File · Nature Communications]

REVIEWER COMMENTS

Reviewer #1 (Remarks to the Author):

31: Even though it is based on deterministic tsunami modeling, most TEWS take into account uncertainties by introducing safety factors or other similar measures. The downside is, of course, the risk of overestimation. Perhaps it would be relevant to comment on this. Also it should be noted that some TEWS use several scenarios before establishing the AL, hence the notion of deterministic becomes less clear.

47 Reference 26 appears to be incomplete

51 Not all TSPs link directly DM with seismic parameters. Japan, Australia and others do so with tsunami data, and in fact implement in the background a cruder version of the PTF, which, although not really probabilistic, at its core follows a similar logic. This notion is present and permeates throughout the text. For instance, it is also suggested in line 56.

59 Envelope or maximum credible methods already do this, although implicitly. The problem is not the uncertainty on itself, but how to take action from it.

55: The Tohoku example is misleading, as it was due to a poor assessment of the earthquake, coupled to a too fine model for the categorization of the Alert Levels. I would think that even a PTF as proposed would struggle with the first assessment provided during the Tohoku EQ.

155: That is a Mediterranean TEWS issue, hardly universal nowadays.

164 but even with all the data available, we can still have significant errors. How would you do it on the fly then?

175: But are simulations results expected to be Gaussian? I think they have a long tail, hence using the 16-84% can be an oversimplification and misleading.

176: Extremely good is a stretch, I think. It is true that statistically the results are reasonable, but you still miss actual maxima at several locations.

181. Precisely! How should a TPS deal with this then?

Fig 4 is hard to read. Again, it highlights several cases of underprediction.

Fig 5. Missed alarms: can we afford this?

1249: we can not budge the bullet here!

275: Perhaps a quantification of this is in order, but for other cases? How many false alarms are reported from systems through the world? How is this issue considered an issue.

278: is there a balance? What would you think is a proper balance?

281 Absolute novelty is relative. See prior comments

287 Again, the paradigm shift has already being taken elsewhere. It is true that not all the science might be in the can in the way it is being treated here, but it does so with a specific goal in mind and hence it has cut some corners.

Reviewer #2 (Remarks to the Author):

Quantifying uncertainties into tsunami forecasting is indeed a topic of interest within the tsunami warning community. A thorough and well tested approach is presented. I think the work is convincing for the Mediterranean region where they have the corresponding probabilistic tsunami hazard assessments. I think scientists from other tsunami warning centers would be very interested in reading this paper. Because of the scientific terminology and statistical nature of the paper, it would probably be much more difficult and of more limited use to the disaster risk management community which should have the final word on such an approach, especially in relation to the what percentile should be used.

Time is of the essence in tsunami warning, in addition to the precision of the forecast. I would have liked to see, at least in the main section of the article a timeline for the issuance of the different products, DM vs PTF.

One of the limitations of the study is the events available are relatively small and there are limited observations. I agree with the authors in testing the methodology.

A caveat within the discussion is that the tsunami forecasts are based on the tsunami associated directly with the earthquake, if a the earthquake generates a landslide then there could be deviations. This should be noted along with the deviations associated with local harbor conditions in the case of the Balears.

Some other specific comments:

Line 21 - missing the word "data" after the phrase "seismic and tsunami"

Line 29 - define the time window, "short" can mean different things to different readers

Line 77 missing "which" before occurred

Line 78 What if long term frequencies are not available as part of the Hazard Definition as part of the PTHA
Instead of using the term tide gauge, suggest "coastal sea level station"

Figure 2 - what are the broken line circles in inset c

Figure 4 is hard to read, event in digital and zooming in

Reviewer #3 (Remarks to the Author):

The study introduces a tsunami forecasting method that quantifies the uncertainties soon after the event occurrence. The test of 12 tsunami events in the Mediterranean region proves that their method (PTF) is able to make a tsunami early warning while controlling the missed alarms and false alarms effectively. Compared with the traditional method using Decision Matrices, the improvement is evident with regard to the missed-vs-false-alarm balance. Hence, the research is of high importance to disaster mitigation. Though the methodology part needs more clarification, and some figures need further improvement, I agree that this manuscript can be accepted after a minor revision. My comments are as follows:

L21: occurrence -> occurrence

L24: missed-vs false-alarm -> missed-vs-false-alarm

L57-58: The DM method connects the seismic source parameters to the ALs, but it is unfair to say the DM mixes scientific TF with non-scientific choices.

Actually, DM method also evaluates the tsunami potential and then transforms them to ALs, though there exist large uncertainties (<http://www.ingv.it/cat/en/tsunami-alert/alert-procedures/decision-matrix>). The new PTF method describes such uncertainties clearly, but it does not change the operation flow completely.

L88: For tsunamis in the Mediterranean Sea, the following two papers could be cited:

1) 2003 Zemmouri-Boumerdes earthquake: <https://doi.org/10.1186/s40562-019-0149-8>

2) 2020 Ierapetra earthquake: <https://doi.org/10.1029/2020JB020293>

These two papers focus on the tsunami data assimilation approach. It is a TF method that does not consider the source and avoids the uncertainties in the source characterization. It is also worthwhile mentioning the data assimilation method in the introduction part.

L96: The details of PTF implementation needs more clarification (also in the method part). The authors are supposed to describe clearly the number of the source ensemble, and why they choose it. In addition, more details about computing the earthquake hypocentre and magnitude probabilities should be provided (e.g., how fast it is during the operation).

L142: The four selected coastal locations are not very clear in Fig. 2. I suggest that the arrows could be even closer to the coastal locations.

L146: In Fig. 3c, the ALs of PTF mean does not match well with the observations, while the ALs of PTF 95th percentile are more consistent. Here more explanations are needed to discuss the performance of different scenarios and the conservatism. The results of PTF 16th-84th percentile could also be added for a better comparison.

L212: The reasons of the systematic underestimation at the Crotona tide-gauge should be explained. If such underestimation is caused by the tsunami resonance inside the harbor, could we use a finer grid size to simulate such resonance behavior?

L262: Gibraltat -> Gibraltar

L275: It is true that existing deterministic practices are more conservative and therefore are prone to false

alarms, but are they prone to missed alarms? The Fig. 5 shows that the DM methods have much less missed alarms than most PTF percentiles, which seems contradictory to this sentence.

L305: What kind of information are 'available information' at the time t ? Here more detailed explanations are expected.

L338: The formulation is in principle also valid for non-seismic tsunami sources. However, for landslide tsunamis or volcanic tsunamis, the source parameters are more difficult to obtain in real time, and there are less pre-calculated scenarios in the database. More discussions are expected for the TF of non-seismic sources.

L386-387: hypocenter -> hypocentre (I understand that both forms are acceptable, but the authors use 'hypocentre' in other parts of the manuscript.)

L386: What are the physical meanings of σ_{xx} , σ_{yy} , etc.? In Equations 1-3, the parameter σ has already been used to represent the earthquake scenario. Here it should not be used repeatedly.

L447: I could not find Eq. S2 in the Supplementary Information. Is it missing?

L458-466: More information about the tsunami simulation are expected. For example, the authors should write the cut-off wavelength of the filter, and the time step of numerical simulation. In addition, in this study, the tsunami numerical simulation is obtained as a linear combination of synthetic tsunamis produced by elementary sources. Therefore the nonlinearly effects cannot be considered. This limitation should be discussed.

L477: A closing bracket is missing in Eq. 4.

L520-523 & Fig. 4: For seven events without evidence of a tsunami, the authors tested that 0.1 m is unlikely at all the locations. However, for those tsunami events, they verified that 0 is not in the tails of the distribution. These two criteria appear in Fig. 4 at the same time, which makes the very difficult to understand. I suggest that the authors should use different colors or symbols to distinguish the two different tests, and to divide the test cases into two categories: with or without tsunamis. Last but not least, the font size in Fig. 4 could be larger.

Reviewer #1

POINT R1.1

31 Even though it is based on deterministic tsunami modeling, most TEWS take into account uncertainties by introducing safety factors or other similar measures. The downside is, of course, the risk of overestimation. Perhaps it would be relevant to comment on this.

Also it should be noted that some TEWS use several scenarios before establishing the AL, hence the notion of deterministic becomes less clear.

Original Manuscript - Lines 29-34: "Tsunamis pose a hazard that may strike a coastal population close to the earthquake within a short amount of time. An issue in today's Tsunami Early Warning Systems (TEWS, Table S1 summarizes all symbols and acronyms) is that Tsunami Forecast (TF) is deterministic, in contrast with the fact that a tsunami impact prediction immediately after the event is subject to large uncertainty. For this reason, deterministic forecasts aim at being conservative, which may lead to increased false alarms undermining confidence in TEWS."

We agree with both suggestions.

Regarding the first one, we realized that we were somehow misleading in the text, giving the impression that TEWSs ignore uncertainty and that a one-to-one correspondence exists between "deterministic" and "conservative increasing false alarms".

Actually, often (but not always) TEWS accommodate uncertainty heuristically introducing conservative choices (for example through a "safety factor"), and this increases the rate of false alarms. We specified this in the abstract and the introduction (lines 17-18 and 40-44). Conversely, deterministic tsunami modelling may lead by itself to both under- and over-estimation, depending on how the simulations are initialized (uncertainty on the source), and on other unknowns/simplifications of the inundation process (uncertainty in the propagation/inundation). This clearly emerges by the new reference non-probabilistic method introduced in this revision as discussed later (**Answer R1.3**).

Regarding the second suggestion, in the original version we used the word "deterministic" as a synonym of "non-probabilistic", meaning that it leads to a single number (one tsunami intensity) of the forecast, without any uncertainty measure communicated along (as now specified in lines 38-41). We did not mean that the forecast is necessarily obtained from one deterministic scenario (as specified in lines 61-62), which is nevertheless different from being fully probabilistic, as discussed later (**Answer R1.3**). To avoid this potential confusion, we systematically adopted "non-probabilistic" in place of "deterministic" when referring to present-day common-practice (e.g., lines 59, 76, 192-193, etc.).

POINT R1.2

47 Reference 26 appears to be incomplete

Fixed, thanks (now reference 31).

POINT R1.3

51 Not all TSPs link directly DM with seismic parameters. Japan, Australia and others do so with tsunami data, and in fact implement in the background a cruder version of the PTF, which, although not really probabilistic, at its core follows a similar logic. This notion is present and permeates throughout the text. For instance, it is also suggested in line 56.

Original Manuscript - Lines 50-58: "The Tsunami Service Providers (TSPs) worldwide (<http://www.ioc-tsunami.org/>) use Decision Matrices (DMs, look-up tables linking earthquake parameters with alert levels) or best-guess scenarios with pre-calculated or on-the-fly tsunami simulations. However, without uncertainty estimation, TF and its underlying assumptions cannot be quantitatively tested and calibrated, even leading to unpredictable outcomes. For example, during the great Tohoku 2011 tsunami, despite a supposedly precautionary approach, the tsunami was initially largely underestimated. Besides, methods like DM directly connect the seismic source parameters to the Alert Levels (ALs) linked to practical actions, thus mixing scientific TF with non-scientific choices."

The reviewer is right: we have not said explicitly that tsunami data may be used to form the forecast, although we have not even said that all TSPs use DMs based on seismic parameters (we wrote that also tsunami scenarios based on best-guesses are used). We modified the text for the sake of clarity (see lines 61-62).

Regarding the use of multiple scenarios implemented (in background) by some TSPs as a "cruder version of the PTF", we do not agree. The core elements of PTF are: the explicit real-time quantification of (time-dependent) uncertainty through a probabilistic method, and the transparent inclusion of uncertainty into the decision-making process of defining alert levels.

The Japanese and Australian TSPs, as well as, to our knowledge, all the other TSPs, do not include these elements to our best knowledge.

The Japanese and Australian systems differ from each other, as the first one is designed to deal with near-field tsunamis, while the second one with tsunamis generated by distant sources. Nevertheless, both systems use non-probabilistic approaches: starting from earthquake and tsunami data, eventually using one or more scenarios, they define a single-outcome forecast, eventually accompanied by a "safety factor". More details on their methods and procedures can be found in the **APPENDIX** to this document.

Differently from probabilistic forecasts, this type of methods do not quantify the uncertainty, nor do they communicate it formally into the definition of alert levels, in order to allow structured decision-making under uncertainty. Therefore, considering one or a few scenarios without probabilistic methods (as in Japan and Australia and several other tsunami warning centres), as well as using an empirical decision matrix (as in all TWSs in Europe), can be

hardly described as “a cruder version of the PTF”. They simply deal with the same problem, that is, the existence of uncertainty. We clarified these concepts in the revised manuscript, especially at the end of the Introduction (lines 63-65).

To further clarify this concept, as reference for comparing with PTF, we added a second non-probabilistic method (the Best-Matching Scenario, BMS) approach, in which the scenario that best fits the available information is selected and alert levels are set from a single-valued outcome (lines 196-199). Both DM and BGS are now used in parallel throughout the paper as non-probabilistic PTF counterparts (see for example the Figures 2c and 6 and related discussions in lines 200-217 and 313-354).

POINT R1.4

59 Envelope or maximum credible methods already do this, although implicitly. The problem is not the uncertainty on itself, but how to take action from it.

Original Manuscript - Lines 59-61: “To remedy, we propose here a new TF procedure, coined Probabilistic Tsunami Forecasting (PTF) that, in contrast to deterministic methods, quantifies probability distributions at multiple forecast points describing the uncertainty of tsunami intensity (e.g., run-up or wave height).”

As said above, we agree that these methods accommodate uncertainty implicitly (**R1.1** and **R1.3**), and we modified the text accordingly.

There is no doubt that how to take decisions accounting for uncertainty from an uncertain forecast is of utmost importance. So far, bypassing the computational and methodological problem of evaluating the probability of the potential scenarios in real-time with a partial knowledge of the source and the uncertainty in propagation, the workaround has been to ask scientists to “stay safe” (eventually blaming them if forecasts result “wrong”, on either side), introducing some level of conservatism in tsunami forecasts, which means biasing the forecast toward higher tsunami intensities [e.g., Melgar et al., 2016 - ref.16; JMA 2013 - ref.31].

Conversely, we propose a quantitative solution to the problem, based on the full quantification of uncertainty, and we also show how it is possible to take action from it with numerous examples. From the PTF, the level of cautiousness can be defined transparently by decision-makers by selecting a percentile to refer to and produce alert levels (see lines 209-214). We also deeply discuss and test the results of this approach and the differences with present-day solutions (see Fig. 2c and relative discussion in lines 200-217 for Zemmouri-Boumerdes, and Figure 6 and related discussions in lines 313-354 for the testing dataset of Fig. 5a, now composed by 13 events).

POINT R1.5

55: The Tohoku example is misleading, as it was due to a poor assessment of the earthquake, coupled to a too fine model for the categorization of the Alert Levels. I would

think that even a PTF as proposed would struggle with the first assessment provided during the Tohoku EQ.

Original Manuscript - Lines 54-56: "For example, during the great Tohoku 2011 tsunami, despite a supposedly precautionary approach, the tsunami was initially largely underestimated."

We understand the reviewer's concern about the Tohoku example, and we agree that, as originally proposed, it may have been misleading.

Of course, if the input estimation of uncertainty results wrong, the PTF is not a remedy. Nevertheless, as recognized by JMA, one of the main issues during Tohoku was first of all a lack of consideration of the uncertainty, and especially biases eventually affecting the seismic parameters. In this respect, we think that this example is relevant.

More specifically, during the Tohoku event, uncertainty on magnitude was not considered at all, particularly the likelihood of underestimating large magnitudes because of saturation. After the event, JMA revised their procedures (JMA 2013 - ref.31) by i) acknowledging that it is pointless to describe the tsunami height very precisely ("*a too fine model for the categorization*" in JMA 2013) in presence of large "*estimation errors*" that are not communicated along with the estimate itself; ii) introducing the use of a predefined maximum magnitude if "*the uncertainty of the magnitude is considered to be large*", in which case "*qualitative terms such as Huge and High are used rather than quantitative expressions*" (JMA 2013).

Thus, one of the main issues during Tohoku was exactly a lack of consideration of uncertainty. Now, we updated the manuscript, to synthetically explain these points (see lines 68-75).

POINT R1.6

155: That is a Mediterranean TEWS issue, hardly universal nowadays.

Original Manuscript - Lines 153-157: "Thus, DMs do not take into account that both source orientation and bathymetry control the tsunami propagation pattern and tsunami features, as clearly seen for this event^{40,41}. Consequently, despite the intended conservatism, DM can produce both AL over- and under-estimates. In contrast, ALs based on PTF embed the tsunami propagation footprint through numerical simulations."

We agree. DM is used mainly in NEAMTWS (North East Atlantic, Mediterranean and connected seas Tsunami Warning System), even if it is used to solve a problem that exists not only in the Mediterranean (that is, near-field warning for tsunamis generated by crustal sources), as demonstrated several recent events like 2018 M7.5 Palu-Indonesia, 2020 M7.7 in Cayman Is., or the 2016 M7.8 Kaikoura-New Zealand. Notably, an ad-hoc working group has been formed within the "TOWS WG Inter-ICG Task Team on Tsunami Watch Operations", coordinated by IOC-UNESCO to discuss worldwide best-practices about these crustal non-megathrust events and other non-seismic sources, see for example

http://legacy.ioc-unesco.org/components/com_oa/oa.php?task=download&id=41864&version=1.0&lang=1&format=1.

For the sake of generality, we have:

1. Discussed the PTF implementation globally, subject to simulation availability (lines 153-158 and Methods), as demonstrated by the introduction of a case study in the Pacific (the 2010 Mw 8.8 Maule earthquake and tsunami, in Chile, lines 252-279), demonstrating that the discussed method is global, subject to simulation availability;
2. Introduced the Best-Matching Scenario (BMS) as second reference for non-probabilistic tsunami forecast, as already discussed in **Answers R1.3 and R1.4**. As already said, we have now two reference non-probabilistic methods to compare the PTF: the DM as representative for empirical conservative methods, and BGS as representative for single-outcome simulation-based forecasts. In our opinion, BGS and DM collectively well represent the range of non-probabilistic solutions adopted by TWSs worldwide, as discussed in the revised manuscript (lines 192-199). Now, both reference methods are kept as reference throughout the manuscript.

The implementation is discussed and tested over a wide range of magnitudes and earthquake types, including the 2003 Mw 6.8 Zemmouri-Boumerdes earthquake and tsunami in the western Mediterranean, the great 2010 Mw 8.8 Maule, Chile, earthquake and tsunami in the Pacific, as well as all Mediterranean earthquakes that triggered alert messages at the Italian tsunami warning since operative (initial Mw ≥ 6.0 since 2015). This important point is reported both in the abstract (lines 25-30) and in the final discussion (lines 397-409).

POINT R1.7

164 but even with all the data available, we can still have significant errors. How would you do it on the fly then?

Original Manuscript - Lines 163-165: "For the Zemmouri-Boumerdes tsunami, several moment tensors and finite-fault model estimates are available in the literature (Table S3). The tide-gauge observations for this tsunami are sparse with mostly coarse sampling rates 40-42 ."

Quantifying uncertainty, PTF allows producing estimations considering the uncertainty at the time of the estimate (lines 104-106). This uncertainty can be reduced as more data become available, but it may remain significant even after a relatively long time. We have proposed an algorithm to deal with uncertainty on the fly and to formulate a forecast with uncertainty at any time, matching operational constraints for near-source tsunami warning (see discussion in lines 126-132).

Of course, the residual uncertainty after years of research (expressed by the different models developed for the 2003 Zemmouri-Boumerdes earthquake, reported in Table S3) will be smaller than the one estimated a few minutes after the event from real-time data and long-term hazard studies, as demonstrated first comparing tsunami forecasts (see red lines and grey bounds in Fig. 3 – reported also below in **Answer R1.9-10** – and discussed in lines 229-

238) and then statistically testing both fault mechanism and tsunami intensity forecasts (generally discussed in lines 292-312, and specifically discussed in Supplementary Text S7).

POINT R1.8

175: But are simulation results expected to be Gaussian? I think they have a long tail, hence using the 16-84% can be an oversimplification and misleading.

We thank the reviewer for this comment, since the text was probably not clear enough on this point. PTF's distributions are likely not Gaussians, and exactly for this reason we do not define standard deviations but we compare mean and median (that are different indeed) and use an interval of percentiles to define a confidence interval to be reported in the plot. This is now specified in lines 234-235.

The 16-84% interval was selected because many people do use Gaussian, and this interval is equivalent to the confidence interval defined in Gaussians by 1 standard deviation. Any other interval can be used, and in the revised manuscript we report instead the equivalent interval defined by the 15-85th percentiles, to avoid confusion (in these particular plots, we also reported other confidence intervals, namely 1-99 and 5-95, to answer the following points **R1.9-10**, and **R3.9**).

We updated the manuscript (see, for example lines 232-233 and 263) and all figures accordingly (for example, Figs. 3, 4, S4, and S5).

POINTS R1.9-10

176: Extremely good is a stretch, I think. It is true that statistically the results are reasonable, but you still miss actual maxima at several locations.

181. Precisely! How should a TPS deal with this then?

Original Manuscript - Lines 175-181: "Simulations from finite-faults fall within PTF's 16th-84th percentile confidence interval and the means are highly clustered. This extremely good comparison with tsunamis from finite-faults indicates that, while our PTF implementation simplifies the source representation, it is capable of reproducing the tsunami uncertainty satisfactorily³⁹. However, this tsunami challenged numerical modellers due to basin and harbour-related amplifications that likely occurred in several areas (e.g. in the Balearic Islands harbours,⁴¹); these features are not locally predicted by finite-fault models and are in the upper tail of the PTF."

We agree that words like "extremely" should be avoided, in favour of more quantitative formulations.

It is also important to deepen into the apparent misfits in the comparison between PTF and tsunami observations, in correspondence of some maxima in the data. We reorganized this discussion and improved the relative figure to make it more clear (lines 218-251 and Fig. 3)

New Figure 3

These maxima cannot be fully captured neither from the PTF's central values (black lines), nor using the most recent fault models (red lines). On the other hand, they are within the PTF tails (grey bands), even if outside 15-85th percentiles. As discussed in lines 239-241, this confirms that these maxima are due to non-modelled local amplification effects, as reported in the scientific literature already, which cannot be predicted by deterministic tsunami propagation models without the adoption of local higher resolution harbour models.

PTF already deals with this problem to some extent. Indeed, it embeds these maxima in the tails of the distributions (Fig. 3b,d). This plays a fundamental role in reducing missed-alarms (Fig. 2c and 6). So, despite using a relatively crude log-normal distribution superimposed to deterministic simulations to quantify propagation uncertainty, this is an indication that an appropriate uncertainty treatment is the right way to address the forecasting of hard-to-predict events. In the figure, the adoption of more advance techniques (like the inclusion of pre-calculated inundation models, Gusman et al. 2014 ref. 85, or the use of specific harbour amplification model, Gailler et al. 2018 ref. 84), as well as more advanced uncertainty models, will probably improve this situation, but this is not yet feasible for real-time near-field applications. This is now specifically discussed in lines 241-251.

POINT R1.11

Fig 4 is hard to read. Again, it highlights several cases of underprediction.

Thanks for this comment. We think that the impression that there are “cases of underprediction” may be due to a misunderstanding, because of the poor quality of the

original version of Figure 4 that misguided the reviewer. From this and other comments below (e.g., points **R2.1** and **R3.19**), we realized the statistical testing part of the original manuscript was probably too heavy, and thus we completely reorganized this part.

Probably the reviewer refers to those events for which no significant tsunami has been observed (now in Figure S5). We have two different statistical tests for the cases in which a tsunami has been observed, and the ones in which it has not been observed. Both these cases are now better explained in Figure 5b,c and in the text (lines 343-348).

Extract of the new figure 4

For the cases in which the tsunami is not observed, the statistical test is reversed and we expect that the PTF forecast (grey bars) mainly in the left tail relative to small tsunami intensities (< 10 cm), the smaller the better. Therefore, they are not underpredictions, but they actually are good predictions. In the original version, the graphics was misleading.

POINTS R1.12-15

(R1.12) Fig 5. Missed alarms: can we afford this?

(R1.13) 249: we can not budge the bullet here!

(R1.14) 275: Perhaps a quantification of this is in order, but for other cases? How many false alarms are reported from systems through the world? How is this issue considered an issue

(R1.15) 278: is there a balance? What would you think is a proper balance?

Original Manuscript - Lines 248-251: “[...] the use of higher PTF percentiles (90 th -95 th) can reduce missed alarms, but at the cost of increasing the number of false alarms. Choosing the best compromise is not a task of tsunami science; it depends on decision-

makers' needs, on acceptable risks, tolerated false/missed alarms rates, and other contextual factors. ”

Original Manuscript - Lines 274-280: “Existing deterministic practices are prone to numerous false or missed alarms. PTF addresses this issue through uncertainty quantification. We have shown that PTF results can be quantitatively tested and transformed in a transparent manner into ALs. We also demonstrated that PTF allows for rational balance of missed vs false alarms. Preliminary comparison using events in the Mediterranean Sea showed that PTF produces fewer false alarms than present-day early warning methods, also very close to the tsunami source.”

All these points regard the management of false- and missed-alarms, and of their balance, which is a very important issue to carefully discuss. We revised and deepened this discussion in the main text, as described below.

(R1.12) Missed-alarms may always happen also with present-day conservative non-probabilistic methods (red bar in the left column of new Figure 6a). Actually, as we explicitly report in the revised manuscript *“the use of higher PTF percentiles (90th-95th) reduces missed alarms (from 4% to 2%)”* when we compare PTF to a conservative DM (see lines 335-336), while *“the percentage of missed-alarms is more than halved considering PTF high percentiles (from 11% to 1.6, 3% and 4% for 95th, 90th and 85th percentiles, respectively)”* when we compare PTF to BGS (see lines 342-344). In the revised manuscript, we also note that *“Intermediate-high PTF’s percentiles (80th or 85th) seem to optimize the performance with respect to both BMS and DM, as they perform as good as BMS in terms of correct assignments (about 85%) but they have a rate of missed-alarms similar to a conservative DM (about 5%).”* (lines 345-347).

(R1.14) False-alarms are clearly also a problem. Following Bernard and Titov (2015, Ref. 4), inaccuracies of the forecasts led to a false alarm rate of 75%, defined as tsunami evacuation accompanied by a non-flooding tsunami. This large number of false alarms is in line with our findings (55% of false alarms for the DM, see Fig. 6), as now discussed in the revised manuscript (see lines 318-320). This can be surely largely attenuated, particularly in the far-field, by tsunami data assimilation, but it is still an issue, and it will remain so in the first minutes, even following strong technological development. This is very well illustrated by Figure 1 of Angove et al. (2019 - Ref. 6), which we report below for the reader’s convenience. In the first minutes, the role of uncertainty quantification is even more important, but it always goes along with good data and robust modelling.

From Angove et al. (2019 - Ref.6). *Tsunami source uncertainty dominates tsunami warning especially in the first minutes in the near-field. Even with the foreseen developments, this uncertainty cannot be eliminated.*

As far as false alarms are concerned, their cost may be very high and must surely be considered an important issue. For example, citing Bernard and Titov (2015 - Ref.4) “*The 1986 false alarm frustrated business owners, enraged the public and labelled the NOAA warning centre as inept. The State of Hawaii estimated this false alarm cost the state about \$41 million (US\$) and led to the loss of credibility for tsunami warning products, which were disconnected from the flooding hazard.*”, pushing the development of DART boys to reduce false alarms at least far from seismic sources. More recently, Angove et al. (2019 – Ref.6), say: “*Since it presently takes hours to precisely determine an earthquake’s forcing mechanism, and observations are limited, emergency managers, particularly in the near field, face large impact uncertainties when deciding how to protect their communities. They must develop and execute preplanned protocols based on broad—sometimes false—assumptions about a tsunami’s potential height and inundation. This can lead to over or under warning, the latter being particularly dangerous.*”.

The fact that, not only missed-alarms, but also false-alarms may have significant economical and societal consequences that have to be considered when defining risk reduction actions. This is now noted also in the revised manuscript (see lines 356-363).

(R1.13 & R1.15) Contrary to present-day non-probabilistic methods, PTF allows for a transparent management of missed- and false-alarms. The tradeoff between false- and missed-alarms can be rationally managed (Fig. 6a). The choice of a compromise between false- and missed-alarms is implicit in all methods, including in existing non-probabilistic methods (one “bar” in fig. 6, as for DM or BMS). At present, the level of conservatism in these methods is not controlled by decision-makers, but a transparent and aware decision-making process is instead always preferable (e.g. Budniz et al., 1997 – Ref.47; Frewer et al., 2003 – Ref.49; Fakhruddin et al., 2020 – Ref.48). A rational way of defining an optimal balance may be to compare the societal and economical consequence of both, through cost-benefit analyses (e.g., Fischhoff, 2015 – Ref.51; Woo 2011 – Ref.50; Rogers and Tsurkinov, 2010 – Ref.52). Seeking this balance stays in the political ground, as only decision-makers have the competences and the political mandate to make choices with potential economical and societal consequences. This is true for many different fields and applications. For example, decision-makers are deciding when it is better to impose a lock-down to avoid

COVID expansion by also taking into account the consequences on the national economy, while virologists should only advise them by forecasting the potential epidemiological aspects of either decision. In this process, the communication of uncertainty is fundamental to allow transparent decision making (Fakhruddin et al. 2020 – Ref.48). This is now explicitly discussed in lines 355-364, commenting the results of Fig. 6.

Overall, this process is similar to the one already in place for managing seismic risk, where the level of protection is usually even enforced by law, to guide engineers that have to implement it by following the building codes (e.g., Solomos et al. 2008 - Ref.55). Is it better to design for the 2% probability in 50 year or for the 10% in 50 years seismic shaking, that is for events with 2475 and 475 years average return period of exceedance respectively? The higher the chosen reference shaking, the higher the construction cost. As scientists, we do not have the competence nor the mandate to take these decisions, and a clear separation between hazard quantification and risk management is becoming a best practice (e.g., Jordan et al., 2014 - ref.52; Field et al., 2016 – ref.53). Notably, this recently extended also for tsunami risk, where tsunami evacuation zones for tsunami warning are designed based on a predefined probability threshold from long-term tsunami hazard analysis (Ref.56 to 60). This separation resolves one of the important drawbacks of present-day non-probabilistic tsunami warning practices, as discussed in the Introduction (lines 104-116).

These points are very important and would represent a major step forward for tsunami warning equivalent to the passage from worst-case scenario to probabilistic hazard quantifications, as stressed in the final discussion (lines 384-395).

POINT R1.16

281 Absolute novelty is relative. See prior comments

Original Manuscript - Lines 281-282: “This novel approach needs more extensive testing and thorough calibration of the uncertainty quantification framework against tsunami data worldwide.

We here summarize what we already discussed in all previous points in detail, in the form of “list of the main novelties”:

- Present-day warning system approaches accommodate uncertainty either being conservative or heuristically by comparing different alternative scenarios, defining a single-outcome forecast without uncertainty quantification. We defined the PTF which represents a novelty in comparison because it defines a complete workflow for tsunami warning in which uncertainty on all potential outcomes is quantified and used to define alert levels (e.g. **R1.3** and **R1.4**). To mark the difference, PTF is now compared with two reference methods that better represent the range of present-day non-probabilistic common-practice (**R1.6**).
- PTF separates the scientific task of tsunami forecasting from the decision-making task of defining alert levels (and consequent risk reduction actions). This separation, which is possible only by adopting a probabilistic formulation (**R1.1**), allows both to quantitatively compare the forecasts with data (e.g., **R1.9-R1.10**, **R1.11**) and to optimally define warnings based on decision-making needs (**R1.12-1.15**). This is

fundamental in decision-making for risk management, and it marks a paradigm shift in tsunami warning operations (this point is further discussed in the following **R1.17**).

- To show its performance, PTF results are challenged through statistical testing, considering a wide range of magnitudes and earthquake types. To this end, we implemented PTF for many events from both the Mediterranean and the Pacific (**R1.6**), including all recent events in the Mediterranean (13 events, including 2 blind tests: the M6.7 Ierapetra and M7.0 Samos-Izmir earthquakes, occurred during this revision) and the great 2010 M8.8 Maule earthquake and tsunami in the Pacific, one of the largest events in this century. The defined tests can be used to challenge PTF against observations for any other past or future event (e.g. **R1.7**, **R1.9-10**, **R1.11**).
- To show PTF usability in warning operation, we have shown a practical PTF implementation with all necessary calculations feasible in real-time for the Mediterranean Sea, matching NEAMTWS operational standards (e.g., **R1.7**, but also the following **R2.2**). Notably, this implementation allows quantifications as soon as the very first information is available (magnitude and location), making the release of the results (initially) independent from the release of focal mechanism or tsunami data. This overcomes one of the main existing limitations for managing tsunami warning in the near-field (e.g., **R1.12-15**). After this, the forecast can be updated through time, as new information is acquired (e.g., this is discussed further for reviewers' 2 and 3 comments, e.g. **R2.2**).

To these points already discussed above, here we also add that the need for uncertainty quantification is even stronger when sensible and regulatory relevant decisions are needed (Budniz et al. 1997 - Ref.47, and all following updates); the evacuation of large sectors of the coasts (moving people and halting all activities) may have indeed a very relevant impact, both in case of false- and missed alarms (**R1.14**). Also for this reason, overall PTF represents a major step forward for tsunami warning, equivalent in our opinion to the transition from worst-case scenarios to probabilistic hazard quantifications, which started from seismic hazard (Cornell 1968, BSSA) and is gradually extending to all hazards, including tsunamis (Grezio et al., 2017 – Ref.93). This important point is stressed in the Discussion (lines 389-396).

POINT R1.17

287 Again, the paradigm shift has already being taken elsewhere. It is true that not all the science might be in the can in the way it is being treated here, but it does so with a specific goal in mind and hence it has cut some corners.

Original Manuscript - Lines 286-290: "After successful testing, the PTF could be exploited to dramatically improve emergency decision-making. The PTF would represent a "science-driven" paradigm shift in the approach to tsunami forecasting, synergic to the "technology-driven" ongoing paradigm shift towards uncertainty reduction through enhanced real-time tsunami monitoring capability (GNSS, DART buoys, SMART cables, 4,16)."

We thank the reviewer for acknowledging that there is a clear difference between having a specific goal in mind and propose heuristic solutions, and addressing the problem with a formal approach. Here, we stress once again that the paradigm shift that we discuss in our

manuscript is about the quantification of uncertainty into tsunami warning operations, which implies the quantification of uncertainty into the tsunami forecasting and the use of this information into the decision-making process of defining alert levels.

The practical implication of this can be summarized commenting Figure 6:

PTF allows optimizing tsunami warning, based on the effective needs of decision-makers. As commented in lines 345-350 of the revised manuscript, we can see that “Intermediate-high PTF percentiles (80th or 85th) seem to optimize the performance with respect to both BMS and DM, as they perform as good as BMS in terms of correct assignments (about 85%) but they have a rate of missed-alarms similar to a conservative DM (about 5%). Even more importantly, PTF results allow better interpretation of the role of conservatism in present-day non-probabilistic methods, quantifying the trade-off between false and missed-alarms and showing that these current-practice methods are end-members within this trend.” As discussed with more details in **R1.15**, optimizing this balance is not a scientific task, but in present-day operations an unknown balance is implicitly selected by those scientists that defined the adopted non-probabilistic tsunami forecast procedure, usually without a precise idea of the selected level of conservatism.

Thus, the change introduced by PTF implies a deeper change in the scientific attitude to tsunami warning, optimizing tsunami warning operations by implementing the hazard/risk

separation principle (see Jordan et al., 2014; Ref.53; **R1.15**). This allows both tsunami scientists and decision-makers concentrating in improving their core businesses. On the contrary, going from “alert levels defined for one scenario” to “alert levels defined from a heuristic extraction from a set of predefined scenarios” does not represent a paradigm shift, as the basic principles remain unchanged (a single column in Figure 6). Indeed, to consider few scenarios to have a quick idea about the variability of the possible tsunami is certainly not a paradigm shift, neither is it to add some uncontrolled predefined safety factor (e.g. **Point R1.1** and **Point R1.3**). So far this has been mainly dictated by the impossibility to produce timely proper forecasts, in particular in the near-field, for which tsunami data and focal mechanisms are not yet available (e.g. Melgar et al., 2016 - Ref 16), but here we demonstrate that this is possible matching operational needs (e.g., **R1.7** and **R1.16**, but also the following **R2.2**).

Reviewer #2

POINT R2.1

Quantifying uncertainties into tsunami forecasting is indeed a topic of interest within the tsunami warning community. A thorough and well tested approach is presented. I think the work is convincing for the Mediterranean region where they have the corresponding probabilistic tsunami hazard assessments. I think scientists from other tsunami warning centers would be very interested in reading this paper. Because of the scientific terminology and statistical nature of the paper, it would probably be much more difficult and of more limited use to the disaster risk management community which should have the final word on such an approach, especially in relation to the what percentile should be used.

Thank you for this comment. The communication between risk managers and the scientific community is of primary importance to allow for rational decisions. Hence, we made an extensive effort to simplify the manuscript to improve readability. Of course, we must note that an effective reciprocal comprehension can be built only with constant interaction over the years.

To improve the readability of the paper for a larger audience, we tried to be more concrete, referring to existing analogous best-practices in different fields that are certainly familiar also to risk managers, and we deeply modified the statistical discussions in the manuscript re-organizing the presentation of the existing contents.

More specifically, we now introduced many examples, for example weather forecast for probabilistic forecasts (line 40), seismic risk reduction for the definition of alert levels (lines 108-114), or the passage from worst-case to probabilistic hazard for the step forward introduced by PTF with respect to present-day non-probabilistic methods (lines 394-396). We also added literature less tsunami-specific, like Fakhruddin et al. (2020 – Ref. 48), which discuss risk communication also in relation to the current COVID-19 pandemic.

We also completely reorganized the discussion of the results of statistical tests. We concentrated the discussion in a single point, presenting Fig. 5 (in the earlier version, it was first introduced in Fig. 2 and then developed along the manuscript). In this part, we better described the reason for the testing and the rationale for selecting the testing dataset (lines 280-286); we introduced explanatory panels to better discuss the way the tests work (Fig. 5b,c and relative legend); we retained the essential of testing results in the main text (lines 292-312), moving all details in the Supplementary Material (Supplementary Text S7).

To improve readability, we also removed some non-necessary acronyms (e.g., “alert level” and “tsunami forecast” are now always reported explicitly, instead of using AL and TF), and we also tried to simplify the terminology.

POINT R2.2

Time is of the essence in tsunami warning, in addition to the precision of the forecast. I would have liked to see, at least in the main section of the article a timeline for the issuance of the different products, DM vs PTF.

We thank the reviewer for this comment, as this is one important point for the manuscript. DM computation is instantaneous, while computational time of PTF is, in its present implementation, < 2 minutes (Table S4), matching operational requirements.

We added the timeline for the tsunami warning in Fig. 1, panel a, and we now specifically discuss the needs for the operational use of PTF for near-field tsunami warning (lines 119-132). We also added details on its computational time in Methods (lines 533-544), reporting for example more details on the processes dominating computational times and the potential procedure to further reduce it.

POINT R2.3

One of the limitations of the study is the events available are relatively small and there are limited observations. I agree with the authors in testing the methodology.

Original Manuscript - Lines 281-284: "This novel approach needs more extensive testing and thorough calibration of the uncertainty quantification framework against tsunami data worldwide. Here, we have set the scene for both hind-casting and blind tests of PTF performance against events of any magnitude, similarly to other applications (<http://cseptest.org/>). We are also preparing to exploit modern high-performance computing infrastructures⁶⁰, enabling quantitative testing of tsunami forecasts and their underlying science worldwide."

In this review, we significantly extended the analysis for larger magnitudes.

At first, we integrated the testing dataset by adding the Mw7.0 Samos-Izmir earthquake occurred in the Mediterranean during the review period; this earthquake is now the largest event in the series and produced a significant tsunami with local run-up values up to 2 m (Yalciner et al., 2020 – Ref. 95). This testing dataset is used to test tsunami forecasts as well as alert level definition (Figs. 4 and 6 and related text, all data and statistical tests reported in the text, tables and figures of the supplementary material). The results of the testing for this event are reported in Fig. 5a (as example of testing tsunami forecasts). Notably, this event (along with the 2020 Mw 6.7 Ierapetra event) are blind tests for PTF, as the complete evaluation system was in place before the events occurred (lines 303-306).

To explore even larger magnitudes, which were previously represented only by the NEAMWave17 synthetic scenario (now discussed only in the Supplementary Material, Text s6), we also added in the revised manuscript the 2010 Mw 8.8 Maule earthquake; one of the largest events in the last century that caused a very significant tsunami in Chile. This work has been possible thanks to the computational resources made available through a PRACE project (added in the acknowledgments) focused on testing the PTF against events in the

Pacific. PTF results are evaluated and compared with all available data (run-up, DART and tide-gauge). This case study is discussed in the main text (see lines 252-279 and Figure 5). Notably, also the results of this case study are statistically tested (lines 306-309 and Supplementary Text S7).

POINT R2.4

A caveat within the discussion is that the tsunami forecasts are based on the tsunami associated directly with the earthquake, if the earthquake generates a landslide then there could be deviations. This should be noted along with the deviations associated with local harbor conditions in the case of the Balears.

We agree. We now revised the text (lines 249-251).

POINT R2.5

Some other specific comments:

Line 21 - missing the word "data" after the phrase "seismic and tsunami"

Fixed, thanks.

POINT R2.6

Line 29 - define the time window, "short" can mean different things to different readers

Original Manuscript - Line 29-30: "Tsunamis pose a hazard that may strike a coastal population close to the earthquake within a short amount of time."

Modified accordingly (lines 32-33), thanks.

POINT R2.7

Line 77 missing "which" before occurred

We revised this part of the manuscript, thanks.

POINT R2.8

Line 78 What if long term frequencies are not available as part of the Hazard Definition as part of the PTHA

Original Manuscript - Lines 77-82: "PTF rigorously embeds uncertainty in TF by quantifying a probability distribution that describes the uncertainty of tsunami intensity (e.g., run-up or wave height) at each forecast point (Fig 1). Uncertainty is managed through an ensemble of tsunami scenarios defined by a set of sources consistent with available real-time observations (Fig. 1a; of any type, e.g. seismic, geodetic, tsunami) and local hazard information (Fig. 1b; pre-calculated earthquake and tsunami scenarios and their long-term frequencies)."

If long term frequencies are not available, existing global studies (e.g., Kagan and Jackson, 2014, <https://doi.org/10.1093/gji/ggu015>) or ad hoc regional or local models may be developed to supply the required information (e.g. Roselli et al., 2018, <https://doi.org/10.1093/gji/ggx383>). We add “*derived from hazard and/or other long-term forecast models*” at the end of the sentence (lines 97-98). As commented in the revised manuscript (in Methods, lines 618-621 “*If this information is not available from previous long-term studies (at global, regional or local scale), maximum ignorance can be modelled using uniform distributions, until real-time information (e.g. focal mechanism and/or moment tensor estimations) become available*”).

POINT R2.9

Instead of using the term tide gauge, suggest "coastal sea level station"

We modified accordingly in its first appearance, reporting “*coastal sea level station data (hereinafter, tide-gauge data)*” (line 220). Similarly, we modified accordingly also other parts of the manuscript, like for example Figure 3 caption (line 450).

POINT R2.10

Figure 2 - what are the broken line circles in inset c

The dashed lines indicate local and regional areas, as defined by NEAMTWS DMs (Table S8). We added this information in the caption.

POINT R2.11

Figure 4 is hard to read, event in digital and zooming in

We significantly revised figure 5 (figure 4 before revision), deeply reorganizing its content. Most of the panels are now reported in the supplementary material (Figs. S4 and S5), in a larger and more detailed version.

Reviewer #3

POINT R3.1

The study introduces a tsunami forecasting method that quantifies the uncertainties soon after the event occurrence. The test of 12 tsunami events in the Mediterranean region proves that their method (PTF) is able to make a tsunami early warning while controlling the missed alarms and false alarms effectively.

Compared with the traditional method using Decision Matrices, the improvement is evident with regard to the missed-vs-false-alarm balance. Hence, the research is of high importance to disaster mitigation. Though the methodology part needs more clarification, and some figures need further improvement, I agree that this manuscript can be accepted after a minor revision.

Several specific details have been added in describing the methodology, like for example the discussion of the timeline of the assessment and the specification of ensemble definition (lines 119-151 and lines 538-544), the details on the required input information (lines 500-505), the management of the magnitude and spatial uncertainty (lines 586-589 and 598-604), or the specifications of the tsunami propagation model (lines 684-699). We also added all the specifications regarding the extension of the implementation to any other source area (e.g., lines 645-649 and 701-708).

Moreover, we significantly improved Figures 2, 3, 5 and 6 (previously Figure 2, 3, 4, 5), by generally trying to make them more informative, immediate, and readable.

POINT R3.2 & R3.3

My comments are as follows:

L21: occurrence -> occurrence

L24: missed-vs false-alarm -> missed-vs-false-alarm

Text modified accordingly, thanks.

POINT R3.4

L57-58: The DM method connects the seismic source parameters to the ALs, but it is unfair to say the DM mixes scientific TF with non-scientific choices.

Actually, DM method also evaluates the tsunami potential and then transforms them to ALs, though there exist large uncertainties (<http://www.ingv.it/cat/en/tsunami-alert/alert-procedures/decision-matrix>). The new PTF method describes such uncertainties clearly, but it does not change the operation flow completely.

Original Manuscript - Lines 56-58: Besides, methods like DM directly connect the seismic source parameters to the Alert Levels (ALs) linked to practical actions, thus mixing scientific TF with non-scientific choices.

It is true that the DM provides a basic forecast (for example, > 1 m for level “Watch”). However, since in the beginning the earthquake focal mechanism is not known and tsunami numerical simulations are not used, DM includes in this forecast some conservatism as a substitute for uncertainty estimation. Think for example of alerting the whole Mediterranean Sea for a magnitude > 7.5, without distinctions or even without considering sub-basins. Therefore, DMs intentionally introduce a bias in the forecasting, in order to minimize missed-alarms, introducing as a consequence a certain rate of false alarms. This goal is perfectly reasonable, but it is not a scientific task (it does not improve the forecast), as it has the goal of reducing the socio-economic consequences. As such, it should be a task for decision makers. Similarly, also the discretization of DM’s forecasts in three levels, corresponding to the used alert levels, is introduced to facilitate the definition of actions. For these reasons, we state that DMs mix scientific with non-scientific choices. This is valid also for any non-probabilistic method. As Fig. 6 demonstrates, any single choice selects a specific ratio between false- and missed-alarms, potentially leading to non-rational decisions from the decision-making point of view. We re-formulate the introduction (lines 76-102) to make these points clearer.

Of course, while PTF preserves as much as possible the operation flow to facilitate its practical implementation, its workflow foresees an intermediate layer (hazard curves) that separates the scientific quantification (tsunami forecast and its uncertainty) and a decision-making phase that links the forecast (and its uncertainty) to alert levels (see, for example, lines 104-116; we deepen more on this answering to **R1.16 and R1.17**, since this is one important point of the paradigm change proposed in this paper).

POINT R3.5

L88: For tsunamis in the Mediterranean Sea, the following two papers could be cited: 1) 2003 Zemmouri-Boumerdes earthquake: <https://doi.org/10.1186/s40562-019-0149-8>; 2) 2020 Ierapetra earthquake: <https://doi.org/10.1029/2020JB020293>. These two papers focus on the tsunami data assimilation approach. It is a TF method that does not consider the source and avoids the uncertainties in the source characterization. It is also worthwhile mentioning the data assimilation method in the introduction part.

We mentioned data assimilation techniques in the introduction (lines 45-48), adding the suggested citations along with the already present Wang et al. (2019, Ref.7) and Maeda et al. (2015 - Ref.10). These techniques are also discussed in Methods (lines 500-503 and 521-523), where the input to PTF equations is discussed.

POINT R3.6

L96: The details of PTF implementation needs more clarification (also in the method part). The authors are supposed to describe clearly the number of the source ensemble, and why they choose it. In addition, more details about computing the earthquake hypocentre and magnitude probabilities should be provided (e.g., how fast it is during the operation).

Original Manuscript - Lines 95-97: "All parameters are expressed through probability distributions that collectively define an ensemble of sources consistent with observations."

We tried to add more details, in the main text, in Methods and in the Supplementary Information (Text S1).

In the main text and in Methods, we better described the definition of the ensemble, its size and its impact in the computational time (lines 144-151 and lines 538-544). We added the timeline for the estimation in Figure 1 that specifies the typical times for earthquake location as well as target warning delivery, describing which are the actual time constraints that are met by the presented PTF implementation (lines 119-132). The size of the ensemble for all earthquakes is reported in Table S3, and this is now better referenced in the main text also (lines 171-172).

In Methods and the Supplementary text, we better described the specific implementation of the uncertainty treatment and we reported more details on magnitude and hypocenter inversion procedures (lines 586-589 and 716-719 in Methods), also making reference to documentation files in which more information can be found (end of Supplementary Text S1).

POINT R3.7

L142: The four selected coastal locations are not very clear in Fig. 2. I suggest that the arrows could be even closer to the coastal locations.

We modified the figure accordingly, thanks for the suggestion.

POINT R3.8

L146: In Fig. 3c, the ALs of PTF mean does not match well with the observations, while the ALs of PTF 95th percentile are more consistent. Here more explanations are needed to discuss the performance of different scenarios and conservatism. The results of PTF 16th-84th percentile could also be added for a better comparison.

We agree that this point deserves to be better explained.

The overall spatial extension and the number of high alert levels (advisory/watch) are strongly correlated to the selected percentile (the higher, the larger the involved area). High percentiles include less likely but still possible larger tsunami intensities from the tail of PTF distributions. In the specific Zemmouri-Boumerdes case-study several peak observations correlate better with high-percentile alert levels because they were locally "anomalously" high (because of resonance, as discussed earlier in **R1.9-10**), resulting in the upper tail of uncertainty distributions. This is now specifically discussed in lines 208-217.

We also updated Figure 2 (panel c) with more percentiles. More percentiles are also reported in the figures comparing forecasts with observations (see grey bands in Figures 3 and 4).

POINT R3.9

L212: The reasons for the systematic underestimation at the Crotone tide-gauge should be explained. If such underestimation is caused by the tsunami resonance inside the harbor, could we use a finer grid size to simulate such resonance behavior?

At the time of the event, for testing purposes, we verified that using the Tsunami-HySEA code with 4 nested bathymetric grids (highest spatial resolution of 10 m) we can reasonably reproduce the signal. As discussed when answering **R1.10**, for specific target points, more detailed simulations can be added in the pre-computed scenarios, solving this kind of issue.

As this was already discussed for the Zemmouri-Boumerdes case study (lines 239-251, see **R1.10**), we preferred to report the details of all statistical tests in the supplementary material to increase the readability of the manuscript (see **R2.1**). This includes the discussion about Crotone (Supplementary Text S7, lines 320-333), in which we now we commented also on the fact that, for locations like Crotone in which signals may be systematically amplified, the inclusion of high-resolution modelling can reduce in the future the forecast uncertainty, especially for such relatively low magnitude events.

POINT R3.10

L262: Gibraltat -> Gibraltar

Corrected, thanks.

POINT R3.11

L275: It is true that existing deterministic practices are more conservative and therefore are prone to false alarms, but are they prone to missed alarms? The Fig. 5 shows that the DM methods have much less missed alarms than most PTF percentiles, which seems contradictory to this sentence.

Original Manuscript - Lines 273-276: "We presented a novel approach dealing with uncertainty in real-time tsunami forecasting, coined Probabilistic Tsunami Forecasting (PTF), for use in tsunami early-warning. Existing deterministic practices are prone to numerous false or missed alarms. PTF addresses this issue through uncertainty quantification."

We agree, this sentence can be misleading.

We reviewed this sentence, linking the reduction of missed alarms to conservative choices (lines 381-383).

We also better explained our results. Indeed, they show that the decision matrix produces also a certain number of missed-alarms (in average, about 5%). Of course, being conservative, the DM produces less missed alarms than PTF, if non conservative choices are made for PTF (e.g., using the mean PTF). However, when higher percentiles are selected, the number of missed-alarms of PTF is smaller than of DM (they are halved, from

4% to 2%), even alerting a much smaller number of forecast points (new Figure 6, panel b). This behaviour is explained now with much more details in section Results (in lines 326-337).

POINT R3.12

L305: What kind of information is 'available information' at the time t ? Here more detailed explanations are expected.

Original Manuscript - Lines 303-305: "The function $hE(x, p, t)$ can be estimated from the uncertain knowledge on E at time t based on an ensemble of tsunami simulations corresponding to tsunami sources compatible with the available information about E at the time t ."

We reviewed this sentence (lines 500-505), mentioning what is the typically available information at a given time and making reference to Fig. 1, in which we now better identified the timeline for input and output of the PTF (see also comment **R2.2**).

POINT R3.13

L338: The formulation is in principle also valid for non-seismic tsunami sources. However, for landslide tsunamis or volcanic tsunamis, the source parameters are more difficult to obtain in real time, and there are less pre-calculated scenarios in the database. More discussions are expected for the TF of non-seismic sources.

Original Manuscript - Lines 338-340: "Noteworthy, the presented formulation is in principle valid also for non-seismic tsunami sources. However, considering that present-day PTHA and TEWS are primarily focused on earthquakes and as a starting point, we here restrict our attention to seismic sources."

Thanks for the comment, we reviewed the text accordingly (lines 545-550).

POINT R3.14

L386-387: hypocenter -> hypocentre (I understand that both forms are acceptable, but the authors use 'hypocentre' in other parts of the manuscript.)

Corrected, thanks.

POINT R3.15

L386: What are the physical meanings of σ_{xx} , σ_{yy} , etc.? In Equations 1-3, the parameter σ has already been used to represent the earthquake scenario. Here it should not be used repeatedly.

We adopted the symbol s for scenarios, keeping σ for the covariance matrix. We also better explained the physical meaning of σ_{xx} , σ_{yy} , etc. (lines 598-604).

POINT R3.16

L447: I could not find Eq. S2 in the Supplementary Information. Is it missing?

We were referring to Eq. 2 (reported a few pages above), and not S2. We corrected this in the text.

POINT R3.17

L458-466: More information about the tsunami simulation are expected. For example, the authors should write the cut-off wavelength of the filter, and the time step of numerical simulation. In addition, in this study, the tsunami numerical simulation is obtained as a linear combination of synthetic tsunamis produced by elementary sources. Therefore the nonlinearly effects cannot be considered. This limitation should be discussed.

We revised the text in the section “Quantification of PTF’s propagation factor” of the Methods, reporting all the requested specifications (lines 684-699). Regarding the nonlinear effects which may be more important during inundation, we haven’t dealt with them explicitly; however, a log-normal distribution is superposed to simulation results (e.g. ref. 93-94 and references therein) to account for non modelled tsunami features (see lines 710-724). This is discussed in presenting the results (see **R1.10**, **R3.8** and **R3.9**).

POINT R3.18

L477: A closing bracket is missing in Eq. 4.

Corrected, thanks.

POINT R3.19 and R3.20

(R3.19) L520-523 & Fig. 4: For seven events without evidence of a tsunami, the authors tested that 0.1 m is unlikely at all the locations. However, for those tsunami events, they verified that 0 is not in the tails of the distribution. These two criteria appear in Fig. 4 at the same time, which makes them very difficult to understand. I suggest that the authors should use different colors or symbols to distinguish the two different tests, and to divide the test cases into two categories: with or without tsunamis.

(R3.20) Last but not least, the font size in Fig. 4 could be larger.

We thank the reviewer for pointing this out, which clearly caused a misunderstanding (see **R1.11**). We significantly revised Figure 4 and the presentation of these results (see **R2.1** and **R2.11**). In particular, in Figure 4 we report only two panels to explain better the meaning of the statistical testing in the two cases (with and without observable tsunami), while all the results are reported and discussed in the supplementary material (Figs. S4 and S5 and Supplementary Text S7).

APPENDIX - Short review of the standard operations in the Japanese and Australian tsunami warning systems

The **Japanese system** has been revised after the great 2011 Tohoku tsunami, mostly for dealing with magnitude underestimation in the first 3 minutes after the earthquake origin time.

The new implementation is described in a document available online (JMA 2013, “Lessons learned from the tsunami disaster caused by the 2011 Great East Japan Earthquake and improvements in JMA's tsunami warning system”; our reference 26 in the original version – now Ref.31).

The new approach is summarized in the following figure:

They also write: “When a large earthquake occurs, the operation system quickly calculates its hypocenter and magnitude, searches the tsunami database with reference to these calculations and selects the most closely matching results.”.

This was already in place before 2011, associated with a “maximum risk” procedure, which selects the maxima of the maxima from a set of plausible scenarios (Kamigaichi, 2014; our Ref.44).

The main difference between before and after Tohoku is the box “Mj underestimation possibility checking”, for which “When JMA recognizes the possibility of underestimation in a calculated magnitude [...] qualitative terms such as Huge and High are used rather than quantitative expressions because the uncertainty of the magnitude is considered to be large”. In other words, “The Agency now issues initial tsunami warnings based on the predefined

maximum magnitude when initial estimation for the scale of the tsunami source (i.e., crustal movement on the seafloor) is uncertain”.

Further than this, JMA improved the system in which magnitude is estimated, updated the timing of messages following the first (3 min), enhanced the observation facilities, and a change in the classes of tsunami intensity to more qualitative ones in recognition of the possibility of large forecast uncertainties, and with “Tsunami height estimations [...] issued as the upper-limit value for each class to create a sense of urgency” (see also discussion answering **R1.5**). Even after sea level anomaly measurements, “JMA issues Currently Observing announcements rather than exact values”, again to prevent the perception of a false sense of precision and/or to avoid negative consequences from initial underestimations.

Therefore, the Japanese systems works with the single Best-Guess Scenario (BGS) (the new reference non-probabilistic method to which PTF is compared in the revised manuscript), accompanied by a “safety factor” (the maximum-risk method); then, if the magnitude may be considered as underestimated, “*qualitative terms such as Huge and High are used rather than quantitative expressions because the uncertainty of the magnitude is considered to be large*”. Finally, “Around 15 minutes after an earthquake, JMA updates tsunami warnings based on more precise analysis with a Mw value and tsunami observations, and issues information on estimated maximum tsunami heights in quantitative terms [...]”.

The different aspects of the **Australian system** are described in Allen et al. (2008 – Ref.37), in Greenslade et al. (2009, Ref. 39), and in the website of the Bureau of Meteorology (BOM, an Australian Government agency, www.bom.gov.au, and more specifically in the webpage <http://www.bom.gov.au/tsunami/about/atws.shtml#:~:text=The%20Australian%20Tsunami%20Warning%20System,tsunami%20warnings%20to%20the%20Australian>).

The system is described by the aside figure.

In a few words, “*Geoscience Australia [...] advises the Bureau of the magnitude, location and characteristics of a seismic event which has the potential to generate a tsunami. Based on this seismic information from GA, the Bureau runs a tsunami model to generate a first estimate of the tsunami size, arrival time and potential impact locations.*”.

However, being Australia in the far-field of tsunami propagation, they do not have the urgency of early (few minutes) estimations.

We further inquired about it by informally contacting our Australian colleagues. They confirmed that there is not any "formal" uncertainty treatment in place. They rather await an Mw/location estimate, and then select a

set of plausible scenarios from the T2 scenario database (all uniform-slip scenarios with the right Mw that contain the epicentre). They do not perform any formal DART inversion. As new data come in, revisions to the forecast are done based on expert judgement rather than algorithmic processes.

REVIEWER COMMENTS

Reviewer #1 (Remarks to the Author):

This is my second review of the manuscript submitted by Dr. Selpa and collaborators. On it, they present a methodology to estimate probabilistic tsunami forecasts within the context of tsunami early warnings. The authors have made a thorough presentation of their response to the previous comments I raised, and included some amendments to the text and the analysis, which I find now more easy to follow and I can understand better the intended meaning. Thank you for the effort, and clarity on the response.

Unfortunately, due to a poor communication from my part, the core of the objections I raised to the previous version apparently was not available to them, and they were only aware of my specific comments. I find that the essence of my previous concerns is still present in the work, and I think it is necessary to address them. Perhaps I took a too subjective approach to my wording that last time, so this time I will try to be more precise.

Generally speaking, the article is well written and the methodology as presented is sound; it provides a formalization and a clear extension of schemes to estimate the tsunami hazard; and applies them to the context of early warning. As for the novelty, I am somewhat skeptical of this claim, as the core of the procedure and its aims are already present, even operationally, in some TEWS around the world. I am aware that the authors assert that this is not the case, as no system provides a probabilistic framework. I concur with this. But at the same time, some systems are aware of the inherent uncertainties, and even sample multiple scenarios, and even rank them which essentially could yield a Cumulative Density Function as obtained here. But they decide to choose not a give probability threshold, but the worst case (which in probabilistic terms would be the PTF P99,999999 or thereabouts). Hence, there was a prior decision regarding what would be the acceptable probability, and the TEWS then proceed forward. This is the case of the Chilean system, (see Catalan et al, 2020), where in their Eq.1 they implicitly suggest that the CDF of the data is sampled at each coastal point, very similar to what it is done here. Of course, the details and background methodology are different. The manuscript does introduce it formally within the context of uncertainties and probabilistic assessments, and does not discard the other scenarios as the other system does.

But the focus of the written work (not the method!!) is different: the most important claim of the work is that this extended information is useful for the decision making. It is of note that in this context, the manuscript does offer a shift in paradigm, as they rightfully claim: The a priori decision of other TEWS to focus on worst case, or high percentile probability has framed how they approach the problem. Here, the authors propose that this decision can be made afterwards or deferred to others. Whether this is meant to take place during an event, or if the the decision what would be the appropriate threshold percentile before implementing the system operationally, is not clear in the text, and it is not discussed. I infer they refer to the former, that is, to use that information in real time.

If this is the case, I disagree with the statement on Line 379: "We present a novel approach dealing with uncertainty in real time tsunami forecasting and alert level definition for tsunami early warning". This is not backed up for the work as presented. Perhaps the wording is confusing, as the methodology "does deal" with uncertainty, but it has not been discussed "how to deal" with that uncertainty within an early warning. These are very different things, and that is the crux of the problem with the article: it correctly "does deal" with the uncertainty, but it does not tell us "how to deal with it". The text seem to focus on the latter, and therefore it strides away from its strengths.

Someone has to make a decision regarding the issuance of evacuation or other actions, and that needs to be fast. Very fast, and resolute. So how does this method helps in dealing with that uncertainty? What would be the information used by the decision maker to determine whether it is a false alarm or not? More controversially, can a system afford taking this route (see below)? The article sets a strong tone towards the aim of minimizing false alarms, which are indeed an Achilles heel of TEWS, but does so at the expense of minimizing the role and effect of missed alarms, albeit implicitly.

So while technically the work is sound, the mindset present in the article highlighting the technological possibilities and benefits of the system, overshadows the real life challenges that a decision maker would confront. I agree that this is a shift of paradigm, and they claim this upfront. However, my concern is that in doing so, they have overlooked some not-so-insignificant details when analyzing their results.

For instance, the assessment of the benefits of the work focuses on the statistical analysis of the 13 cases. I agree that results are promising, but it is of note that in Fig. 5b, the distribution is skewed towards cases where the misfit underpredicts the relevant Tsunami Intensity Measure. Hence, more cases of the method show to be less conservative. This is consistent with the PTF-Mean having a somewhat large number of missed alarms in Fig 6. But, this is the ensemble of 13 cases. Individual cases show a lot of variability on their individual performance. This is the "does deal" part. But the "how to deal" part poses the following questions: Do you expect to have a standard threshold for the decision maker? If yes, how do you expect it can be defined? If not, how do envision someone can make a decision with these data? How would they know where do they fall in terms of overprediction or underprediction, in the absence of data (you talk about 2 mins after getting the EQ data, so is is full forecasted information)? Again, I am aware that you intend to defer that decision outside of the scope of the article, but I think it is very relevant that you could at least provide some guidance. Otherwise, we haven't learn how to deal with the uncertainties, and its usefulness remains questionable.

The approach to the analyze some of the results can also be seen here: In analyzing the specific case of Zemmouri-Boumerdes, the authors claim " a few observations from Eastern Spain ..." (L230). I counted that 6 out 16 fit the criteria. That is 38%, hardly "a few". Similarly, in L232-233 they claim that "simulations ...largely fall within PTF's inner confidence interval". Rather than largely, I think it is "barely", as 7/16 of the data are clearly in the outer confidence intervals in Fig 3 bcd (this is very different for Fig.4, but I digress). What I mean is that in the text, the more optimistic and uplifting tone for the work results is often present. This also may explain overlooking that in Fig. 2c, the PTF P95 best matches quite well the observations (bar overpredicting in the near field). This can be seen to lend support to the system that aims for the worst case scenario, (not the BSM!!). This is not discussed.

The other aspect is that one of the premises of the work is insufficiently backed up. The excessive rate of false alarms is mentioned as something to be avoided several times in the text, and guides the narrative. I note the reference is Bernard and Titov, 2015, but I think they were reporting older, before 2004, systems. Whether this issue has been quantified in modern systems I am not aware, I confess. However, what is absent in the article is what would be the effect of missed alarms, and whether that cost can be afforded by society. In some countries, missed alarms mean lives. As before, I am aware that the authors would like to defer this decision to others, but it leaves the premise for the work on unstable ground.

I have some specific comments as well:

L42: The idea of excessive false alarms is introduced here. How does the method helps in reducing these false alarms is somewhat presented later, and also the balance with conservadurism. But what is the cost of missing alarms is not mentioned. I don't feel at ease giving a higher weight to false alarms over missed alarms, as it is implied in the text.

L47 "directly constrain" : I'd say just "help to constrain". They are also subject to uncertainties.

L61: The TEWS of Chile, Catalan et al, 2020 (Coastal Engineering Journal) takes a different route. It is not considered in the analysis nor the review.

L70-72: The author mention that "Yet, quantification of uncertainty is critical in managing such situations". I find this non sequitur, as if the incoming data is in error, the resulting analysis would also be in error, probabilistic or not. The resulting computed uncertainties can therefore be meaningless.

L76: I don't follow what is meant by "single intensity" here.

L80: "However, this process may lead to non-rational choices". It is a strong sentence, and I don't see the non-rationality. Is trying to be conservative to save lives non-rational? Perhaps it did follow a rational assessment and it was found that the cost of false alarms can be afforded whereas no-action can not?

L84: I think this a somewhat fallacious argument. Other phenomena can optimize the risk reduction management by using probabilities because the time-and spatial scales considered allow for that. For instance, in weather prediction. But a tsunami is very fast and wide ranging. Risk reduction during an emergency is performed by reducing exposure (ie, evacuation) and every minute gained is precious. I am not so sure whether these tools really allow for such "optimization" in the case of tsunamis. This also affects to L109-116, where the authors compare with seismic design and PTHA. The difference is the time scale of evaluation, which allows for estimation of uncertain occurrence of events, that is, in terms of recurrence rates. In TEWS, the event has occurred and we need to deal with it. The mitigation strategies are significantly different, and the data sources, and methods might not be transferrable.

L94: Throughout the text the term "tsunami intensity" is used, but there can be many Tsunami Intensity Measures (TIMs). I gather most of the time the authors refer to tsunami amplitude at coastal points (peak coastal amplitude) and/or DARTs, but sometimes they use runup. Perhaps it would be best to be specific in terms of the TIM of choice on each case. That being said, since they are not modeling runup, it could be discarded altogether.

L200-212: I miss the analysis of the case that best matches observations: P95. It is fairly close to the envelope approach used in the Chilean TWS for example, which is very conservative.

L214-217: Here lies a key problem: How does a decision maker would know some observations (or the tsunami itself) were particularly large? I find a bit baffling that this is not analyzed in detail with regards to the real-life benefit of the PTF under such circumstances.

L230: Are 6 out of 16 really "a few"?

L229: What does it mean "well reproduced"? That they fall within the range of variability modeled? That they fit a given percentile? Too vague a statement.

L236: "it can reproduce the tsunami uncertainty" What do you mean by reproducing the uncertainty? is it the range of variability?

L239-240: So the modeling does not reproduce phenomena that could lead to amplification, yet the approach is to reduce "conservadurism" using a model that is incomplete. It seems risky, to say the least. It is true that some of the results do encompass these cases, but since you don't provide guidance regarding what would be the appropriate P% to be used, we don't know whether this is a problem, or not. This is fairly inconclusive.

L248: Are the references correct? There is an ample body of literature working on tsunami amplification (Eric Geist, Alex Rabinovich, and more recently the Chileans) but these references seem to point in other direction.

L264: Are you modeling runup or PCA (figure legend says Maximum amplitude)? if runup is not modeled, perhaps is better not to consider it. It is a different TIM altogether.

Fig 3 vs Fig 4: They offer a very contrasting performance of the method, where in the ZB case models tend to underestimate whereas in Maule is the opposite. Then, how can this be generalized to come with a certain degree of guidance on how to actually use the PTFs?

Fig 4: Note that Catalan et al, 2020, also compare their method against Maule 2010.

L267-268: "A possible reason ...wavelengths" This sentence is fairly speculative, and it is not backed up since Chile is also subject to active resonance and amplification, during several tsunamis: See Catalan et al, 2014, GRL, Cortes et al., 2017, Aranguiz et al, 2019. The authors imply that it affects more the events in the Mediterranean?

L296: "Tsunami data and forecasts are compared". I think is relevant to specify that you compare against the mean (which is not a proper probability btw, so how does it fall within the PTF method is unclear). The result will vary if you compare against other PTF threshold of choice. (perhaps a sensitivity analysis is in order?)

L321: Here we can see where the PTF is somewhat sugar coated. □The simpler BMS yields results that are fairly comparable with most of the PTF results (Fig 6, Table S9), and even yields better results (on average) than the PTF-mean, that is the base comparison for Fig. 5. Most of the following discussion focuses on the DM method (L322-338), whereas the comparison with BMS is mostly colloquial and vague in the main text (eg, L341: "Nevertheless, with higher percentiles, PTF enable a reduction of missed alarms while keeping a high rate of correct assignments"). The reader is referred to table S9, (note the use of BGS instead of BMS there), where the differences are minimal when we reach P75 or above, but then we would need to compare against the worst case scenario, I would think. The wording in the text seems to obfuscate this.

L343: The sentence is vague, and the sample percentiles are inconsistent within the parenthesis, as four values are presented, and only three PXX values are mentioned.

L345-350: Although a more fair comparison with BMS is presented here, there are vague statements. For instance, what does it mean to "optimize performance " in L345? Next, PTF are mentioned to perform "as good as BMS" (so does BMS take the upper hand?)..., "but they have a rate of missed alarms similar to a conservative DM". I can not figure whether this is good or bad in the eyes of the authors (inconclusive). Also, I don't find the reported value of 5% missed alarms of the conservative DM (L347), and I read it to be about 55% in Fig. 6 and Table S9. I guess is a typo, but then they are not comparable.

L349-354: All of this is certainly correct, but other than stating that some approaches can be conservative what do we do with this? Also, I don't think is true that DM and BMS are the end members of the class. Clearly BMS performs as well as the PTF method for P60-P75 (you say so in L346). On the other hand, the PTF values for P55 or lower have larger missed alarms (certainly a bad thing?), and the PTF-mean has more false alarms. Hence, the BMS method falls somewhere in between, not an end member. The bottom panel of Fig.6 is thus misleading, and the PTF has a whole has a wide range of performance.

L355-356: I strongly disagree with this statement, but I recognize that this is a subjective standpoint. As a scientist, I think we should be transparent (what you aim here), but also offer guidance in how to interpret and use the results. No action needed.

L355-356: What do you think would be the Key Performance Indicators in such a cost-benefit analysis?

L361: What if a version of the cost-benefit analysis has been done by decision makers of systems elsewhere and the result is that false alarms are acceptable, but missed alarms are not? Is there evidence that this analysis has not been considered? (the only tsunami reference here is somewhat dated with regards to current TEWS)

L388: "This allow decision-makers to make more conscious decisions considering uncertainty". In the way it is presented in this work, this seems aspirational rather than a reality. The method presents the uncertainty, but leaves completely unanswered the question of how it can be used. Therefore, to say that "it allows more conscious decisions" is unsupported. It doesn't mention either when this decision can or should be taken. In real time? In advance, and have a threshold PXX% to be used throughout? Those are two of many options.

L394: "performance can be optimized". Again, what is the cost function to be optimized is left open. We don't know if such an optimal point even exists.

L400. Indeed, the results are statistically sound. But so it's the BMS, as per your own comparison. But statistical accuracy does not solve the problem to tell the people when to evacuate.

L410: "After further successful testing". What is the metric of success here? I don't know at this point whether the PTF has been shown to be successful. We have a new, very polished method. True. And what else?

L410: "exploited to dramatically improve emergency decision making" seems very aspirational, and it is not supported by the evidence presenter herein.

L411: "science driven": I think this is technique-driven. The development of other TWS has also been supported by the science available at the time. Some of the basic geophysical science regarding earthquakes and tsunamis is similar among those other systems and this. The methods, models and computational power, have allowed several developments. But at the end of the day, the question that underpins a TEWS is how to deal with lack of accuracy. What this articles provides is a sound methodology to have more data available for the decision, but whether its used, or whether it is useful, remains to be seen.

In summary, I think the methodology and approach used are sound, it is carried out very formally and properly, and attempt to maximize the amount of data available. I think this is worth to published. However, I think the article narrative points on a different way. As mentioned above, whether these data is what is needed or how it can or should be used, the manuscript in the current state does not offer enough support to objectively validate it. Even if upon revision it is found that these data are useful within a TEWS context, the authors should consider to explain or provide actual guidance regarding how it could be used. As of now, this is left unanswered and many of the somewhat grand claims of the article lack sufficient support as of now. I recommend a revision aimed at clarifying the article's goals, and how to validate them properly within the analysis and discussion.

Reviewer #2 (Remarks to the Author):

This version has several significant improvements, as noted prviously the authors are presenting a novel methodology for addressing the establishment of tsunami threat and warning. It will influence the field of Tsunami Early Warning Systems. By including more recent events in the Mediterranean, as well as the Maule event, the analysis and discussion of the proposed methodology is strengthened.

A few points...

Lines 17-18 say: Tsunami warning centres face the challenging task of rapidly forecasting impending tsunamis immediately after an earthquake, I recommend saying "Tsunami Warning Centres face the challenging task of rapidly forecasting tsunami threat immediately after an earthquake"...

Lines 27-28 -national italian Tsunami Warning Centre if national is part of the name of the Italian Tsunami Warning Centre, the "n" should be capitalized, if not "national" can be removed as it is implied in the name.

Lines 33-34. Text sasy "Tsunami Early Warning Systems (TEWS) must forecast the expected tsunami rapidly

following any potentially tsunamigenic earthquake." consider "Tsunami Early Warning Systems (TEWS) must forecast the tsunami threat rapidly following any potentially tsunamigenic earthquake."

Lines Many TSPs and NTWCs use BM and DM processes just for initial forecasts, therefore lines 49 and 62 I suggest putting the word "initial" before the word forecasts.

Line 63 - I think the strategies used by the NTWC and TSP are not "ad hoc" they have been developed based on the study of historical tsunamis and research.

Lines 262 and 414 the word buoy after DART, the buoy is only the communication platform of the DART system).

Page 10. When referring to the new methodology as a move to "science driven paradigm shift", implies that current methods are not driven/based on "science", when this is not the case, eg. Green Laws functions for computing coastal amplitudes, CMT solutions couple with DART observations, etc. --

The article refers a lot to tsunami intensity, but this term is not defined, but based on its use it refers to the tsunami amplitude or run up. According to the UNESCO - IOC tsunami glossary, tsunami intensity is "Size of a tsunami based on the macroscopic observation of a tsunami's effect on humans, objects including various sizes of marine vessels, and buildings". Either define intensity for the purpose of the article or choose another term, eg water level.

Christa von Hillebrandt-Andrade

Reviewer #3 (Remarks to the Author):

The updated manuscript is presented more clearly and logically after the revision. In the methodology part, the authors added clarifications on the details of PTF implementation. They also explicitly discussed the behavior of missed alarms to different conservative choices. The quality of figures was greatly improved. For example, in Figure 5 (previously Figure 4), the author separated the earthquake events without evidence of a tsunami, and those with a tsunami, to test the PTF model again earthquakes. The revised graph is easy-understanding. Hence, I am satisfied with this version, and I recommend that this paper could be accepted. I have one minor comment on the example of Tohoku earthquake (Line 69-71). I agree with the first reviewer, that the failure of tsunami forecasting of the Tohoku earthquake is mainly because of a poor assessment of earthquake. The earthquake was so large that seismometers at near-field stations were off-scale, making it unable to solve the CMT solution initially. This is more relevant to a problem of hardware device, rather than the lack of uncertainty quantification. The authors could consider removing this example to avoid misunderstanding.

Reviewer #1

POINT R1.1

This is my second review of the manuscript submitted by Dr. Selva and collaborators. On it, they present a methodology to estimate probabilistic tsunami forecasts within the context of tsunami early warnings. The authors have made a thorough presentation of their response to the previous comments I raised, and included some amendments to the text and the analysis, which I find now more easy to follow and I can understand better the intended meaning. Thank you for the effort, and clarity on the response.

Unfortunately, due to a poor communication from my part, the core of the objections I raised to the previous version apparently was not available to them, and they were only aware of my specific comments. I find that the essence of my previous concerns is still present in the work, and I think it is necessary to address them. Perhaps I took a too subjective approach to my wording that last time, so this time I will try to be more precise.

We thank the reviewer since this extended review helped us significantly improve the manuscript. We are confident that this has greatly helped us to solve several issues and, overall, to improve the presentation of the results. Now several discussions have been reviewed to make them more focused and clearer, thus somewhat adjusting also the narrative of the manuscript. We also made a balancing effort to avoid altering the elements that were already well received by the other reviewers.

Moreover, several elements in our manuscript that could have led to some misunderstandings have been better highlighted. We had in fact the impression that there were at least a couple of important misunderstandings. Perhaps, some of the concerns were related to them.

First, it is not our intention to demonstrate that specific percentiles of the PTF work better than others, thus suggesting to reduce conservatism. It is beyond our scope and competences to recommend one of those recipes over another to the decision makers. We only intend to illustrate that, once the political choice by the decision makers is included, the PTF can be carried until the alert levels, implementing any desired level of conservatism. We show (as in the previous version) different ways in which alert levels could be defined (using various alternative recipes), and we focus on the simplest one, based on percentiles. Then, we show that, with different percentiles, PTF covers the full range of conservatism of present-day methods, and that each percentile can be connected to a quantitative definition of the level of conservatism and to an expected rate, over multiple events, of correct/missed/false alarms. We do not provide indications on how to select a specific percentile. This would lead to changing the scope of this manuscript. However, we do provide input information and potential recipes to enable the decision makers to - possibly - take their political decisions. This is better discussed later in this letter.

Second, we wish to make clear up front that the decisions cannot or should not be made in real-time; rather, a predefined rule should be applied automatically.

In what follows, we provide point-by-point answers, and we also detail how we modified the text to meet the raised concerns. Note that all references to specific lines are referred to the track-changes version.

POINT R1.2

Generally speaking, the article is well written and the methodology as presented is sound; it provides a formalization and a clear extension of schemes to estimate the tsunami hazard; and applies them to the context of early warning. As for the novelty, I am somewhat skeptical of this claim, as the core of the procedure and its aims are already present, even operationally, in some TEWS around the world. I am aware that the authors

assert that this is not the case, as no system provides a probabilistic framework. I concur with this. But at the same time, some systems are aware of the inherent uncertainties, and even sample multiple scenarios, and even rank them which essentially could yield a Cumulative Density Function as obtained here. But they decide to choose not a give probability threshold, but the worst case (which in probabilistic terms would be the PTF P99,999999 or thereabouts). Hence, there was a prior decision regarding what would be the acceptable probability, and the TEWS then proceed forward. This is the case of the Chilean system, (see Catalan et al, 2020), where in their Eq.1 they implicitly suggest that the CDF of the data is sampled at each coastal point, very similar to what it is done here. Of course, the details and background methodology are different. The manuscript does introduce it formally within the context of uncertainties and probabilistic assessments, and does not discard the other scenarios as the other system does.

This last sentence describes exactly the principal difference between our approach and the others (e.g., Catalan et al. 2020): we provide the decision-makers with tsunami forecasting together with the full uncertainty treatment in a clear and transparent way. This decouples the tsunami forecasting (research + technology) from the decision making (e.g. from civil protection agencies). The rules for the latter component have to be determined in advance and involve a political decision.

It is true that most tsunami early warning systems “are aware of the inherent uncertainties” and try to account for them by, e.g., taking the credible worst case or envelope of possible scenarios (like in Catalan et al, 2020 and GITEWS); by doing this (calculating the scenarios and picking the worst), the scientific and political aspects are fused together and exerted at the same time. Probably, it is agreed in advance with the political authorities that the choice will be worst-case oriented.

However, this is not equivalent to formally assigning a probability to each scenario and combining the scenario outputs weighted according to their probabilities into the forecast, based on effective available real-time information. If quantitative information about “how certain is this forecast” is not available, is not based on the effective real-time uncertainty on observations, and is not communicated, it is not possible to be sure regarding the degree of conservatism that is being applied. For example, how does the “pre-selection area” for the scenarios compare with the effective hypocentral uncertainty in real-time? How much this may vary through time? In a sense, decision makers do not have all the necessary information to rest assured that their indication of adopting the worst case scenarios has been actually applied.

To show as clearly as possible that this is the case, we anticipate here that in this new version of the manuscript we have implemented a third term of comparison for the PTF, besides the decision matrix and the best guess scenario that were already present. This new term of comparison, which uses the envelope of all considered scenarios in real-time, is of the same type as the one used by the Chilean system as detailed in Catalan et al. (2020) and later commented by the reviewer. We found (see Figure 6, right-hand side bar) that this approach is comparable to the 95th percentile of the PTF and it is potentially subject to occasional false alarms in some cases and at some locations, for the specific use-cases considered. This finding somehow contradicts the Reviewer’s claim that this approach performs as the PTF at 99,999999 or thereabouts, at least in this implementation extended to manage non-subduction sources. PTF allows quantifying this explicitly.

Then, PTF is a paradigm change in the sense that our method (i) clearly and transparently quantifies both the forecast and its certainty level, (ii) uses this (more complete) real-time information to define uncertainty-informed alert levels, (iii) conveys the full information to the decision makers. This is the reason why the PTF can be considered truly novel. The core of the PTF method is the quantification in real-time of the probability distributions, as graphically represented in Fig. 1. Such probability distributions represent the basis of all the results, including testing the forecasts (Fig. 5) and translating forecasts into warnings (Fig. 6). This probabilistic approach to early warning, as the Reviewer recognizes upfront, is indeed a novelty.

So, eventually, we developed a quantitative tool, possibly to be further calibrated against tsunami observations, that nonetheless permits to assess the performance of existing methods in a quantitative fashion, to address what

would be on the average the expected relative amounts of false, true and missed alarms (e.g. like in Fig. 6). The conversion algorithm from PTF to alert levels can be tuned to meet asymptotically the choices of the decision makers. We explain this last part in more detail in the next answers. Anyway, we definitely consider this as another novelty.

Now we modified the text to better declinate the PTF method in this respect (e.g. lines 99, 146, 390-391, 400-402, 432-438, 453-455, 472-473; line numbers refer to the track-changes version of the manuscript). In particular, the revision of the discussion about Fig. 6 (lines 355-419) further clarifies the central role of the probabilistic framework. As said, an “envelope procedure” derived from Catalan et al. (2020) is now implemented as a third non-probabilistic procedure used as terms of comparison for the PTF, and its results are included into the variability results for the different PTF percentiles.

POINT R1.3

But the focus of the written work (not the method!!) is different: the most important claim of the work is that this extended information is useful for the decision making. It is of note that in this context, the manuscript does offer a shift in paradigm, as they rightfully claim: The a priori decision of other TEWS to focus on worst case, or high percentile probability has framed how they approach the problem. Here, the authors propose that this decision can be made afterwards or deferred to others.

We respectfully disagree with this last statement. As just noted, we do not intend to say that this decision should be made afterwards, but that it should be made explicit and weighted according to available real-time uncertainty information. With this study we demonstrate that we are now technically ready to produce accurate real-time tsunami forecasts quantifying uncertainty and to link this to the warning. Based on this, one can still opt for the worst-case, but on a more solid and quantitative basis.

Consequently, beside presenting and testing real-time tsunami forecasts, we also provide different examples, for illustrative purposes, on how the decision makers could address this problem starting from a range of different methods and approaches, on which to elaborate their own approach and choice (two classes of possible approaches are described in Supplementary Text S2). We then pick one of these classes (use of percentiles) to show an end-to-end application of the PTF. We discuss the impact of different choices, demonstrating that we can encompass the full range of current practice methods, and link each percentile to an expected rate, over multiple events, of correct/missed/false alarms. These choices are to be tailored to the end-users needs: we explain this again in more detail later.

We agree with the reviewer that it can be reasonable to make conservative choices (even pointing to the worst-case among the selected ones), which may be perfectly acceptable for example when human lives are at stake. We also agree that these decisions should not be made during an emergency, as we think that the rules should be set from the beginning and automatically implemented during an emergency. However, more/less conservative approaches may be needed for different targets or for different actions (evacuation, stop to harbour operation, pre-alert to fire departments, etc.), based on a predefined tolerance to missed/false alarms and/or based on the potential risk (and not on the hazard). All we are saying is that these decisions require competences that go beyond tsunami science, and should go beyond our scope. As scientists, our primary task is to provide a transparent and quantitative probability estimation, which is useful for informed decision-making. We can also suggest potential rules for defining alert levels, quantifying the potential consequences in terms of expected correct/missed/false alarm rates. Then, if within our role in the framework of the mandate of the TEWS, we are also in charge also for decision-making, we can use this scientific information in a transparent way. Our point is not to prescribe what should be everyone’s role, which is established by the national legislation and by the international agreements; rather, to make the two different phases of forecasting and decision (the “does deal” and the “how to deal”, to use the Reviewer’s definition in **R1.5**) clearly recognizable so to avoid any unwanted confusion of roles.

In summary, our claim is that while the “worst-case” oriented approaches (declined in different ways in different TEWs) have been a reasonable interim conservative approach, we are now technically ready to produce accurate real-time tsunami forecasts quantifying uncertainty. Thus, a new debate can be opened based on this new opportunity about how this information can be used in defining risk reduction actions.

We try to make this clear throughout the manuscript. To solve the important misunderstanding about the scope of our manuscript that emerges from this comment, we adjusted the narrative of the presentation of results, highlighting the starting point, our goals, and the results that we obtained and that we intend to back up, separating them from the discussions. To this end, we better defined the goals, which are now clearly described upfront, in the last paragraph of the introduction (lines 120-125), and we revised the entire manuscript, including the abstract. We completely revised the part discussing Fig. 6 (lines 355-419, more details in response to **Point R1.6**), and we either removed (e.g., 416-419) or repositioned into the “Discussion” (e.g., lines 408-416 moved to 456-467) the more speculative parts originally reported in “Results”. In the Discussion, we reviewed and reordered the content to better reflect the goals (e.g. lines 432-438, 482-487 moved to 439-444), and we enreached the discussion about the potentiality of exploiting the real-time uncertainty forecasts to improve and optimize the risk management in tsunami warning (lines 445-481), as discussed in more detail in the following points.

POINT R1.4

Whether this is meant to take place during an event, or if the decision what would be the appropriate threshold percentile before implementing the system operationally, is not clear in the text, and it is not discussed. I infer they refer to the former, that is, to use that information in real time.

We thank the reviewer for this comment which points out a possible weakness in our presentation. The definition of the alert levels must be based on “predefined rules” (lines 104-106), such as “selecting a target probability value (e.g., one particular percentile), corresponding to a pre-defined level of conservatism for risk reduction actions” (lines 115-117). “Different methods can be defined based on PTF statistics and/or on evaluating the probability of predefined TIM intervals (see Supplementary Text S2).” (lines 191-193).

In Figure 6, we show that, based on a specific PTF-to-alert conversion rule, and depending on the political choice, the expected (over multiple events) ratio of missed vs false alarms (and “true” alarms) can be selected. However, because of uncertainty, it is impossible to know in advance with certainty if in the specific case there would be a false alarm or not; but it is possible to anticipate, thanks to PTF, how many times we expect to make correct or wrong choices over multiple events, as already discussed in the previous points above.

In the revised manuscript we made more explicit this very important point by integrating the text both in the introduction (lines 88-89) and in the conclusions (line 445-446).

POINT R1.5

If this is the case, I disagree with the statement on Line 379: “We present a novel approach dealing with uncertainty in real time tsunami forecasting and alert level definition for tsunami early warning”. This is not backed up for the work as presented. Perhaps the wording is confusing, as the methodology “does deal” with uncertainty, but it has not been discussed “how to deal” with that uncertainty within an early warning. These are very different things, and that is the crux of the problem with the article: it correctly “does deal” with the uncertainty, but it does not tell us “how to deal with it”. The text seem to focus on the latter, and therefore it strides away from its strengths.

This is not the case, indeed. As discussed in **Point R1.4**, we do not foresee the possibility of defining decisions in real-time. So, probably the problem with the statement at Line 379 can be considered solved? Nevertheless, we modified the part of the text on this aspect to avoid confusion (see lines 432-438).

As already discussed at large for **Point R1.3**, we revised the entire text to make the narrative clearer and better define the goals, and here we stress once again that in the paper we address both the aspects: the “does deal” and the “how to deal”, to use the Reviewer’s definition.

The “does deal”, that is the forecasting, is fully addressed in probabilistic terms, as also agreed with by the Reviewer. This does represent the core of the manuscript, and thus to this aspect we dedicated most of the text, 5 out of 6 figures, and all the sections in Methods.

The “how to deal” aspect, that is the decision-making for converting the forecasting into alert levels, is kept separated but it is addressed as said also in the response to **Point R1.3**.

POINT R1.6

Someone has to make a decision regarding the issuance of evacuation or other actions, and that needs to be fast. Very fast, and resolute. So how does this method helps in dealing with that uncertainty? What would be the information used by the decision maker to determine whether it is a false alarm or not? More controversially, can a system afford taking this route (see below)? The article sets a strong tone towards the aim of minimizing false alarms, which are indeed an Achilles heel of TEWS, but does so at the expense of minimizing the role and effect of missed alarms, albeit implicitly.

Pre-defining the translation rules (**Point R1.4**), the procedure is already “fast and resolute”. The ratio of missed and false alarms is controlled by the uncertainties: this again links directly with the political decisions for selecting “how to deal” (**Point R1.3**). When this is decided up front, it is indeed fast and resolute.

Nonetheless, we recognize that we should tone down the rhetoric regarding the reduction of false alarms. Thank you. We now looked for a better balance to avoid disregarding the issue of missed alarms.

We tried to adopt more appropriate wording and examples in this respect. For example, we removed or reviewed all sentences pointing to false alarms (e.g. lines 19-20, 42-45, 435-437). We deeply reviewed the discussion of Fig. 6 (lines 350-4419). In particular, we eliminated the statistical testing about the percentage of missed/false alarms (also from the Method, lines 869-885), as well as all the other specific comparisons made in terms of better/worse performance. We also moved in section “Discussion” (instead of “Results”) the discussions about the competences required for choosing the level of conservatism and about the potential optimization of risk mitigation actions if PTF (or another similar probabilistic method) is adopted, leading to a paradigm change (line 408-416 moved to 456-467), while we focused the discussion about Fig. 6 on the capability of PTF to encompass the full range of conservatism levels (lines 365-389). Meanwhile, we eliminated the part discussing the cost-benefit analysis (we do not want to claim that this is our preferred solution; lines 466-467) and we integrated the text with a description of potential strategies to exploit probabilistic forecasts accounting for real-time uncertainty, in comparison with current practice (lines 464-481).

POINT R1.7

So while technically the work is sound, the mindset present in the article highlighting the technological possibilities and benefits of the system, overshadows the real life challenges that a decision maker would confront. I agree that this is a shift of paradigm, and they claim this upfront. However, my concern is that in doing so, they have overlooked some not-so-insignificant details when analyzing their results.

Thank you for having acknowledged the paradigm shift. At the same time, if we have not been clear enough regarding the general scopes of the manuscript and the possibilities that this method offers to the decision-makers, we then understand the reviewer’s concern.

As discussed above, we revised the text accordingly to try and prevent any potential misunderstanding. We have already clarified these aspects when responding to **Points R1.3 to R1.6**. In extreme synthesis, we demonstrate that we are able to produce accurate real-time tsunami forecasts estimating uncertainty, and that the rule for setting the alert levels (see examples in Supplementary Text S2) can be used to achieve, on the average, the desired level of conservatism. To this end, we presented a recipe for the decision-makers to implement their own choices, showing that all levels of conservatism can actually be reached for whatever risk reduction action (also beyond evacuation) they intend to design.

POINT R1.8

For instance, the assessment of the benefits of the work focuses on the statistical analysis of the 13 cases. I agree that results are promising, but it is of note that in Fig. 5b, the distribution is skewed towards cases where the misfit underpredicts the relevant Tsunami Intensity Measure. Hence, more cases of the method show to be less conservative. This is consistent with the PTF-Mean having a somewhat large number of missed alarms in Fig 6. But, this is the ensemble of 13 cases. Individual cases show a lot of variability on their individual performance. This is the “does deal” part.

On the one hand, to judge a model after a single event may be misleading, and probabilistic forecasts by definition are related to the average behavior over multiple events. The PTF results may show variability across individual cases: this is due to the uncertainty, and it is the reason why a probabilistic approach is required. There is nothing one can do, in a high-uncertainty regime just after an earthquake, as already explained for **Points R1.6 and R1.7**, to be 100% sure that there will not be, for example, an underestimation. We have shown that this happens also with the envelope approach similar to that applied by the Chilean system (Fig. 6). For the “does deal” part, it is then perhaps important to know how many times it is expected to be “wrong” or, in other words, if there will be a bias and how large is the uncertainty “bar”. And to adjust, if possible, knowing also what would be the “price to pay”, in terms of increased false alarm possibility. This is the core of the PTF.

On the other hand, it is true that if a tendency toward under/over prediction can be systematically observed, this can be used to calibrate the model. The fact that collectively the performance is promising, as the reviewer acknowledges, is important and this is one of the messages that we intend to pass to the readers. This is now commented on lines 347-349.

On the other hand, as noted also later by the reviewer, there are hints about possible magnitude-dependent tendency to under/over estimation. We comment further on this later on, discussing Fig. 3 and 4 (**Point R1.12_Fig3vsFig4**). We anticipate that this is discussed in lines 297-312. We think that this kind of hints can be very important for focusing future model testing, to improve calibration in both tsunami modelling and uncertainty treatment, as commented both in “Results” (lines 312-315) and in “Discussion” (lines 487-489).

POINT R1.9

But the “how to deal” part poses the following questions: Do you expect to have a standard threshold for the decision maker? If yes, how do you expect it can be defined? If not, how do envision someone can make a decision with these data? How would they know where do they fall in terms of overprediction or underprediction, in the absence of data (you talk about 2 mins after getting the EQ data, so is is full forecasted information)? Again, I am aware that you intend to defer that decision outside of the scope of the article, but I think it is very relevant that you could at least provide some guidance. Otherwise, we haven’t learn how to deal with the uncertainties, and its usefulness remains questionable.

It should be clear at this point that there cannot be a standard for the threshold, or at least we cannot impose it, and nonetheless that the guidance for a decision is to be made in advance.

For example, Fig. 6 shows that the three non-deterministic approaches directly derived from actual operational definitions in different TEWSs lead to different results. By comparing them with PTF, we can quantify the level of conservatism of the choices behind the definition of such methods. Starting from PTF, a quantitative and explicit choice about the level of conservatism could have been selected explicitly.

Does this make things simpler? Not necessarily (sometimes not knowing a certain aspect makes things simpler). Indeed, selecting a threshold is not trivial at all. However, this allows for differentiating the choice depending on the target risk as well as on the risk-reduction action that one wants to address.

We now clarify these points in “Discussion” (line 445-456).

POINT R1.10

The approach to analyze some of the results can also be seen here: In analyzing the specific case of Zemmouri-Boumerdes, the authors claim “a few observations from Eastern Spain ...” (L230). I counted that 6 out of 16 fit the criteria. That is 38%, hardly “a few”. Similarly, in L232-233 they claim that “simulations ...largely fall within PTF’s inner confidence interval”. Rather than largely, I think it is “barely”, as 7/16 of the data are clearly in the outer confidence intervals in Fig 3 bcd (this is very different for Fig.4, but I digress). What I mean is that in the text, the more optimistic and uplifting tone for the work results is often present. This also may explain overlooking that in Fig. 2c, the PTF P95 best matches quite well the observations (bar overpredicting in the near field). This can be seen to lend support to the system that aims for the worst case scenario, (not the BSM!!). This is not discussed.

Agree, we now try to be more careful in the wording in the discussion of the results. And, as already discussed, we are not arguing against tending toward the “worst-case” with conservative choices. We simply say that this should be discussed and the impact of different choices be quantified.

For the specific case, the Zemmouri-Boumerdes tsunami is indeed challenging, as now commented on line 233-234, because it is a relatively small earthquake but with a steep faulting angle compared to a subduction interface one, so it is relatively more tsunamigenic, and with the short-wavelengths resonating more with the harbour eigen-frequencies. It is a small tsunami with significant amplifications. We did not perform (probably nobody did) high-resolution modelling; so, this is a specific technical limitation at some location that cannot be used to judge the overall performance of the method.

The sentence describing this case as challenging was in the first version of the manuscript but was removed during the revision. We now re-added it. For this reason, in this case, any conservative approach (ENV, DM, or high percentiles of PTF) may work better at some locations, as now commented in lines 231-237 and 269-272. In general, these problems can only be attenuated with better local modelling or calibration, which are both complementary aspects to PTF. This is true for any forecasting method, and improvements will be happening more and more in the future due to the development of computational science and technology. Yet, a robust probabilistic approach will remain useful.

Nevertheless, we also tried to remove all potential wording leading to any “optimistic bias”. More specifically, regarding the sentence here criticised, we put “several” instead of “a few” (line 263). “Largely” was referred to the “numerical simulations from finite source models” (red lines) and not to the observations (yellow dots). To avoid any further confusion, 1) we separated the two comparisons (against observations and against numerical simulations) into two paragraphs, 2) we changed the reference to the second confidence interval (5-95 percentiles), which includes all simulations curves (red curves in Fig. 3), and we use “generally” instead of “largely”. All these changes are in lines 250-256.

POINT R1.11

The other aspect is that one of the premises of the work is insufficiently backed up. The excessive rate of false alarms is mentioned as something to be avoided several times in the text, and guides the narrative. I note the reference is Bernard and Titov, 2015, but I think they were reporting older, before 2004, systems. Whether this issue has been quantified in modern systems I am not aware, I confess. However, what is absent in the article is what would be the effect of missed alarms, and whether that cost can be afforded by society. In some countries, missed alarms mean lives. As before, I am aware that the authors would like to defer this decision to others, but it leaves the premise for the work on unstable ground.

Actually, Bernard and Titov deal with pre- and post-2004 systems, as the paper was published in 2015, after Tohoku's event. Nevertheless, again the importance of this comment by the Reviewer is to be acknowledged, since it helped us to stay more focused on the approach itself, rather than on the potential consequences of a specific choice, as, for example, the possibility of controlling the reduction of false alarms, depending on the choice made by the decision makers.

Conversely, we do not agree that we should focus on the consequences of missed alarms for society. This is not a risk assessment. Neither do we leave “the premise for the work on unstable ground.”, as it is a fact that different present-day methods lead to different rates of correct/missed/alarms, and that conservative methods by definition lead to high rates of false alarms (the “Achille’s heel” of TEWS, using Reviewer’s words; P99,999999 indeed means P0.00001 of exceedance).

We have repeatedly commented on this previously. We also hope that at this point we have clarified how the decision makers could use the PTF input to serve their specific purposes, by knowing the expected proportions of false, missed, and correct alarms in advance.

POINT R1.12

I have some specific comments as well:

L42: The idea of excessive false alarms is introduced here. How does the method help in reducing these false alarms is somewhat presented later, and also the balance with conservatism. But what is the cost of missing alarms is not mentioned. I don't feel at ease giving a higher weight to false alarms over missed alarms, as it is implied in the text.

We agree with the reviewer that we do not have to judge if the rate of false alarms is excessive, as noted in **Point R1.6**. Indeed, it is not our intention to give more weight to false alarms over missed alarms, but simply to say that they should be quantitatively dealt with, and we revised all the text in this direction (see **Point R1.6** for more details). As already explained above (**Points R1.3** and **R1.11**), estimating the cost of missing/false alarms is beyond our scope.

L47 “directly constrain”: I'd say just "help to constrain". They are also subject to uncertainties.
Agreed. We removed “directly” and leaving “help constraining” (line 48).

L61: The TEWS of Chile, Catalan et al, 2020 (Coastal Engineering Journal) takes a different route. It is not considered in the analysis nor the review.

Agreed. Thanks, we have now implemented a third term of comparison for PTF, similar to the approach of Catalan et al. (2020) that we called ENV. See **Point R1.2**. To extend for non-subduction zones, the spatial selection criterion is extended to all directions. The concept is introduced here (line 62), while the implementation that we adopted is described later in “Results” (line 207-211). Afterwards, the three methods (DM, BMS, ENV) are commented in parallel for all relevant results.

L70-72: The author mention that "Yet, quantification of uncertainty is critical in managing such situations". I find this non sequitur, as if the incoming data is in error, the resulting analysis would also be in error, probabilistic or not. The resulting computed uncertainties can therefore be meaningless.

Agreed. We removed this example (lines 70-77).

L76: I don't follow what is meant by "single intensity" here.

We now specified that we talk about "a single estimation of the tsunami intensity" (line 78), making reference to the concept introduced in lines 39-42. Note that this definition includes both defining the tsunami intensity from a single simulation, or extracting the maximum from many.

L80: "However, this process may lead to non-rational choices". It is a strong sentence, and I don't see the non-rationality. Is trying to be conservative to save lives non-rational? Perhaps it did follow a rational assessment and it was found that the cost of false alarms can be afforded whereas no-action can not?

We agree, this could be misleading. We do not mean that being conservative is non-rational. We want to say that the desired choice can be more easily achieved if information regarding the uncertainty is available. We removed this sentence (lines 82-84), and we postponed the discussion about potential optimizations in the "Discussion" section (line 445-481), where we better specify the core of the question (as better discussed in **Point R1.3**).

L84: I think this a somewhat fallacious argument. Other phenomena can optimize the risk reduction management by using probabilities because the time-and spatial scales considered allow for that. For instance, in weather prediction. But a tsunami is very fast and wide ranging. Risk reduction during an emergency is performed by reducing exposure (i.e., evacuation) and every minute gained is precious. I am not so sure whether these tools really allow for such "optimization" in the case of tsunamis.

We respectfully disagree with the reviewer. The PTF does not consume time per-se: at most, the expert of the TEWS would spend some time to judge the seismic solution, as it may occur also with the present operational procedures. Conversely, the definition of the choices as a function of the forecast intensity distribution parameters occurs when plans are being set up, not during the emergency, as already discussed. Therefore, the risk reduction management is independent by how quickly the phenomenon evolves in real-time, at least concerning near-field warning. In this part of the text, we added a sentence at the end of the paragraph (lines 88-89).

This also affects to L109-116, where the authors compare with seismic design and PTHA. The difference is the time scale of evaluation, which allows for estimation of uncertain occurrence of events, that is, in terms of recurrence rates. In TEWS, the event has occurred and we need to deal with it. The mitigation strategies are significantly different, and the data sources, and methods might not be transferrable.

Yes, the time scale of evaluation is different, and the uncertainty is inherently epistemic since the event already occurred, but the logical scheme is similar: on one side, there is the scientific component that evaluates the hazard, on the other side there is the political authority that sets the thresholds (e.g. a specific average return period for the design). In these lines, we simply say that "the separation between scientific components [...] and political duties [...]" can be rooted in the definition of a probability threshold, which is what is done for seismic and tsunami building codes and for setting the evacuation areas for the warning in New Zealand and Italy (ref 58-61). This is now better specified and it is recalled when the same procedure is used for defining alert levels out of PTF (lines 114-117 and 199-202).

L94: Throughout the text the term "tsunami intensity" is used, but there can be many Tsunami Intensity Measures (TIMs). I gather most of the time the authors refer to tsunami amplitude at coastal points (peak coastal amplitude) and/or DARTs, but sometimes they use run-up. Perhaps it would be best to be specific in terms of the TIM of choice on each case. That being said, since they are not modeling run-up, it could be discarded altogether.

Agreed. Thank you for the suggestion. We changed "tsunami intensity" to "Tsunami Intensity Measure (TIM)" (lines 93-94), and we specified its use throughout the text (e.g., lines 97,154, 181-182, 193, etc., as well as Fig.2

caption). In the application, we select as TIM the maximum wave amplitude extrapolated at 1 m depth (line 153-154). For run-up, we now specify that “To compare with tsunami amplitude at the coast, run-up data are halved (Power et al. 2007 and reference therein)” in line 292-293. In any case, this was already said both in the Fig. 4 caption and in Text S6.

L200-212: I miss the analysis of the case that best matches observations: P95. It is fairly close to the envelope approach used in the Chilean TWS for example, which is very conservative.

Exactly, P95 (not P99) results close to the Chilean TWS! We now added this comparison, mentioning in parallel both P95 and ENV (line 231-237).

L214-217: Here lies a key problem: How does a decision maker would know some observations (or the tsunami itself) were particularly large? I find a bit baffling that this is not analyzed in detail with regards to the real-life benefit of the PTF under such circumstances.

We understand that also this concern is connected to the misunderstandings discussed at **Point R1.4**: the decision-makers cannot know in real-time, at least not before sending the first alert, as we are assuming to deal with near-field warning where typically tsunami observations are not yet available. To avoid - on the average! - as much as possible missed-alarm for particularly large values, the only option is to use a conservative method. This is now explicitly commented (line 235-237). However, as we show in our example in Figure 6, this carries along false alarms and in general overestimations.

L230: Are 6 out of 16 really "a few"?

Agreed. To minimize this is indeed not necessary at all (see **Point R1.10**). We now report several (line 263).

L229: What does it mean “well reproduced”? That they fall within the range of variability modelled? That they fit a given percentile? Too vague a statement.

We agree with the Reviewer that this part was not clear enough. Therefore, we revised this part (line 262-265), also by reorganizing its content (see **Point R1.10**).

L236: "it can reproduce the tsunami uncertainty" What do you mean by reproducing the uncertainty? is it the range of variability?

We agree with the Reviewer that this part was not clear enough. We now specify that “the source variability in the PTF ensemble and the lognormal distribution we use to quantify the uncertainty embed the tsunami source uncertainty, as quantified by the range of available finite-fault models” (lines 258-260).

L239-240: So the modeling does not reproduce phenomena that could lead to amplification, yet the approach is to reduce "conservatism" using a model that is incomplete. It seems risky, to say the least. It is true that some of the results do encompass these cases, but since you don't provide guidance regarding what would be the appropriate P% to be used, we don't know whether this is a problem, or not. This is fairly inconclusive.

This is a good point in fact. The issue of local amplification should be probably treated by differentiating the choices/threshold case by case, and/or by adopting higher resolution models locally. But we expect this to be an issue for all systems and all approaches. Improving the modelling tools - as long as the observing systems - to reduce uncertainty is complementary to PTF and always welcome! We recall that we use the Green's law here (reporting amplitudes at 1 m in front of the coast), as for example the JMA, and also Catalan (2020) does. And we are not aware of many other more advanced approaches that are already fully implemented worldwide. So, this is not a problem strictly inherent to PTF, and our method is not less complete than others in this respect.

Besides, we do not intend at all to suggest to reduce conservatism, as repeatedly stated in this point-by-point response. After careful calibration with observations from past events, and considering all the limitations, pros and cons in relation to the specific application, thanks to the explicit probabilistic treatment, the desired level of conservatism can be adopted. The existence of these amplifications, for example, may lead decision makers to adopt more conservative choices. More in general, in case the quality of modeling is not deemed sufficient one can still apply a high level of conservatism.

Generally speaking, regarding conservatism, we fully acknowledged quite upfront that we should have made a greater effort for the sake of clarity in this respect, and we think that this review solves this important issue.

Regarding lines 239-240 (old version), we modified the sentence “As they fall inside the tail of PTF distributions, only the alert level corresponding to conservative choices (high percentiles of the PTF) include such maxima [...]”, explicitly mentioning the need of conservative choices (see line 269-272).

L248: Are the references correct? There is an ample body of literature working on tsunami amplification (Eric Geist, Alex Rabinovich, and more recently the Chileans) but these references seem to point in other direction. The references were mainly connected to the treatment of uncertainty. We now revised the sentence accordingly (line 277).

L264: Are you modeling run-up or PCA (figure legend says Maximum amplitude)? if run-up is not modeled, perhaps is better not to consider it. It is a different TIM altogether. As commented for **Point R1.12_L94**, for run-up, “to compare with tsunami amplitude at the coast, run-up data are halved (Power et al. 2007 and reference therein)” (lines 292-293). This is commented also in Fig. 4 caption and in the Supplementary Text S6.

Fig 3 vs Fig 4: They offer a very contrasting performance of the method, where in the ZB case models tend to underestimate whereas in Maule is the opposite. Then, how can this be generalized to come with a certain degree of guidance on how to actually use the PTFs? Fig 4: Note that Catalan et al, 2020, also compare their method against Maule 2010.

We agree. This is an important point, thank you. This tendency was already commented in the text (line 297-299), trying to interpret this result. This may be a hint of a possible magnitude-dependent tendency to under / over estimation, with a tendency toward overprediction for larger magnitude events. A possible interpretation of this is provided in lines 299-312. As commented above (**Point R1.8**), this kind of hints is fundamental for future calibration of the model. Given the positive results of testing with our testing dataset, we think that more testing is required to confirm or better understand this tendency, and this may allow in the future a fine-tuning of the methods. This is now explicitly commented both in “Results” (lines 312-315) and in “Discussion” (lines 487-489).

As far as the comparison with Maule in Catalan et al. 2020 is concerned, we think that the results are coherent. Indeed, we already say that “PTF inner confidence intervals (15th-85th percentiles) encompass all the observations”. This is coherent with the fact that a conservative method like the one of Catalan et al. 2020 (which we found roughly correspondent to P95 of PTF, see **Point R1.12_L61**) overcomes all observations. We now comment that “This result is coherent with the results of Catalan et al. 43, who show that the scenario envelope includes observations.” (lines 296-297).

L267-268: "A possible reason ...wavelengths" This sentence is fairly speculative, and it is not backed up since Chile is also subject to active resonance and amplification, during several tsunamis: See Catalan et al, 2014, GRL, Cortes et al., 2017, Aranguiz et al, 2019. The authors imply that it affects more the events in the Mediterranean?

We now better specify that “A possible reason is that, for smaller earthquakes on steeper faults like this, local resonances and amplification play a more important relative role due poorly modelled for smaller tsunami wavelengths; for the Mw 8.7 Maule event, shelf and basin resonances occur also at longer periods 98, but they are well-captured on a 30 arc-sec grid (see Methods).”. We also added the citation to Aranguiz et al., 2019 (Ref. 98).

L296: "Tsunami data and forecasts are compared". I think it is relevant to specify that you compare against the mean (which is not a proper probability btw, so how does it fall within the PTF method is unclear). The result will vary if you compare against other PTF threshold of choice. (perhaps a sensitivity analysis is in order?)

We thank the reviewer also for this comment, which points out another potential source of misunderstanding. We do not compare against the mean. We first sample the full source uncertainty from the source factor, then we average the log-normal distribution that accounts for the uncertainty in the propagation factor, taking only its mean (these two factors are defined in Eq. 1). Finally, we stack over all target points. Therefore, only the uncertainty on the propagation is averaged. Conversely, the uncertainty on the source is fully considered. This procedure, described in “Method – Testing tsunami forecasts”, is required for preserving spatial correlations (spatially uncorrelated sampling of the log-normal would hide such spatial correlations).

Therefore, the testing considers the entire distribution, and it does not depend on a selected percentile. We now better specify the procedure in “Method – Testing tsunami forecasts” (lines 846-853), and we revised both Figure 5 and its caption, as this misunderstanding may have been caused by the labels and the caption of this figure, which were misleading indeed.

L321: Here we can see where the PTF is somewhat sugar coated. The simpler BMS yields results that are fairly comparable with most of the PTF results (Fig 6, Table S9), and even yields better results (on average) than the PTF-mean, that is the base comparison for Fig. 5. Most of the following discussion focuses on the DM method (L322-338), whereas the comparison with BMS is mostly colloquial and vague in the main text (eg, L341: "Nevertheless, with higher percentiles, PTF enable a reduction of missed alarms while keeping a high rate of correct assignments"). The reader is referred to table S9, (note the use of BGS instead of BMS there), where the differences are minimal when we reach P75 or above, but then we would need to compare against the worst case scenario, I would think. The wording in the text seems to obfuscate this.

This comment is probably connected to some of the mentioned misunderstandings (**Points R1.1**). We respectfully disagree with the Reviewer. As discussed in detail in **Points R1.3** and **R1.6**, the goal here is to demonstrate that the full variability among existing non-probabilistic methods is covered by the different percentiles of the PTF. If not, it would not be possible to make the choice of the conservatism level using the PTF (or any other probabilistic method). As said above, the conservatism level is instead implicitly assumed in standard non-probabilistic methods. Therefore, BMS is “fairly comparable” with all intermediate percentiles, and DM (and now ENV) is “fairly comparable” with high PTF percentiles. It would have been a problem if this was not the case! This is a very important result which is now better commented (see lines 390-402), within the revised discussion of Fig. 6, and recalled in the “Discussion” (lines 452-456).

For the comment on testing, note that, as said in **Point R1.12_L296**, the tests are not based on the PTF-mean but on the entire PTF ensemble of sources.

As the discussion of this important result probably created some of the core misunderstandings, we deeply revised it, as outlined in **Point R1.6**. In particular, we reorganized and shortened the text and we removed some potentially misleading parts (statistical test, references to optimizations, etc.). To make things simpler, we also reduced the number of percentiles considered in Fig. 6 and we added P99 to further explore the tail of the distribution.

L343: The sentence is vague, and the sample percentiles are inconsistent within the parenthesis, as four values are presented, and only three PXX values are mentioned.

Agreed. We simplified this part, leaving only the full range of variability of the percentages of both missed and false alarms (see lines 390-396).

L345-350: Although a more fair comparison with BMS is presented here, there are vague statements. For instance, what does it mean to "optimize performance " in L345? Next, PTF are mentioned to perform "as good as BMS" (so does BMS take the upper hand?)...., "but they have a rate of missed alarms similar to a conservative DM". I can not figure whether this is good or bad in the eyes of the authors (inconclusive). Also, I don't find the reported value of 5% missed alarms of the conservative DM (L347), and I read it to be about 55% in Fig. 6 and Table S9. I guess is a typo, but then they are not comparable.

We agree. As discussed in the previous comments, we revised this part, removing such types of comparisons, which may be misleading. As it regards the meaning of optimization and the scope of the comparison in Fig. 6, we already acknowledged that more attention to discuss this was required (see **Point R1.3**), and we revised the entire discussion accordingly. In this revision, all the mentioned vague statements have been removed. The rate of missed alarms for DM is 3% (table S9); “about 5%” was referred to the 80-85th percentile. In any case, also these numbers have now been removed from the text (lines 390-399).

L349-354: All of this is certainly correct, but other than stating that some approaches can be conservative what do we do with this? Also, I don't think is true that DM and BMS are the end members of the class. Clearly BMS performs as well as the PTF method for P60-P75 (you say so in L346). On the other hand, the PTF values for P55 or lower have larger missed alarms (certainly a bad thing?), and the PTF-mean has more false alarms. Hence, the BMS method falls somewhere in between, not an end member. The bottom panel of Fig.6 is thus misleading, and the PTF has a whole has a wide range of performance.

Thank you for pointing out that DM and BMS were not end-members. Now, we indeed comment on the fact that PTF also allows for going beyond the reference methods in both directions (lines 390-391). We thank the reviewer for this comment, as this is indeed an important aspect that we did not comment on so far.

For Fig. 6, we now better specify that we talk about the range of behaviours in terms of missed-vs-false-alarm ratio defined by “worst-case” oriented (DM and ENV) and “best-match” oriented (BMS) methods” (line 365-389). As already discussed, the goal is indeed to reproduce the full range, representing opposite attitudes in terms of conservatism. Thus, showing that PTF has “a large variability” (line 367) is indeed a good result.

Note also that Fig. 6b, as now noted in lines 376-377, reports the number of alerted points independently from the fact of having observations, differently from Fig. 6a that reports only the points with observations (being a compassion with data, line 352). Thus Fig. 6b includes some important additional information with respect to Fig. 6a, which we think is worth keeping.

L355-356: I strongly disagree with this statement, but I recognize that this is a subjective standpoint. As a scientist, I think we should be transparent (what you aim here), but also offer guidance in how to interpret and use the results. No action needed.

We agree that we should offer some guidance, and then such choices should emerge from a dialogue between decision-makers (who bring further competences) and tsunami scientists. Here, our main point is simply that decisions on risk reduction actions need also competencies outside the field of tsunami science, which is something we all acknowledge, we think. Thus, we reviewed this sentence to make this clear (lines 456-457). In any case, note that we moved the sentence to Discussion, as to demonstrate this is not a goal of this manuscript (**Point R1.6**).

L355-356: What do you think would be the Key Performance Indicators in such a cost-benefit analysis?

Some examples are available in literature. Marzocchi et al. (2015; SRL; doi: 10.1785/0220150129) and Iervolino et al. (2015; BSSA, <https://doi.org/10.1785/0120140344>), for operational short-term seismic forecasts, proposed to use “individual risk of death (IRD).” The same papers discuss the thresholds suggested or adopted in different frameworks (e.g., World Health Organization) and countries (e.g., Hong Kong, Switzerland, Western Australia, the Netherlands, Iceland), which can also be adopted in this case. Woo (2007; <https://doi.org/10.1007/s11069-007-9171-9>) and Marzocchi and Woo (2007; <https://doi.org/10.1029/2007GL031922>) instead discussed probability thresholds for evacuation in case of volcanic eruptions, placing a “standard minimum economic value on human losses.”

However, such cost/benefit analysis is outside our field of expertise. Thus, we integrated this part in the discussion, adding the mentioned references, and removing references to specific techniques like those for decision-making and cost-benefit analyses (line 464-467). As discussed above (e.g., **Points R1.3 and R1.6**), the goal of this manuscript is not to define this strategy but it is to open the possibility for this to happen.

L361: What if a version of the cost-benefit analysis has been done by decision makers of systems elsewhere and the result is that false alarms are acceptable, but missed alarms are not? Is there evidence that this analysis has not been considered? (the only tsunami reference here is somewhat dated with regards to current TEWS)

This is certainly possible. But probably a quantitative approach to the forecast is required anyway, as we have shown, to explicitly quantify the expected rate of false, missed and true alarms. Also, as discussed above (**Point R1.3**), a quantitative approach enables an explicit control of the level of conservatism, making possible the choice of different levels of conservatism for different actions (beyond evacuation) and areas (distinguishing based on the consequent risk, for example). Nevertheless, this part has been moved to discussion and revised (lines 462-467).

We are not aware of references for tsunamis in which this analysis has been discussed explicitly, apart from the one cited. We also added the references mentioned in **Point R1.12_L355-356** and some other references derived from the seismological and volcanological communities (Goltz J.D. 2002; Marzocchi and Woo 2007; Woo 2008; Iervolino et al. 2007, 2015; Iervolino 2012). We would be happy to add more as we become aware of any. This is a further reason why it is indeed important to open this discussion within the tsunami science and disaster risk management community.

L388: “This allow decision-makers to make more conscious decisions considering uncertainty”. In the way it is presented in this work, this seems aspirational rather than a reality. The method presents the uncertainty, but leaves completely unanswered the question of how it can be used. Therefore, to say that “it allows more conscious decisions” is unsupported. It doesn’t mention either when this decision can or should be taken. In real time? In advance, and have a threshold PXX% to be used throughout? Those are two of many options.

Yes, when there is no time for decisions as in the near-field, the decision is to be taken “automatically”, having a threshold that was fixed in advance. We largely reviewed and integrated this part, as already commented in **Points R1.3, R1.4, and R1.6**. In short, we better specify that these choices should be made in advance and applied to the real-time uncertainty quantification; and that the explicit quantification of uncertainty allows for selecting explicitly the average performance, allowing optimizing choices for each risk reduction action” (lines 454-456).

Notably, as already discussed (for example, in **Point R1.12_L61**), all present-day methods correspond to a single column in **Fig. 6**. This means that this decision is taken. However, being taken without a probabilistic method, the specific level of conservatism is not explicitly quantified, for example, evaluating the expected rate of missed and false alarms. Making this explicit allows for “more conscious decisions.” This is not “aspirational”: it is what Fig. 6 demonstrates. However, we simplified and reorganized the discussion, going more directly to the point (lines 445-467).

L394: "performance can be optimized". Again, what is the cost function to be optimized is left open. We don't know if such an optimal point even exists.

As we already commented (**Point R1.6 and R1.12_L355-356**), we revised this part, integrating to provide some examples of potential strategies to exploit probabilistic forecasts accounting for real-time uncertainty, leading to an optimization or risk reduction measures.

L400. Indeed, the results are statistically sound. But so it's the BMS, as per your own comparison. But statistical accuracy does not solve the problem to tell the people when to evacuate.

As already discussed (**Point R1.6**), we removed all this kind of comparisons (also removing the associated statistical test), since they could be misleading.

In general, in defining alert levels, BMS has performances comparable to PTF central values, as commented in lines 382-385, and different from higher percentiles. However, as no uncertainty is foreseen in BMS, the used testing procedure (which is based on sampling source uncertainty) is not applicable (see the answer to **R1.12_L296**).

We agree that statistical accuracy does not tell when to evacuate. However, it tells us that the tsunami forecasting procedure and related uncertainty are accurate, providing a robust basis to define alert levels. This means also that, when a rule is selected to define alert levels, we can estimate how many times (over multiple events) it is expected to call for an unnecessary evacuation (false alarms), or for a missing one, with respect to the number of times in which the evacuation would prove - retrospectively - appropriate. This is indeed an important input for decision making.

L410: "After further successful testing". What is the metric of success here? I don't know at this point whether the PTF has been shown to be successful. We have a new, very polished method. True. And what else?

We agree that this was not clear enough. Testing is a way to verify model calibration, and measure the robustness of forecasts. Increasing the testing dataset, new observations can be used to further test the forecast capability, eventually obtaining new specific indications (for example, more indications to discuss the differences observed between Zemmouri-Boumerdes and Maule, see **Point R1.12_Fig3vsFig4**), and surely many further cases of big events need to be analyzed for PTF calibration. The performance can be further tested by adopting the testing procedure defined here (e.g., Fig. 5) or even developing new tests. This concept is now explicitly commented both in "Results" (lines 312-315) and in "Discussion" (lines 487-489), and it is directly linked to the description of further potential extensions of testing reported in the main text (line 501-504).

L410: "exploited to dramatically improve emergency decision making" seems very aspirational, and it is not supported by the evidence presenter herein.

We removed this sentence.

L411: "science driven": I think this is technique-driven. The development of other TWS has also been supported by the science available at the time. Some of the basic geophysical science regarding earthquakes and tsunamis is similar among those other systems and this. The methods, models and computational power, have allowed several developments. But at the end of the day, the question that underpins a TEWS is how to deal with lack of accuracy. What this articles provides is a sound methodology to have more data available for the decision, but whether its used, or whether it is useful, remains to be seen. "Science driven" was compared to the technologically-driven improvements of monitoring leading to uncertainty reduction and not to other TWS procedures. In any case, we removed both "science-driven" and "technologically-driven" (not necessary), and we reviewed the sentence to avoid misinterpretations (lines 473-479), adding also a further complementing development that regards the real-time computational capabilities (lines 476-477).

POINT R1.13

In summary, I think the methodology and approach used are sound, it is carried out very formally and properly, and attempt to maximize the amount of data available. I think this is worth to published. However, I think the article narrative points on a different way. As mentioned above, whether these data is what is needed or how it can or should be used, the manuscript in the current state does not offer enough support to objectively validate it. Even if upon revision it is found that these data are useful within a TEWS context, the authors should consider to explain or provide actual guidance regarding how it could be used. As of now, this is left unanswered and many of the somewhat grand claims of the article lack sufficient support as of now. I recommend a revision aimed at clarifying the article's goals, and how to validate them properly within the analysis and discussion.

We deeply discussed all these points above. We think that this review clarified the narrative of this paper, with discussions more focused on the specific goals of the manuscript, which are now better defined upfront. Answering reviewer's comments, we also eliminated several sources of potential misunderstanding and potential unintentional (as well as useless) "optimistic bias" in commenting the results. Overall, the manuscript has clearly improved and we thank the reviewer for this.

Reviewer #2

This version has several significant improvements, as noted previously the authors are presenting a novel methodology for addressing the establishment of tsunami threat and warning. It will influence the field of Tsunami Early Warning Systems. By including more recent events in the Mediterranean, as well as the Maule event, the analysis and discussion of the proposed methodology is strengthened.

A few points...

Lines 17-18 say: Tsunami warning centres face the challenging task of rapidly forecasting impending tsunamis immediately after an earthquake, I recommend saying "Tsunami Warning Centres face the challenging task of rapidly forecasting tsunami threat immediately after an earthquake"...

Corrected, thanks (lines 17-18; line numbers are referred to the track-changes version).

Lines 27-28 -national italian Tsunami Warning Centre if national is part of the name of the Italian Tsunami Warning Centre, the "n" should be capitalized, if not "national" can be removed as it is implied in the name.

We removed national, since it is not part of the name (line 28).

Lines 33-34. Text says "Tsunami Early Warning Systems (TEWS) must forecast the expected tsunami rapidly following any potentially tsunamigenic earthquake." consider "Tsunami Early Warning Systems (TEWS) must forecast the tsunami threat rapidly following any potentially tsunamigenic earthquake."

Corrected, thanks (line 34).

Lines Many TSPs and NTWCs use BM and DM processes just for initial forecasts, therefore lines 49 and 62 I suggest putting the word "initial" before the word forecasts.

Corrected, thanks (lines 50 and 64).

Line 63 - I think the strategies used by the NTWC and TSP are not "ad hoc" they have been developed based on the study of historical tsunamis and research.

We agree. We modified the sentence as follows: "Specific strategies (e.g., maximum credible magnitude, safety factors, etc.), usually rooted in the analysis of past events, are sometimes adopted to implicitly replace uncertainty quantification" (lines 64-66).

Lines 262 and 414 the word buoy after DART, the buoy is only the communication platform of the DART system).

Corrected (we removed "buoy"), thanks (lines 292 and 476).

Page 10. When referring to the new methodology as a move to "science driven paradigm shift", implies that current methods are not driven/based on "science", when this is not the case, eg. Green Lanes functions for computing coastal amplitudes, CMT solutions couple with DART observations, etc. –

This "science-driven" was intended to be opposite to the "technology-driven" mentioned afterward, more than to present-day practice. We reformulated the sentence: "This would represent a paradigm shift in the approach to tsunami warning, complementing the ongoing paradigm shift towards uncertainty reduction through enhanced real-time tsunami monitoring capability (GNSS, DART, SMART cables 6,21) [...]." (lines 473-476).

The article refers a lot to tsunami intensity, but this term is not defined, but based on its use it refers to the tsunami amplitude or run up. According to the UNESCO - IOC tsunami glossary, tsunami intensity is "Size of a tsunami based on the macroscopic observation of a tsunami's effect on humans, objects including various sizes of marine vessels, and buildings". Either define intensity for the purpose of the article or choose another term, eg water level.

We agree. Here, we are following the engineering community using “tsunami intensity” to indicate a scalar value extracted from tsunami time series used to describe the strength of the tsunami, in input to fragility and risk quantifications. Following reviewer #1 suggestion, we changed “tsunami intensity” to “Tsunami Intensity Measure (TIM)” (lines 93-94), and we specified its use throughout the text (e.g., lines 97,154, 181-182, 193, etc., as well as Fig.2 caption).

Christa von Hillebrandt-Andrade

Reviewer #3

The updated manuscript is presented more clearly and logically after the revision. In the methodology part, the authors added clarifications on the details of PTF implementation. They also explicitly discussed the behavior of missed alarms to different conservative choices. The quality of figures was greatly improved. For example, in Figure 5 (previously Figure 4), the author separated the earthquake events without evidence of a tsunami, and those with a tsunami, to test the PTF model again earthquakes. The revised graph is easy-understanding. Hence, I am satisfied with this version, and I recommend that this paper could be accepted.

I have one minor comment on the example of Tohoku earthquake (Line 69-71). I agree with the first reviewer, that the failure of tsunami forecasting of the Tohoku earthquake is mainly because of a poor assessment of earthquake. The earthquake was so large that seismometers at near-field stations were off-scale, making it unable to solve the CMT solution initially. This is more relevant to a problem of hardware device, rather than the lack of uncertainty quantification. The authors could consider removing this example to avoid misunderstanding.

In the new version, we removed this example, as suggested (lines 70-77 in the track-changes version of the manuscript).

REVIEWERS' COMMENTS

Reviewer #1 (Remarks to the Author):

This is my third review of the manuscript by Selva and collaborators. I commend the authors for an insightful discussion on their response letter, and also note that the article has been updated in a way that I think places the work more precisely within the overall context.

However, I still think there is one aspect that it is not properly addressed. At the beginning of the discussion, the authors propose a "We present a novel approach dealing with uncertainty in real-time tsunami forecasting and linking alert level definition for tsunami early warning to such uncertainty, coined Probabilistic Tsunami Forecasting (PTF). Current practices do not quantify uncertainty in tsunami forecasting and define alert levels deterministically. To reduce missed-alarms, they typically adopt safety factors that increase the number of false alarms. PTF addresses this issue through explicit uncertainty quantification, linking alert levels to the desired level of conservatism." (L368-373)

As I mentioned earlier, I do not agree the overall conceptual approach is exactly novel. I think this comes down to the wording and phrasing. In the referred sentences above, it is clear that the authors refer to the "novel approach dealing with uncertainty in real time", which is very explicit in the sentence and refers to the details of the method. But later on in the Discussion, the wording becomes more generic, and refers to the PTF approach (eg, Line 381 and 398), which is more generic. I think the PTF approach per se, as a concept, is not novel, albeit it has been dealt with rather crudely. This change of emphasis I think can be misleading.

Unfortunately, I omitted in previous review that the Australian Tsunami Warning System updated their original procedures (Allen and Greensdale, 2008, 10.1107/s11069000791808) by an improved version where they use 95% percentiles (Allen and Greensdale, 2010 10.5194/nhess-10-2631-2010). To reach that value, they do consider a measure of uncertainty, albeit cruder than in the present implementation. The same applies to the ENV method, which includes a set of candidate sources that are sampled, ranked and then maximum values are used. The implicit assumption is that uncertainties ought to be included, otherwise a BMS approach would have been used. These prior works do both processes and report the final decision, without providing too much detail to how the PTFs were estimated. We note that Allen and Greensdale provide cumulative probability density functions before determining their chosen 95% threshold (their Fig 2). Implicitly, one can guess that the ENV method does the same. So, with that in mind, the use of a probability density function as a proxy for the PTF appears to predate this work. However, I have recognized in my previous reviews that this work does provide a more closer look to how these are built, and it formalizes and provides more details than others on how they are estimated, especially through the PTF's source factor. This is relevant, because as science progresses, we could introduce newer methods and approaches to quantify this more precisely.

I note the authors do discuss about this on their reply letter (especially in the introduction in page 2, first three paragraphs, mentioning that " the scientific and political aspects are fused together and exerted at the same time. Probably, it is agreed in advance with the political authorities that the choice will be worst-case oriented" and "this is not equivalent to assigning a probability to each scenario outputs weighted according to their probabilities into the forecast, based on effective real time information". I think herein lies the crux of the problem. The two examples that use a measure of the pdf in their analysis do weight the forecasts, but giving equal weights to all sources, because in both cases the available real time information is/was scarce and they have a significant time constraint . Though already a 11 year old work, Allen and Greensdale argue that they have less than 10 min, and so is the case for Chile (eg Williamson and Newman, 2018, 10.1007/s00024-018-1898-6).

While the discussion provided in the rebuttal I think is very relevant, I do not find it well transcribed into the text, especially in the Discussion. I would recommend a minor revision, including the Australian TEWS and their details in the analysis for completeness, and also highlighting where the formalism introduced is placed relative to other methods, in the Discussion.

L274: typo: The magnitude is 8.8, not 8.7

Reviewer #1

Point R1.1

This is my third review of the manuscript by Selva and collaborators. I commend the authors for an insightful discussion on their response letter, and also note that the article has been updated in a way that I think places the work more precisely within the overall context.

However, I still think there is one aspect that it is not properly addressed. At the beginning of the discussion, the authors propose a "We present a novel approach dealing with uncertainty in real-time tsunami forecasting and linking alert level definition for tsunami early warning to such uncertainty, coined Probabilistic Tsunami Forecasting (PTF). Current practices do not quantify uncertainty in tsunami forecasting and define alert levels deterministically. To reduce missed-alarms, they typically adopt safety factors that increase the number of false alarms. PTF addresses this issue through explicit uncertainty quantification, linking alert levels to the desired level of conservatism." (L368-373)

As I mentioned earlier, I do not agree the overall conceptual approach is exactly novel. I think this comes down to the wording and phrasing. In the referred sentences above, it is clear that the authors refer to the "novel approach dealing with uncertainty in real time", which is very explicit in the sentence and refers to the details of the method. But later on in the Discussion, the wording becomes more generic, and refers to the PTF approach (eg, Line 381 and 398), which is more generic. I think the PTF approach per se, as a concept, is not novel, albeit it has been dealt with rather crudely. This change of emphasis I think can be misleading.

Unfortunately, I omitted in previous review that the Australian Tsunami Warning System updated their original procedures (Allen and Greensdale, 2008, 10.1107/s11069000791808) by an improved version where they use 95% percentiles (Allen and Greensdale, 2010 10.5194/nhess-10-2631-2010). To reach that value, they do consider a measure of uncertainty, albeit cruder than in the present implementation. The same applies to the ENV method, which includes a set of candidate sources that are sampled, ranked and then maximum values are used. The implicit assumption is that uncertainties ought to be included, otherwise a BMS approach would have been used. These prior works do both processes and report the final decision, without providing too much detail to how the PTFs were estimated. We note that Allen and Greensdale provide cumulative probability density functions before determining their chosen 95% threshold (their Fig 2). Implicitly, one can guess that the ENV method does the same. So, with that in mind, the use of a probability density function as a proxy for the PTF appears to predate this work. However, I have recognized in my previous reviews that this work does provide a more closer look to how these are built, and it formalizes and provides more details than others on how they are estimated, especially through the PTF's source factor. This is relevant, because as science progresses, we could introduce newer methods and approaches to quantify this more precisely.

In Allen and Greenslade (2010), the procedure is described as it follows (section 3.2) “*For each of the events described in Sect. 3.1, the closest T2 scenario was selected (see Table 1) and the 95th percentile value of H max within each coastal zone was found.*”. This means that, for each target event for which to make the forecast, they consider only one scenario (the closest to the available source solution; using our terminology the Best-Matching Scenario – BMS) and they look at the variability along the coast, picking the 95th percentile of this variability. Note that this is not to account for the uncertainty on the source and its impact in the forecast, but to manage the variability along the coast as alert levels are defined over large areas. This is very different from what we do (quantify the uncertainty on the forecast).

Point R1.2

I note the authors do discuss about this on their reply letter (especially in the introduction in page 2, first three paragraphs, mentioning that " the scientific and political aspects are fused together and exerted at the same time. Probably, it is agreed in advance with the political authorities that the choice will be worst-case oriented" and "this is not equivalent to assigning a probability to each scenario outputs weighted according to their probabilities into the forecast, based on effective real time information". I think herein lies the crux of the problem. The two examples that use a measure of the pdf in their analysis do weight the forecasts, but giving equal weights to all sources, because in both cases the available real time information is/was scarce and they have a significant time constraint . Though already a 11 year old work, Allen and Greensdale argue that they have less than 10 min, and so is the case for Chile (eg Williamson and Newman, 2018,10.1007/s00024-018-1898-6).

While the discussion provided in the rebuttal I think is very relevant, I do not find it well transcribed into the text, especially in the Discussion. I would recommend a minor revision, including the Australian TEWS and their details in the analysis for completeness, and also highlighting where the formalism introduced is placed relative to other methods, in the Discussion.

We added the citation to Allen and Greensdale in both the Results and the Discussion sections (lines 64, 69, and 397).

As suggested by the reviewer, we also better acknowledged the contributions of the Australian and the Chilean procedures as “precursors” of PTF. In particular, in Introduction, we explicitly reported their procedures as example about how, in current practice, a proxy of uncertainty can be derived (lines 67-69). In Discussion, we better marked the difference between such procedures and PTF (lines 395-401, 417-418), being more specific about PTF novelties.

Point R1.3

L274: typo: The magnitude is 8.8, not 8.7

Corrected, thanks.